# Constrained Linear Best Arm Identification with Covariate Selection

## Abstract

This paper studies a constrained linear best arm identification problem with covariate selection in the fixed-confidence setting, where each arm is evaluated across multiple performance metrics. The mean performance of each metric depends linearly on the feature vectors of both arms and covariates. The goal is to identify the arm with the highest expected value of one targeted metric while ensuring that the means of the remaining metrics stay below specified thresholds for each covariate. We first establish an instance-dependent lower bound on the sample complexity, formulated as a multi-level optimization problem that captures both feasibility and optimality. We then prove that this bound is tight by designing an algorithm that asymptotically matches it. Since the original algorithm is computationally intensive, we develop a relaxed version of the bound through a surrogate optimization problem and derive its convex dual. Using this bound, we propose a duality-based decomposition algorithm that is computationally efficient, updating only two coordinates and performing a single gradient step per iteration. We further show that the algorithm achieves the relaxed bound in theory and demonstrates its practical effectiveness through numerical experiments.

## 1 Introduction

Best arm identification (BAI) is a well-studied problem in machine learning, with broad applications in areas such as large language models (Shi et al., 2024), quantum computing (Wanner et al., 2025), and pharmaceutical development (Wang et al., 2024). This paper studies a constrained linear BAI problem with covariate selection. In this setting, each arm is evaluated across multiple performance metrics, where the mean of each metric is modeled as a linear function of feature vectors associated with both arms and covariates. Given a specific covariate, the goal is to identify the arm with the highest expected value in a target metric, while ensuring that the means of the remaining metrics remain below predefined thresholds. At each time step $t$, the agent selects an arm-covariate pair to sample and observes an independent random performance vector covering all metrics. In the fixed-confidence setting, the agent seeks to learn the underlying performance functions through sampling, identify the best arm for each covariate with probability at least $1 - \delta$, and minimize the total number of samples required.

Compared to the canonical BAI setting, constrained linear BAI with covariate selection is particularly well-suited for personalized decision-making problems. For example, in personalized medicine (Shen et al., 2021), each treatment option (arm) is associated with multiple performance metrics, such as therapeutic efficacy and side effects, which can only be observed through noisy clinical trial data. The mean outcome of each metric depends on both patient characteristics (covariates) and the chemical composition of the drug. The objective is to identify the drug with the highest expected efficacy while ensuring that the expected side effects remain below predefined thresholds. Similar scenarios arise in inventory management (Ban & Rudin, 2019), where metrics like revenue, lead time, and customer satisfaction depend on observable factors such as seasonality, economic indicators, and market conditions, as well as the chosen order quantity. The goal is to identify the order quantity that maximizes average revenue while ensuring that the mean values of the other metrics remain within acceptable limits.

Two key challenges set constrained linear BAI with covariate selection apart from the canonical BAI problem (Garivier & Kaufmann, 2016), making existing algorithms insufficient for this setting.

First, unlike the standard BAI framework, which focuses solely on identifying the optimal arm, the constrained version requires balancing both optimality and feasibility. This trade-off between optimality and feasibility requires new theoretical insights to understand its effect on sample complexity and to guide the design of optimal algorithms. Second, covariate selection introduces an additional layer of complexity. The agent must determine an optimal sampling rule over arm-covariate pairs at each iteration. In contrast, canonical linear BAI (Jedra & Proutiere, 2020) and contextual bandit settings (Slivkins et al., 2019) typically assume that covariates are passively observed, limiting the agent's control to selecting a single arm. As we demonstrate in this work, leveraging both linear structure and active covariate selection can significantly improve sampling efficiency and necessitates a fundamentally different algorithmic approach.

The contributions of this paper are summarized as follows:

- Motivated by practical personalized decision-making scenarios, we study a constrained BAI problem with covariate selection. We derive an instance-dependent lower bound on the sample complexity, formulated as a multi-level optimization problem, and characterize how both the feasibility and optimality of each arm influence this bound. Moreover, we demonstrate the tightness of this bound by constructing a Track-and-Stop algorithm whose sample complexity matches it asymptotically.

- Due to the computational intractability of the Track-and-Stop algorithm, we introduce a relaxed sample complexity bound derived from a surrogate optimization problem. We further derive its convex dual, which possesses favorable structural properties and can be solved efficiently. Notably, the dual formulation provides a closed-form mapping to the primal optimal solution and offers an intuitive interpretation of the optimal sampling ratio.

- Leveraging the specific structure of the dual problem, we propose a duality-based decomposition algorithm. This algorithm has two key features: first, it updates two coordinates of the dual solution at a time; second, it performs a one-step gradient descent at each iteration. These features contribute to its high efficiency. We theoretically demonstrate that the algorithm's sample complexity attains the relaxed bound and validate its practical effectiveness through numerical experiments.

Our study connects to three principal strands of the existing literature:

**Best Arm Identification.** BAI is one of the most extensively studied problems in the bandit literature (Audibert & Bubeck, 2010; Gabillon et al., 2012). This work contributes to the growing body of research on BAI in the fixed-confidence setting, also known as pure exploration (Kaufmann et al., 2016; Garivier & Kaufmann, 2016; Juneja & Krishnasamy, 2019; Degenne & Koolen, 2019), which focuses on deriving instance-dependent lower bounds on sample complexity and designing adaptive, asymptotically optimal algorithms (Degenne et al., 2019; Wang et al., 2021). Jedra & Proutiere (2020) extended these results to the linear BAI setting. Our formulation generalizes both the canonical and linear BAI problems as special cases. Furthermore, the proposed algorithm introduces a duality-based perspective, enhancing both efficiency and practicality compared to methods that rely on access to an optimization oracle.

**Constrained Best Arm Identification.** The multi-performance constrained BAI problem has received relatively limited attention in the literature. While recent studies have begun exploring multi-objective settings aimed at identifying the Pareto set (Kone et al., 2023; 2024b;a; 2025), these problems are fundamentally different from our constrained formulation, and the algorithms proposed in those works are not applicable to our setting. Yang et al. (2025) and Hu & Hu (2024) consider constrained BAI problems that are more closely related to ours. However, Yang et al. (2025) proposes a top-two Thompson sampling algorithm under a fixed-budget setting, without leveraging linear structure or considering covariate information, resulting in a simplified optimization problem compared to our setting. Meanwhile, Hu & Hu (2024) primarily focuses on risk constraints rather than the mean-based constraints studied here, and their algorithm is not readily adaptable to our framework.

**Covariate Selection.** Decision-making with covariate information has been a central research theme across various domains, including operations research (Bertsimas & Kallus, 2020), simulation optimization (Shen et al., 2021; Du et al., 2024), and bandit problems (Lattimore & Szepesvári, 2020; Kato & Ariu, 2021). However, the covariate selection problem studied in this paper differs from the classical contextual bandit setting, where covariates are observed passively and drawn randomly.

Kato et al. (2024) investigates covariate selection in the context of experimental design, focusing on minimizing the semi-parametric efficiency bound. In contrast, we extend the notion of covariate selection to the BAI setting, with the objective of maximizing the probability of correct identification.

## 2 PROBLEM FORMULATION

This section presents the formulation of the constrained BAI problem with covariate selection and introduces the notation used throughout the paper.

Consider $K$ different arms, denoted by $\mathcal{X} = \{x_1, \ldots, x_K\} \subset \mathbb{R}^{\mathcal{X}}$, where each arm is associated with a vector $x_i$. We assume a finite set of $M$ possible covariates, denoted by $\mathcal{C} = \{c_1, \ldots, c_M\} \subset \mathbb{R}^{\mathcal{C}}$. For problems involving continuous covariate spaces, it is common to discretize the feature space and group covariate values accordingly. The performance of arm $x_i$ under covariate $c_j$ is represented by a random vector $(F(x_i, c_j), G(x_i, c_j)) \in \mathbb{R}^2$, where $F(x_i, c_j)$ and $G(x_i, c_j)$ correspond to the objective-related and constraint-related performance metrics, respectively. The agent aims to solve the following stochastic optimization problem:

$$\max_{x_i \in \mathcal{X}} f(x_i, c_j) \triangleq \mathbb{E}[F(x_i, c_j)] \quad \text{s.t.} \quad g(x_i, c_j) \triangleq \mathbb{E}[G(x_i, c_j)] \leq b, \tag{1}$$

for all covariate $c_j \in \mathcal{C}$. For notational simplicity, we consider a single-constraint setting. Extending our theoretical results and algorithm to accommodate multiple constraints is straightforward (see Appendix A.4). A problem instance is defined as $\mathcal{P} = (f(x_i, c_j), g(x_i, c_j))_{x_i \in \mathcal{X}, c_j \in \mathcal{C}}$. To facilitate the analysis, we adopt the following standard assumptions, which are commonly used in the BAI literature.

**Assumption 1.** *The problem instance $\mathcal{P}$ belongs to the set $\mathcal{S}$ of instances such that, for each covariate $c_j \in \mathcal{C}$, there exists a unique best arm $x_{i^*(c_j)}$ that solves problem (1), and no arm lies exactly on the constraint, i.e., $g(x_i, c_j) \neq b, \forall x_i \in \mathcal{X}$.*

**Assumption 2.** *For each arm-covariate pair $(x_i, c_j) \in \mathcal{X} \times \mathcal{C}$, the mean performances are given by $f(x_i, c_j) = \theta^\top \phi(x_i, c_j)$ and $g(x_i, c_j) = \beta^\top \phi(x_i, c_j)$, where $\phi(\cdot, \cdot) : \mathcal{X} \times \mathcal{C} \to \mathbb{R}^D$ is a known feature map, and $\theta, \beta \in \mathbb{R}^D$ are unknown parameter vectors.*

**Assumption 3.** *The observed performances are given by $F(x_i, c_j) = f(x_i, c_j) + \epsilon_{ij}$ and $G(x_i, c_j) = g(x_i, c_j) + \epsilon'_{ij}$, where the noise terms $\epsilon_{ij}$ and $\epsilon'_{ij}$ are independent and identically distributed Gaussian random variables with mean zero and variance $\sigma_{ij}^2$.*

Assumption 1 is standard in the canonical BAI literature (Garivier & Kaufmann, 2016; Jedra & Proutiere, 2020) and can be relaxed by identifying $\epsilon$-optimal and feasible arms, as discussed in Appendix A.3. Assumption 2 imposes a linear relationship between the mean performances and feature vectors. Despite its simplicity, the linear model effectively captures structural relationships across arms and covariates, enhances interpretability, and is widely used in linear bandit problems (Soare et al., 2014; Jedra & Proutiere, 2020) as well as personalized medicine (Shen et al., 2021; Du et al., 2024). Lastly, the Gaussian noise assumption in Assumption 3 is a standard choice in classical linear regression and enables the derivation of closed-form sample complexity lower bound.

**Design points.** In this paper, we use a fixed set of design points, denoted by $\mathcal{Z} = \{z_1, \ldots, z_D\}$, to estimate $\theta$ and $\beta$. Each design point $z_h$ corresponds to an arm-covariate pair $(x_i, c_j) \in \mathcal{X} \times \mathcal{C}$, and we simplify the notation by writing $F(z_h) = F(x_i, c_j)$. The motivations for adopting a fixed set of design points can be categorized into three aspects. First, De la Garza (1954) shows that to estimate the $D$-dimensional parameters $\theta$ and $\beta$ via regression, sampling only $D$ design points captures the same amount of information as sampling more than $D$ points. Second, this formulation has been widely used in the transductive linear bandits literature (Fiez et al., 2019). Third, concentrating on a fixed set of $D$ design points allows for the decomposition of regression variance, which facilitates the design of efficient algorithms.

**Learning problem.** In the online setting, at each iteration $t$, the agent selects a design point $z_{h(t)} \in \mathcal{Z}$ to sample. It then observes a random performance vector $Z_t = (Z_t^{(1)}, Z_t^{(2)})$, drawn independently according to the distribution of the corresponding random vector $(F(z_{h(t)}), G(z_{h(t)}))$. An algorithm in this setting is characterized by three components: the sampling rule $\{z_{h(t)}\}_t$, which determines the design point to sample based on the historical sampling decisions and observations

up to time $t$; the stopping rule $\tau$, which decides when to terminate the algorithm based on the collected information; and the recommendation rule $\{x_{\hat{i}(c_j,\tau)}\}_{c_j \in \mathcal{C}}$, which specifies the recommended best arm for each covariate $c_j \in \mathcal{C}$. The goal is to find a $\delta$-Probably Approximately Correct (PAC) algorithm (see Definition 1) while minimizing the sample complexity $\mathbb{E}[\tau]$.

**Definition 1** ($\delta$-PAC algorithm). *An algorithm $\mathcal{L} = (\{z_{h(t)}\}_t; \tau; \{x_{\hat{i}(c_j,\tau)}\}_{c_j \in \mathcal{C}})$ is said to be $\delta$-PAC if for every problem instance $\mathcal{P} \in \mathcal{S}$, it satisfies $\mathbb{P}_{\mathcal{P}}(\forall c_j \in \mathcal{C}, x_{\hat{i}(c_j,\tau)} = x_{i^*(c_j)}) \geq 1 - \delta$.*

**Notation.** For a positive integer $K$, let $[K] = \{1, \ldots, K\}$. Denote by $N_h(t)$ the number of samples drawn from design point $z_h$ up to time $t$, and define the corresponding sampling ratio $\omega_h(t) = N_h(t)/t$. Let $\Omega \triangleq \{\omega \in \mathbb{R}_+^D : \sum_{h \in D} \omega_h = 1\}$ denote the probability simplex over the design points. Let $\mathbb{I}(\cdot)$ denote the indicator function, which takes the value 1 if the condition is true, and 0 otherwise.

## 3 SAMPLE COMPLEXITY

In this section, we first derive a lower bound on the sample complexity. We then introduce a Track-and-Stop algorithm that asymptotically achieves this lower bound. However, this algorithm is computationally expensive, motivating the development of a duality-based approach. This perspective enables the design of a more efficient algorithm, which we present in the next section.

### 3.1 SAMPLE COMPLEXITY LOWER BOUND

This subsection presents a tight, instance-dependent lower bound on the sample complexity $\mathbb{E}[\tau]$, which provides a benchmark for evaluating the performance of any $\delta$-PAC algorithm.

The characterization of sample complexity relies on the transportation lemma from (Kaufmann et al., 2016), which establishes a relationship between the sample complexity, the Kullback-Leibler (KL) divergence between two problem instances, and the confidence level $\delta$. However, the constrained BAI problem with covariate selection is more challenging. Specifically, different types of arms contribute differently to the sample complexity depending on their feasibility and optimality. To capture this effect, we classify the arms into four categories for each covariate: the best arm $x_{i^*(c_j)}$, suboptimal feasible arms

$$\mathcal{D}_1(c_j) \triangleq \{x_i \in \mathcal{X} : f(x_i, c_j) < f(x_{i^*(c_j)}, c_j), g(x_i, c_j) \leq b\},$$

infeasible arms with better performance

$$\mathcal{D}_2(c_j) \triangleq \{x_i \in \mathcal{X} : f(x_i, c_j) > f(x_{i^*(c_j)}, c_j), g(x_i, c_j) > b\},$$

and infeasible arms with worse performance

$$\mathcal{D}_3(c_j) \triangleq \{x_i \in \mathcal{X} : f(x_i, c_j) < f(x_{i^*(c_j)}, c_j), g(x_i, c_j) > b\}.$$

Then, leveraging the linear structure in Assumption 2 and the Gaussian noise in Assumption 3, we derive a closed-form lower bound on the sample complexity in Theorem 1.

**Theorem 1.** *Under Assumptions 1-3, for a fixed confidence level $\delta \in (0, 1/2)$, any $\delta$-PAC algorithm applied to problem instance $\mathcal{P} \in \mathcal{S}$ must satisfy*

$$\mathbb{E}[\tau] \geq \mathcal{H}^*(\mathcal{P}) kl(\delta, 1 - \delta), \tag{2}$$

*which leads to*

$$\liminf_{\delta \to 0} \frac{\mathbb{E}[\tau]}{\log(1/\delta)} \geq \mathcal{H}^*(\mathcal{P}), \tag{3}$$

*where $\mathcal{H}^*(\mathcal{P})^{-1} = \max_{\omega \in \Omega} \min_{c_j \in \mathcal{C}} \Gamma(\omega, c_j, \mathcal{P})$,*

$$\Gamma(\omega, c_j, \mathcal{P}) = \min \left( \min_{x_i \neq x_{i^*(c_j)}} \left( \frac{((\phi(x_{i^*(c_j)}, c_j) - \phi(x_i, c_j))^\top \theta)^2}{\|\phi(x_{i^*(c_j)}, c_j) - \phi(x_i, c_j)\|_{\Lambda(\omega)^{-1}}^2} \mathbb{I}(x_i \in \mathcal{D}_1(c_j) \cup \mathcal{D}_3(c_j)) \right. \right.$$

$$\left. \left. + \frac{(b - \beta^\top \phi(x_i, c_j))^2}{\|\phi(x_i, c_j)\|_{\Lambda(\omega)^{-1}}^2} \mathbb{I}(x_i \in \mathcal{D}_2(c_j) \cup \mathcal{D}_3(c_j)) \right), \frac{(b - \beta^\top \phi(x_{i^*(c_j)}, c_j))^2}{\|\phi(x_{i^*(c_j)}, c_j)\|_{\Lambda(\omega)^{-1}}^2} \right),$$

$$\tag{4}$$

$\Lambda(\omega) = \sum_{z_h \in \mathcal{Z}} \frac{\omega_h}{2\sigma_h^2} \phi(z_h) \phi(z_h)^\top$, *and* $kl(\delta, 1 - \delta) \triangleq \delta \log(\delta/1 - \delta) + (1 - \delta) \log((1 - \delta)/\delta)$.

The derivation of the sample complexity result in Theorem 1 has an intuitive game-theoretic interpretation: the agent aims to select a randomized sampling strategy $\omega \in \Omega$ that maximizes the KL divergence between two instances, while the environment chooses an alternative instance $\tilde{\mathcal{P}}$ that is difficult to distinguish from $\mathcal{P}$. In the case of Gaussian noise, this formulation yields the closed-form expression in (58). Additionally, the sample complexity is influenced by the feasibility of the best arm $x_{i^*(c_j)}$, the performance of infeasible arms (both better arms in $\mathcal{D}_2(c_j)$ and worse arms in $\mathcal{D}_3(c_j)$), and the optimality of suboptimal feasible arms in $\mathcal{D}_1(c_j)$ as well as infeasible arms with worse performance in $\mathcal{D}_3(c_j)$.

Theorem 1 can be viewed as an extension of the linear BAI problem to the constrained setting with covariate selection. When the agent knows that all arms are feasible and there is only one covariate, Theorem 1 reduces to the sample complexity result in (Jedra & Proutiere, 2020), making it a special case of our framework.

### 3.2 SAMPLE COMPLEXITY UPPER BOUND

This section demonstrates the existence of an algorithm that asymptotically matches the sample complexity lower bound in Theorem 1 as $\delta \to 0$.

**Definition 2** (Asymptotic optimality). *An algorithm $\mathcal{L} = (\{z_{h(t)}\}_t; \tau; \{x_{\hat{i}(c_j), \tau}\}_{c_j \in \mathcal{C}})$ is said to be asymptotically optimal if for every problem instance $\mathcal{P} \in \mathcal{S}$, it is $\delta$-PAC and*

$$\limsup_{\delta \to 0} \frac{\mathbb{E}[\tau]}{\log(1/\delta)} \leq \mathcal{H}^*(\mathcal{P}). \tag{5}$$

The intuition behind the algorithm design is as follows. The sample complexity lower bound in Theorem 1 depends on the hardness of the problem instance $\mathcal{H}^*(\mathcal{P})$ and the confidence level $\delta$. The quantity $\mathcal{H}^*(\mathcal{P})$ is defined through an optimization problem that yields the optimal static sampling ratio

$$\omega^*(\mathcal{P}) = \arg\max_{\omega \in \Omega} \min_{c_j \in \mathcal{C}} \Gamma(\omega, c_j). \tag{6}$$

Therefore, an optimal algorithm must ensure that the empirical sampling ratio $\omega(t) = \{\omega_h(t)\}_{h \in [D]}$ converges to the optimal ratio $\omega^*(\mathcal{P})$.

Since the problem instance $\mathcal{P}$ is unknown, we must estimate it based on empirical observations. For each design point $z_h \in \mathcal{Z}$, define the empirical estimates of $F(z_h)$ and $G(z_h)$ up to time $t$ as

$$\bar{F}(z_h; t) = \frac{1}{N_h(t)} \sum_{s \leq t} Z_t^{(1)} \mathbb{I}(z_{h(t)} = z_h), \quad \bar{G}(z_h; t) = \frac{1}{N_h(t)} \sum_{s \leq t} Z_t^{(2)} \mathbb{I}(z_{h(t)} = z_h). \tag{7}$$

Then, the least squares estimators of the unknown parameters $\theta$ and $\beta$ up to time $t$ are given by

$$\hat{\theta}(t) = \Lambda(\omega(t))^{-1} \sum_{z_h \in \mathcal{Z}} \frac{\omega_h(t)}{\sigma_h^2} \phi(z_h) \bar{F}(z_h; t), \quad \hat{\beta}(t) = \Lambda(\omega(t))^{-1} \sum_{z_h \in \mathcal{Z}} \frac{\omega_h(t)}{\sigma_h^2} \phi(z_h) \bar{G}(z_h; t). \tag{8}$$

Using the least squares estimators in (8), we estimate $\mathcal{P}$ by $\hat{\mathcal{P}}(t)$, calculated from $\hat{\theta}(t)$ and $\hat{\beta}(t)$, and compute the corresponding empirical static ratio $\omega^*(\hat{\mathcal{P}}(t))$.

To ensure that the estimate $\hat{\mathcal{P}}(t)$ converges to the true problem instance $\mathcal{P}$, it is necessary to sample each design point infinitely often. Define the set of undersampled design points up to time $t$ as

$$\mathcal{B}_t = \{z_h \in \mathcal{Z} : N_h(t) < \sqrt{t} - D/2\}. \tag{9}$$

Consider the following sampling rule

$$z_{h(t+1)} = \begin{cases} \arg\min_{z_h \in \mathcal{B}_t} N_h(t) & \text{if } \mathcal{B}_t \neq \emptyset, \\ \arg\min_{z_h \in \mathcal{Z}} N_h(t) - t\omega_h^*(\hat{\mathcal{P}}(t)) & \text{otherwise} \end{cases}, \tag{10}$$

which continuously updates the estimate $\hat{\mathcal{P}}(t)$ and adaptively tracks the empirical static ratio $\omega^*(\hat{\mathcal{P}}(t))$. Under this rule, we can show that $\hat{\mathcal{P}}(t) \to \mathcal{P}$ and $\omega(t) \to \omega^*(\mathcal{P})$ as $t \to \infty$.

Finally, we apply the generalized likelihood ratio test method to ensure that the algorithm satisfies the $\delta$-PAC guarantee described in Definition 1. Define the stopping rule as

$$\tau = \inf\{t \in \mathbb{N} : t\mathcal{H}(\hat{\mathcal{P}}(t), \omega(t))^{-1} > \rho(t, \delta)\}, \tag{11}$$

where $\mathcal{H}(\hat{\mathcal{P}}(t), \omega(t))^{-1} = \min_{c_j \in \mathcal{C}} \Gamma(\omega(t), c_j, \hat{\mathcal{P}}(t))$. This rule ensures the algorithm terminates once the accumulated empirical evidence exceeds the confidence threshold $\rho(t, \delta)$, thus supporting the $\delta$-PAC guarantee and contributing to its asymptotic optimality, as shown in Proposition 1.

This algorithmic framework, known as Track-and-Stop, is widely used to address the BAI problem in various settings (Garivier & Kaufmann, 2016; Juneja & Krishnasamy, 2019; Jedra & Proutiere, 2020). Further details are provided in Algorithm 1.

---

**Algorithm 1:** Track-and-Stop Algorithm

**1 Input:** Covariate set $\mathcal{C}$, arm set $\mathcal{X}$, design point set $\mathcal{Z}$, confidence level $\delta$.
**2 Initialization:** Sample each design point $z_h \in \mathcal{Z}$ $n_0$ times.
**3** Set $t \leftarrow n_0 D$ and update $N_h(t), \omega_h(t), \hat{\mathcal{P}}(t), \Lambda(\omega(t))$.
**4 while** $t\mathcal{H}(\hat{\mathcal{P}}(t), \omega(t))^{-1} < \rho(t, \delta)$ **do**
**5**      **if** $\mathcal{B}_t \neq \emptyset$ **then**
**6**          $z_{h(t+1)} = \arg\min_{z_h \in \mathcal{B}_t} N_h(t)$
**7**      **else**
**8**          $\omega^*(\hat{\mathcal{P}}(t)) \leftarrow \arg\max_{\omega \in \Omega} \mathcal{H}(\hat{\mathcal{P}}(t), \omega)^{-1}$
**9**          $z_{h(t+1)} = \arg\min_{z_h \in \mathcal{Z}} N_h(t) - t\omega_h^*(\hat{\mathcal{P}}(t))$
**10**      Sample the design point $z_{h(t+1)}$ and obtain the observation $Z_{t+1}$.
**11**      Set $t \leftarrow t + 1$, and update $N_h(t), \omega_h(t), \hat{\mathcal{P}}(t), \Lambda(\omega(t))$.
**12 return** For each covariate $c_j \in \mathcal{C}$, recommend the estimated best arm:
$$x_{\hat{i}(c_j;\tau)} = \arg\max_{x_i \in \mathcal{X}} \hat{\theta}(\tau)^\top \phi(x_i, c_j) \quad \text{s.t.} \quad \hat{\beta}(\tau)^\top \phi(x_i, c_j) \leq b$$

---

**Proposition 1.** *Under Assumptions 1-3, there exists a constant $C > 0$ such that, with the stopping rule in (11) and $\rho(t, \delta) = \log(Ct^\alpha/\delta)$, Algorithm 1 is asymptotically optimal up to $\alpha$.*

Proposition 1 follows directly by extending the proof technique of Jedra & Proutiere (2020). It shows that the sample complexity upper bound of Algorithm 1 matches the lower bound exactly, establishing its asymptotic optimality.

### 3.3 A DUALITY PERSPECTIVE

Although Algorithm 1 provides strong theoretical guarantees, it is impractical for implementation. The primary challenge arises from the fact that the lower bound involves a complex, multi-level optimization problem, which makes computing $\omega^*(\hat{\mathcal{P}}(t))$ at each iteration computationally prohibitive. Additionally, the presence of constraints and the linear structure complicates the analysis of the KKT conditions, unlike in the canonical BAI setting (Kaufmann et al., 2016), making it difficult to apply existing algorithms to our problem.

**Surrogate Objective Function.** We first introduce a surrogate objective function to reduce the computational burden. By merging the sets $\mathcal{D}_2(c_j)$ and $\mathcal{D}_3(c_j)$ for each covariate $c_j \in \mathcal{C}$ and focusing solely on the feasibility of the corresponding arms, we derive the following surrogate objective function for $\Gamma(\omega, c_j, \mathcal{P})$ in (58):

$$\Gamma^s(\omega, c_j, \mathcal{P}) = \min_{x_i \in \mathcal{X}} \left( \frac{((\phi(x_{i^*(c_j)}, c_j) - \phi(x_i, c_j))^\top \theta)^2}{\|\phi(x_{i^*(c_j)}, c_j) - \phi(x_i, c_j)\|_{\Lambda(\omega)^{-1}}^2} \mathbb{I}(x_i \in \mathcal{D}_1(c_j)) \right.$$
$$\left. + \frac{(b - \beta^\top \phi(x_i, c_j))^2}{\|\phi(x_i, c_j)\|_{\Lambda(\omega)^{-1}}^2} \mathbb{I}(x_i \in \{x_{i^*(c_j)}\} \cup \mathcal{D}_2(c_j) \cup \mathcal{D}_3(c_j)) \right). \tag{12}$$

Compared to the original objective function $\Gamma(\omega, c_j, \mathcal{P})$, the surrogate function $\Gamma^s(\omega, c_j, \mathcal{P})$ exhibits a better decomposition property, which can be leveraged to design a highly efficient algorithm.

**Lemma 1.** *Let $\mathcal{U}^*(\mathcal{P})^{-1} = \max_{\omega \in \Omega} \min_{c_j \in \mathcal{C}} \Gamma^s(\omega, c_j, \mathcal{P})$. Then, it holds that $\mathcal{H}^*(\mathcal{P}) \leq \mathcal{U}^*(\mathcal{P})$.*

Lemma 1 shows that the surrogate optimal value $\mathcal{U}^*(\mathcal{P})$ provides an upper bound for the optimal value $\mathcal{H}^*(\mathcal{P})$ under the original objective function. This implies that $\mathcal{U}^*(\mathcal{P})$ can serve as a relaxed performance measure for the algorithms. In Appendix A.7, we establish a constant relaxation gap, i.e., $\mathcal{U}^*(\mathcal{P}) \leq C\mathcal{H}^*(\mathcal{P})$ for some positive constant $C > 1$.

**Dual Optimization Problem.** Although the primal multi-level optimization problem

$$\max_{\omega \in \Omega} \min_{c_j \in \mathcal{C}} \Gamma^s(\omega, c_j, \mathcal{P}) \tag{13}$$

is complex; it admits a dual problem that can be efficiently solved using a decomposition algorithm.

**Theorem 2.** *The dual of the primal optimization problem in (13) is equivalent to*

$$\min_{\lambda} \mathcal{Q}(\lambda, \mathcal{P}) = - \sum_{h \in [D]} \sqrt{\sum_{i \in [K], j \in [M]} \lambda_{ij} \chi_h(x_i, c_j)}$$

$$s.t. \quad \sum_{i \in [K], j \in [M]} \lambda_{ij} = 1, \quad \lambda_{ij} \geq 0, \quad \forall i \in [K], j \in [M], \tag{14}$$

*where for each $c_j \in \mathcal{C}$,*

$$\chi_h(x_i, c_j) = \begin{cases} \dfrac{\sigma_h^2 \left[ (\Phi^\top)^{-1} (\phi(x_{i^*(c_j)}, c_j) - \phi(x_i, c_j)) \right]_h^2}{\left( (\phi(x_{i^*(c_j)}, c_j) - \phi(x_i, c_j))^\top \theta \right)^2} & \text{if } x_i \in \mathcal{D}_1(c_j), \\[2ex] \dfrac{\sigma_h^2 \left[ (\Phi^\top)^{-1} \phi(x_i, c_j) \right]_h^2}{(b - \beta^\top \phi(x_i, c_j))^2} & \text{if } x_i \in \{x_{i^*(c_j)}\} \cup \mathcal{D}_2(c_j) \cup \mathcal{D}_3(c_j), \end{cases} \tag{15}$$

*$\Phi$ is the $D \times D$ design matrix, and $[v]_h$ denotes the $h$th element of the vector $v$.*

The dual optimization problem in (14) is a convex optimization problem over the unit simplex, which can be efficiently solved using off-the-shelf gradient-based algorithms. The following Lemma 2 establishes that strong duality holds.

**Lemma 2.** *The primal optimization problem in (13) is convex, strong duality holds, and it admits a unique optimal solution.*

According to Lemma 2, given a dual optimal solution $\lambda^*$, an optimal static sampling ratio $\omega^*(\mathcal{P})$ can be recovered as follows:

$$\omega_h^*(\mathcal{P}) = \frac{\sqrt{\sum_{i \in [K], j \in [M]} \lambda_{ij}^* \chi_h(x_i, c_j)}}{\sum_{l \in [D]} \sqrt{\sum_{i \in [K], j \in [M]} \lambda_{ij}^* \chi_l(x_i, c_j)}}. \tag{16}$$

We provide an intuitive explanation of the optimal static sampling ratio $\omega^*(\mathcal{P})$. The optimal dual solution $\lambda^*$ represents the importance of each arm-covariate pair. The term $\chi_h(x_i, c_j)$ quantifies the benefit of sampling the design point $z_h$ for identifying a specific arm-covariate pair $(x_i, c_j)$. This quantity depends on the signal variance, the location in the feature space, and the optimality or feasibility gap. Consequently, the optimal sampling ratio must balance these factors, weighted by the relative importance of each arm-covariate pair, to minimize the overall sample complexity.

## 4 DUALITY-BASED DECOMPOSITION ALGORITHM

In this section, we introduce a duality-based decomposition algorithm based on Theorem 2. Furthermore, we demonstrate that this algorithm asymptotically achieves the relaxed sample complexity bound $\mathcal{U}^*(\mathcal{P}) \log(1/\delta)$.

Leveraging the specific structure of problem (14), we design a decomposition algorithm that updates two coordinates at a time to reduce computational complexity.

**Lemma 3.** *Let $\lambda$ be a feasible dual solution such that $\lambda_{mn} > 0$, for some $m \in [K], n \in [M]$. Then, $\lambda$ is a stationary point of problem (14) if and only if*

$$\nabla \mathcal{Q}(\lambda, \mathcal{P})^\top d \geq 0, \forall d \in \mathcal{D}^{m,n}(\lambda), \tag{17}$$

*where $\mathcal{D}^{m,n}(\lambda) = \{e_{ij} - e_{mn} : i \neq m \, or \, j \neq n\} \cup \{e_{mn} - e_{ij} : i \neq m \, or \, j \neq n, \lambda_{ij} > 0\}$, $e_{ij} \in \mathbb{R}^{KM}$ is obtained by letting $\lambda_{ij}$ equal to one and other elements equal to zero.*

Note that Lin et al. (2009) analyzes the decomposition structure of general singly linearly constrained problems with lower and upper bounds, and our dual problem (14) falls within this class. However, the problem is more challenging in our case because the problem instance $\mathcal{P}$ is unknown. Similar to Algorithm 1, we replace $\mathcal{P}$ with the estimated instance $\hat{\mathcal{P}}(t)$ to solve the empirical version of problem (14). Instead of performing full gradient descent to obtain the optimal static sampling ratio $\omega^*(\hat{\mathcal{P}}(t))$, we apply a single gradient step, alternating with the estimate update $\hat{\mathcal{P}}(t)$, which is sufficient to ensure asymptotic convergence while significantly reducing computational cost.

Algorithm 2 outlines the one-step gradient descent procedure. It begins by randomly selecting two coordinates and then determines a descent direction along with the corresponding maximal step size. If the decrease in the objective function exceeds a given threshold, the algorithm employs the canonical line search to determine the step size and update the dual solution. A feasible sampling ratio can then be computed using (16). We also compare the per-iteration complexity of Algorithm 1 and 2 (see Appendix A.12), showing that the proposed procedure is highly efficient.

---

**Algorithm 2:** One-Step Gradient Descent Algorithm

---

1 **Input:** Covariate set $\mathcal{C}$, arm set $\mathcal{X}$, design point set $\mathcal{Z}$, a small positive constant $\kappa_0$ and
$\eta < \frac{1}{KM}, \hat{\mathcal{P}}(t), \hat{\theta}(t), \hat{\beta}(t), \lambda(t-1)$.

2 **Initialization:** Let $x_{\hat{i}(c_j;t)} = \arg\max_{x_i \in \mathcal{X}} \hat{\theta}(t)^\top \phi(x_i, c_j)$ s.t. $\hat{\beta}(t)^\top \phi(x_i, c_j) \leq b$ for each
covariate $c_j \in \mathcal{C}$.

3 Randomly choose $(m(t), n(t))$ from $\{(i, j) : \lambda_{ij}(t-1) \geq \eta\}$.

4 Compute the descent direction $d(t)$, and determine the maximum step size $s^{max}$:

$$d(t), s^{max} = \underset{s \in \mathbb{R}_+, d \in \mathbb{R}^{KM}}{\arg\min} \, s\nabla \mathcal{Q}(\lambda(t-1), \hat{\mathcal{P}}(t))^\top d,$$

$$\text{s.t. } \lambda_{ij}(t-1) + sd_{ij} \in [0, 1], \forall i \in [K], j \in [M]$$

$$d \in \mathcal{D}^{(m(t), n(t))}(\lambda(t-1)).$$

5 Define $\mathcal{W}(t) = \nabla \mathcal{Q}(\lambda(t-1), \hat{\mathcal{P}}(t))^\top d(t)$.

6 **if** $\mathcal{W}(t) < \max\{-\kappa_0, -(\log t/t)^{1/4}\}$ *and* $s^{max}\mathcal{W}(t) < \max\{-\kappa_0, -(\log t/t)^{1/2}\}$ **then**

7 $\quad \lambda(t) = \lambda(t-1) + s(t)d(t)$ where $s(t) = $ LineSearch Algorithm $(s^{max})$

8 **else**

9 $\quad \lambda(t) = \lambda(t-1)$

10 **Return:** Sampling ratio $\gamma(\hat{\mathcal{P}}(t))$ calculated according to (16) based on $\lambda(t)$.

---

The one-step gradient descent idea has appeared in the simulation literature (Zhou et al., 2024; Du et al., 2024), but our approach differs in two key ways. First, we tackle a more complex constrained BAI problem with covariate selection, which has not been previously explored. Second, we analyze the algorithm in the fixed-confidence setting to assess its statistical validity and sample complexity, whereas existing work focuses on sampling ratio convergence under the fixed-budget setting.

The algorithmic framework is the same as Algorithm 1, except for a modified sampling rule:

$$z_{h(t+1)} = \begin{cases} \arg\min_{z_h \in \mathcal{B}_t} N_h(t) & \text{if} \quad \mathcal{B}_t \neq \emptyset \\ \arg\min_{z_h \in \mathcal{Z}} N_h(t) - t\gamma_h(\hat{\mathcal{P}}(t)) & \text{otherwise} \end{cases}, \tag{18}$$

where $\gamma(\hat{\mathcal{P}}(t)) = \{\gamma_h(\hat{\mathcal{P}}(t))\}_{h \in D}$ denotes the sampling ratio returned by Algorithm 2. To mitigate the effect of estimation error, $\lambda(t)$ is reset to $1/KM$ whenever the optimal arms are challenged. We refer to this algorithm as the duality-based decomposition algorithm. Theorem 3 shows that the algorithm asymptotically matches the relaxed bound $\mathcal{U}^*(\mathcal{P}) \log(1/\delta)$ on sample complexity.

**Theorem 3.** *Under Assumptions 1-3, the duality-based decomposition algorithm is δ-PAC and satisfies*

$$\mathbb{P}\left( \limsup_{\delta \to 0} \frac{\tau}{\log(1/\delta)} \leq \mathcal{U}^*(\mathcal{P}) \right) = 1, \limsup_{\delta \to 0} \frac{\mathbb{E}[\tau]}{\log(1/\delta)} \leq \mathcal{U}^*(\mathcal{P}). \tag{19}$$

## 5 NUMERICAL EXPERIMENT

In this section, we evaluate the practical performance of the proposed duality-based decomposition algorithm. Detailed parameter settings and pseudo-code are provided in Appendix A.13.

We consider a problem with two covariates, four arms, and one constraint. For the first covariate, there is one optimal arm and three suboptimal arms. For the second, there is one optimal, one suboptimal, and two infeasible arms, i.e., one with better performance and one with worse performance than the optimal arm.

Since no existing methods directly address our problem, we propose the following benchmarks for comparison: (1) **USR**: Allocate an equal number of samples to each design point. (2) **BCSR**: A modified Best Challenger algorithm (Garivier & Kaufmann, 2016) based solely on arm optimality, representing the state-of-the-art for BAI. (3) **GOSR**: A greedy algorithm for problem (13) that relies solely on arm optimality. (4) **GFSR**: A greedy algorithm for problem (13) that relies solely on arm feasibility. We refer to our proposed duality-based decomposition algorithm as **DSR**.

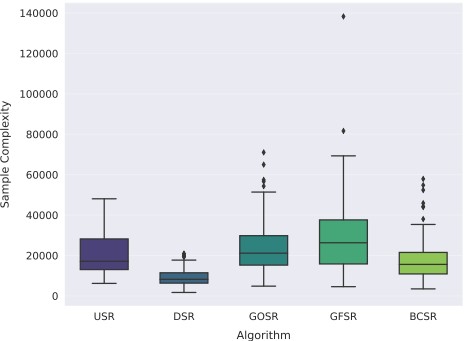

(a) Empirical sample complexity over 100 runs

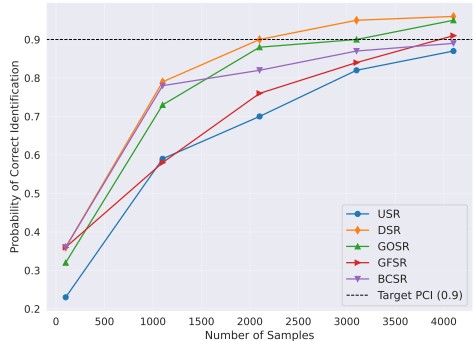

(b) Empirical PCI over 100 runs

Figure 1: Performance comparison of various algorithms

Figure 1 illustrates the empirical sample complexity and probability of correct identification (PCI) based on 100 independent macro-replications of various algorithms, with $\delta = 0.1$ and $n_0 = 1$. The results demonstrate that DSR achieves the lowest sample complexity among all benchmarks, with an average of 9205.46 samples. Furthermore, the findings highlight the statistical conservatism of the fixed-confidence setting: with 4000 samples, the empirical PCI of both DSR and GOSR exceeds the target PCI. Notably, the DSR algorithm outperforms all other benchmarks in terms of the PCI measure. This conclusion holds consistently across different problem instances and noise distributions (Appendix A.13). We also present an application example on personalized treatment for diabetes management in Appendix A.14, which verifies the practical performance of DSR.

## 6 CONCLUSION

This paper studies a constrained linear BAI problem with covariate selection, where each arm has multiple performance metrics, and the goal is to identify the best feasible arm per covariate. Our main contributions include an instance-dependent lower bound, a relaxed bound derived from a surrogate optimization problem, a duality-based formulation, and an efficient decomposition algorithm with theoretical guarantees. This work opens several avenues for future research, including extending the framework to continuous covariate spaces and generalizing the linear model to more flexible statistical structures, such as Gaussian Process Regression.

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

## A    Technical Appendices and Supplementary Material

### A.1    Large Language Models usage

ChatGPT was used for wording refinement and expression improvement.

### A.2    Proof of Theorem 1

*Proof.* To prove Theorem 1, we first introduce additional notation that was simplified or omitted in the main paper for clarity. Let $x_{i^*(c_j,\mathcal{P})}$ denote the best arm for covariate $c_j$ under the problem instance $\mathcal{P}$; when no ambiguity arises, we abbreviate this as $x_{i^*(c_j)}$. We define $d(f(z_h), \tilde{f}(z_h))$ as the KL divergence between two Gaussian random variables with means $f(z_h)$ and $\tilde{f}(z_h)$, sharing a common variance $\sigma_h^2$. The subscript $h$ indexes design points; for instance, if $z_h$ corresponds to the arm-covariate pair $(x_i, c_j)$, then $f(z_h) = f(x_i, c_j), \sigma_h^2 = \sigma_{ij}^2$.

A problem instance can be represented as $\mathcal{P} = (f(x_i, c_j), g(x_i, c_j))_{x_i \in \mathcal{X}, c_j \in \mathcal{C}}$. Consider the set of alternative instances

$$\mathcal{A}(\mathcal{P}) = \left\{ \tilde{\mathcal{P}} \in \mathcal{S} : \exists c_i \in \mathcal{C}, x_{i^*(c_j,\mathcal{P})} \neq x_{i^*(c_j,\tilde{\mathcal{P}})} \right\}, \tag{20}$$

which includes all problem instances $\tilde{\mathcal{P}} = (\tilde{f}(x_i, c_j), \tilde{g}(x_i, c_j))_{x_i \in \mathcal{X}, c_j \in \mathcal{C}}$ for which the optimal arm differs from that of $\mathcal{P}$ for at least one covariate.

In the fixed confidence setting, for a given confidence level $\delta \in (0, 1)$, the $\delta$-PAC condition requires that

$$\mathbb{P}_{\mathcal{P}} \left( \forall c_j \in \mathcal{C}, x_{\hat{i}(c_j,\tau)} = x_{i^*(c_j,\mathcal{P})} \right) \geq 1 - \delta, \tag{21}$$

and for any alternative instance $\tilde{\mathcal{P}} \in \mathcal{A}(\mathcal{P})$,

$$\mathbb{P}_{\tilde{\mathcal{P}}}\left(\forall c_j \in \mathcal{C}, x_{\hat{i}(c_j,\tau)} = x_{i^*(c_j,\mathcal{P})}\right) \leq \delta. \tag{22}$$

As the event

$$\left\{\forall c_j \in \mathcal{C}, x_{\hat{i}(c_j,\tau)} = x_{i^*(c_j,\mathcal{P})}\right\} \tag{23}$$

belongs to the filtration generated by all observations collected up to the stopping time $\tau$. Thus, applying the transportation inequality (Lemma 1) from Kaufmann et al. (2016), we obtain a fundamental information-theoretic lower bound:

$$\forall \tilde{\mathcal{P}} \in \mathcal{A}(\mathcal{P}), \sum_{h \in [D]} \mathbb{E}[N_h]\left(d(f(z_h), \tilde{f}(z_h)) + d(g(z_h), \tilde{g}(z_h))\right) \geq kl(\delta, 1 - \delta). \tag{24}$$

Consequently, we have the following sequence of inequalities:

$$\begin{aligned}
kl(\delta, 1 - \delta) &\leq \sum_{h \in [D]} \mathbb{E}[N_h]\left(d(f(z_h), \tilde{f}(z_h)) + d(g(z_h), \tilde{g}(z_h))\right) \\
&\leq \inf_{\tilde{\mathcal{P}} \in \mathcal{A}(\mathcal{P})} \sum_{h \in [D]} \mathbb{E}[N_h]\left(d(f(z_h), \tilde{f}(z_h)) + d(g(z_h), \tilde{g}(z_h))\right) \\
&\leq \sup_{\omega \in \Omega} \inf_{\tilde{\mathcal{P}} \in \mathcal{A}(\mathcal{P})} \sum_{h \in [D]} \mathbb{E}[N_h]\left(d(f(z_h), \tilde{f}(z_h)) + d(g(z_h), \tilde{g}(z_h))\right) \\
&= \mathbb{E}[\tau] \sup_{\omega \in \Omega} \inf_{\tilde{\mathcal{P}} \in \mathcal{A}(\mathcal{P})} \sum_{h \in [D]} \frac{\mathbb{E}[N_h]}{\mathbb{E}[\tau]}\left(d(f(z_h), \tilde{f}(z_h)) + d(g(z_h), \tilde{g}(z_h))\right) \\
&\leq \mathbb{E}[\tau] \sup_{\omega \in \Omega} \inf_{\tilde{\mathcal{P}} \in \mathcal{A}(\mathcal{P})} \sum_{h \in [D]} \omega_h\left(d(f(z_h), \tilde{f}(z_h)) + d(g(z_h), \tilde{g}(z_h))\right),
\end{aligned} \tag{25}$$

where $\omega_h = \mathbb{E}[N_h]/\mathbb{E}[\tau]$ represents the expected sampling proportion at design point $z_h$. This leads to the following lower bound on the sample complexity:

$$\mathbb{E}[\tau] \geq \mathcal{H}^*(\mathcal{P})kl(\delta, 1 - \delta), \tag{26}$$

where the instance-dependent complexity term is defined as

$$\begin{aligned}
\mathcal{H}^*(\mathcal{P})^{-1} &= \sup_{\omega \in \Omega} \inf_{\tilde{\mathcal{P}} \in \mathcal{A}(\mathcal{P})} \mathcal{H}(\omega, \mathcal{P}, \tilde{\mathcal{P}})^{-1} \\
&= \sup_{\omega \in \Omega} \inf_{\tilde{\mathcal{P}} \in \mathcal{A}(\mathcal{P})} \sum_{h \in [D]} \omega_h\left(d(f(z_h), \tilde{f}(z_h)) + d(g(z_h), \tilde{g}(z_h))\right).
\end{aligned} \tag{27}$$

For each covariate $c_j \in \mathcal{C}$, define the following sets:

$$\mathcal{O}(x_{i^*(c_j,\mathcal{P})}, c_j) = \left\{\tilde{\mathcal{P}} \in \mathcal{S} : \tilde{\beta}^\top \phi(x_{i^*(c_j,\mathcal{P})}, c_j) > b\right\}, \tag{28}$$

and

$$\mathcal{O}(x_i, c_j) = \left\{\tilde{\mathcal{P}} \in \mathcal{S} : \tilde{\theta}^\top(\phi(x_i, c_j) - \phi(x_{i^*(c_j,\mathcal{P})}, c_j)) > 0, \tilde{\beta}^\top \phi(x_i, c_j) \leq b\right\}. \tag{29}$$

Then, the set $\mathcal{A}(\mathcal{P})$ can be decomposed as

$$
\begin{aligned}
\mathcal{A}(\mathcal{P}) &= \left\{ \tilde{\mathcal{P}} \in \mathcal{S} : \exists c_i \in \mathcal{C}, x_{i^*(c_j, \mathcal{P})} \neq x_{i^*(c_j, \tilde{\mathcal{P}})} \right\} \\
&= \bigcup_{c_i \in \mathcal{C}} \left\{ \tilde{\mathcal{P}} \in \mathcal{S} : x_{i^*(c_j, \mathcal{P})} \neq x_{i^*(c_j, \tilde{\mathcal{P}})} \right\} \\
&= \bigcup_{c_i \in \mathcal{C}} \left( \left\{ \tilde{\mathcal{P}} \in \mathcal{S} : \tilde{\beta}^\top \phi(x_{i^*(c_j, \mathcal{P})}, c_j) > b \right\} \right. \\
&\qquad \left. \bigcup \left\{ \tilde{\mathcal{P}} \in \mathcal{S} : \exists x_i \in \mathcal{X}, \tilde{\theta}^\top (\phi(x_i, c_j) - \phi(x_{i^*(c_j, \mathcal{P})}, c_j)) > 0, \tilde{\beta}^\top \phi(x_i, c_j) \leq b \right\} \right) \\
&= \bigcup_{c_i \in \mathcal{C}} \left( \mathcal{O}(x_{i^*(c_j, \mathcal{P})}, c_j) \bigcup \left( \bigcup_{x_i \in \mathcal{X} \backslash x_{i^*(c_j, \mathcal{P})}} \mathcal{O}(x_i, c_j) \right) \right)
\end{aligned}
\tag{30}
$$

Then, we can express $\mathcal{H}^*(\mathcal{P})^{-1}$ as:

$$
\begin{aligned}
\mathcal{H}^*(\mathcal{P})^{-1} &= \sup_{\omega \in \Omega} \inf_{\tilde{\mathcal{P}} \in \mathcal{A}(\mathcal{P})} \mathcal{H}(\omega, \mathcal{P}, \tilde{\mathcal{P}})^{-1} \\
&= \sup_{\omega \in \Omega} \min_{c_j \in \mathcal{C}} \min \left( \inf_{\tilde{\mathcal{P}} \in \mathcal{O}(x_{i^*(c_j, \mathcal{P})}, c_j)} \mathcal{H}(\omega, \mathcal{P}, \tilde{\mathcal{P}})^{-1}, \min_{x_i \in \mathcal{X} \backslash x_{i^*(c_j, \mathcal{P})}} \inf_{\tilde{\mathcal{P}} \in \mathcal{O}(x_i, c_j)} \mathcal{H}(\omega, \mathcal{P}, \tilde{\mathcal{P}})^{-1} \right).
\end{aligned}
\tag{31}
$$

Next, we leverage the linear model structure and Gaussian noise assumptions from Assumptions 2 and 3 to derive a closed-form expression for $\mathcal{H}^*(\mathcal{P})$. Recall that for two univariate Gaussian distributions with equal variance, the KL divergence is given by

$$
d(f(z_h), \tilde{f}(z_h)) = \frac{(f(z_h) - \tilde{f}(z_h))^2}{2\sigma_h^2} = \frac{(\theta - \tilde{\theta})^\top \phi(z_h) \phi(z_h)^\top (\theta - \tilde{\theta})}{2\sigma_h^2}.
\tag{32}
$$

Using this result, the function $\mathcal{H}(\omega, \mathcal{P}, \tilde{\mathcal{P}})^{-1}$ admits the following closed-form:

$$
\mathcal{H}(\omega, \mathcal{P}, \tilde{\mathcal{P}})^{-1} = \sum_{h \in [D]} \omega_h \left( \frac{(\theta - \tilde{\theta})^\top \phi(z_h) \phi(z_h)^\top (\theta - \tilde{\theta})}{2\sigma_h^2} + \frac{(\beta - \tilde{\beta})^\top \phi(z_h) \phi(z_h)^\top (\beta - \tilde{\beta})}{2\sigma_h^2} \right).
\tag{33}
$$

We now consider the following sub-optimization problem:

$$
\begin{aligned}
&\inf_{\tilde{\mathcal{P}} \in \mathcal{O}(x_{i^*(c_j, \mathcal{P})}, c_j)} \mathcal{H}(\omega, \mathcal{P}, \tilde{\mathcal{P}})^{-1} \\
&= \inf_{\tilde{\beta}^\top \phi(x_{i^*(c_j, \mathcal{P})}, c_j) > b} \sum_{h \in [D]} \omega_h \left( \frac{(\theta - \tilde{\theta})^\top \phi(z_h) \phi(z_h)^\top (\theta - \tilde{\theta})}{2\sigma_h^2} + \frac{(\beta - \tilde{\beta})^\top \phi(z_h) \phi(z_h)^\top (\beta - \tilde{\beta})}{2\sigma_h^2} \right) \\
&= \inf_{\tilde{\beta}^\top \phi(x_{i^*(c_j, \mathcal{P})}, c_j) > b} \sum_{h \in [D]} \omega_h \frac{(\beta - \tilde{\beta})^\top \phi(z_h) \phi(z_h)^\top (\beta - \tilde{\beta})}{2\sigma_h^2} \\
&= \inf_{\tilde{\beta}^\top \phi(x_{i^*(c_j, \mathcal{P})}, c_j) > b} (\beta - \tilde{\beta})^\top \left( \sum_{h \in [D]} \omega_h \frac{\phi(z_h) \phi(z_h)^\top}{2\sigma_h^2} \right) (\beta - \tilde{\beta}) \\
&= \inf_{\tilde{\beta}^\top \phi(x_{i^*(c_j, \mathcal{P})}, c_j) > b} (\beta - \tilde{\beta})^\top \Lambda(\omega) (\beta - \tilde{\beta}),
\end{aligned}
\tag{34}
$$

where we define

$$
\Lambda(\omega) = \sum_{h \in [D]} \omega_h \frac{\phi(z_h) \phi(z_h)^\top}{2\sigma_h^2}.
\tag{35}
$$

Thus, the subproblem reduces to the following constrained quadratic minimization:

$$\inf_{\tilde{\beta}} \quad (\beta - \tilde{\beta})^\top \Lambda(\omega)(\beta - \tilde{\beta}) \tag{36}$$
$$\text{s.t.} \quad \tilde{\beta}^\top \phi(x_{i^*(c_j,\mathcal{P})}, c_j) > b \quad (\lambda)$$

The Karush–Kuhn–Tucker (KKT) conditions for the above optimization problem are given by

$$2\Lambda(\omega)(\beta - \tilde{\beta}) + \lambda \phi(x_{i^*(c_j,\mathcal{P})}, c_j) = 0 \tag{37}$$
$$\tilde{\beta}^\top \phi(x_{i^*(c_j,\mathcal{P})}, c_j) = b,$$

where $\lambda$ is the Lagrange multiplier associated with the inequality constraint. According to the first equation in (37), it holds that

$$\tilde{\beta} = \beta + \frac{1}{2}\lambda \Lambda(\omega)^{-1} \phi(x_{i^*(c_j,\mathcal{P})}, c_j). \tag{38}$$

Plug (46) into the second equation in (37), we have that

$$\lambda^* = \frac{2(b - \beta^\top \phi(x_{i^*(c_j,\mathcal{P})}, c_j))}{\|\phi(x_{i^*(c_j,\mathcal{P})}, c_j)\|^2_{\Lambda(\omega)^{-1}}}. \tag{39}$$

Plug (47) into (46) yields the optimal solution

$$\tilde{\beta}^* = \beta + \frac{b - \beta^\top \phi(x_{i^*(c_j,\mathcal{P})}, c_j)}{\|\phi(x_{i^*(c_j,\mathcal{P})}, c_j)\|^2_{\Lambda(\omega)^{-1}}} \Lambda(\omega)^{-1} \phi(x_{i^*(c_j,\mathcal{P})}, c_j). \tag{40}$$

The corresponding optimal value of the objective function is

$$\frac{(b - \beta^\top \phi(x_{i^*(c_j,\mathcal{P})}, c_j))^2}{\|\phi(x_{i^*(c_j,\mathcal{P})}, c_j)\|^2_{\Lambda(\omega)^{-1}}}. \tag{41}$$

Next, we consider the complementary sub-optimization problem

$$\min_{x_i \in \mathcal{X} \setminus x_{i^*(c_j,\mathcal{P})}} \inf_{\tilde{\mathcal{P}} \in \mathcal{O}(x_i, c_j)} \mathcal{H}(\omega, \mathcal{P}, \tilde{\mathcal{P}})^{-1}$$
$$= \min \left( \min_{x_i \in \mathcal{D}_1(c_j)} \inf_{\tilde{\mathcal{P}} \in \mathcal{O}(x_i, c_j)} \mathcal{H}(\omega, \mathcal{P}, \tilde{\mathcal{P}})^{-1}, \min_{x_i \in \mathcal{D}_2(c_j)} \inf_{\tilde{\mathcal{P}} \in \mathcal{O}(x_i, c_j)} \mathcal{H}(\omega, \mathcal{P}, \tilde{\mathcal{P}})^{-1}, \right. \tag{42}$$
$$\left. \min_{x_i \in \mathcal{D}_3(c_j)} \inf_{\tilde{\mathcal{P}} \in \mathcal{O}(x_i, c_j)} \mathcal{H}(\omega, \mathcal{P}, \tilde{\mathcal{P}})^{-1} \right).$$

Consider the analysis of the following optimization problem as an example:

$$\min_{x_i \in \mathcal{D}_1(c_j)} \inf_{\tilde{\mathcal{P}} \in \mathcal{O}(x_i, c_j)} \mathcal{H}(\omega, \mathcal{P}, \tilde{\mathcal{P}})^{-1}$$
$$= \min_{x_i \in \mathcal{D}_1(c_j)} \inf_{\tilde{\mathcal{P}} \in \mathcal{O}(x_i, c_j)} \sum_{h \in [D]} \omega_h \left( \frac{(\theta - \tilde{\theta})^\top \phi(z_h)\phi(z_h)^\top (\theta - \tilde{\theta})}{2\sigma_h^2} + \frac{(\beta - \tilde{\beta})^\top \phi(z_h)\phi(z_h)^\top (\beta - \tilde{\beta})}{2\sigma_h^2} \right)$$
$$= \min_{x_i \in \mathcal{D}_1(c_j)} \inf_{\tilde{\mathcal{P}} \in \mathcal{O}(x_i, c_j)} \sum_{h \in [D]} \omega_h \left( \frac{(\theta - \tilde{\theta})^\top \phi(z_h)\phi(z_h)^\top (\theta - \tilde{\theta})}{2\sigma_h^2} \right)$$
$$= \min_{x_i \in \mathcal{D}_1(c_j)} \inf_{\tilde{\mathcal{P}} \in \mathcal{O}(x_i, c_j)} (\theta - \tilde{\theta})^\top \Lambda(\omega)(\theta - \tilde{\theta}) \tag{43}$$

The inner optimization problem is therefore

$$\inf_{\tilde{\theta}} \quad (\theta - \tilde{\theta})^\top \Lambda(\omega)(\theta - \tilde{\theta}) \tag{44}$$
$$\text{s.t.} \quad \tilde{\theta}^\top (\phi(x_i, c_j) - \phi(x_{i^*(c_j,\mathcal{P})}, c_j)) \geq 0 \quad (\lambda)$$

The KKT conditions are given by

$$
\begin{aligned}
2\Lambda(\omega)(\theta - \tilde{\theta}) + \lambda(\phi(x_i, c_j) - \phi(x_{i^*(c_j, \mathcal{P})}, c_j)) &= 0 \\
\tilde{\theta}^\top(\phi(x_i, c_j) - \phi(x_{i^*(c_j, \mathcal{P})}, c_j)) &= 0
\end{aligned}
\tag{45}
$$

According to the first equation in (45), it holds that

$$
\tilde{\theta} = \theta + \frac{1}{2}\lambda\Lambda(\omega)^{-1}(\phi(x_i, c_j) - \phi(x_{i^*(c_j, \mathcal{P})}, c_j)).
\tag{46}
$$

Plug (46) into the second equation in (45), we have that

$$
\lambda^* = \frac{2(\theta^\top(\phi(x_{i^*(c_j, \mathcal{P})}, c_j) - \phi(x_i, c_j)))}{\|\phi(x_i, c_j) - \phi(x_{i^*(c_j, \mathcal{P})}, c_j)\|^2_{\Lambda(\omega)^{-1}}}.
\tag{47}
$$

Plug (47) into (46) yields the optimal solution.

$$
\tilde{\theta}^* = \theta + \frac{\theta^\top(\phi(x_{i^*(c_j, \mathcal{P})}, c_j) - \phi(x_i, c_j))}{\|\phi(x_{i^*(c_j, \mathcal{P})}, c_j) - \phi(x_i, c_j)\|^2_{\Lambda(\omega)^{-1}}}\Lambda(\omega)^{-1}(\phi(x_i, c_j) - \phi(x_{i^*(c_j, \mathcal{P})}, c_j)),
\tag{48}
$$

The corresponding optimal value is

$$
\frac{(\theta^\top(\phi(x_{i^*(c_j, \mathcal{P})}, c_j) - \phi(x_i, c_j)))^2}{\|\phi(x_{i^*(c_j, \mathcal{P})}, c_j) - \phi(x_i, c_j)\|^2_{\Lambda(\omega)^{-1}}}.
\tag{49}
$$

The analyses for the subproblems

$$
\min_{x_i \in \mathcal{D}_2(c_j)} \inf_{\tilde{\mathcal{P}} \in \mathcal{O}(x_i, c_j)} \mathcal{H}(\omega, \mathcal{P}, \tilde{\mathcal{P}})^{-1}
\tag{50}
$$

and

$$
\min_{x_i \in \mathcal{D}_3(c_j)} \inf_{\tilde{\mathcal{P}} \in \mathcal{O}(x_i, c_j)} \mathcal{H}(\omega, \mathcal{P}, \tilde{\mathcal{P}})^{-1}
\tag{51}
$$

follow analogous steps. Their optimal values are respectively

$$
\frac{(b - \beta^\top\phi(x_i, c_j))^2}{\|\phi(x_i, c_j)\|^2_{\Lambda(\omega)^{-1}}}
\tag{52}
$$

and

$$
\frac{(\theta^\top(\phi(x_{i^*(c_j, \mathcal{P})}, c_j) - \phi(x_i, c_j)))^2}{\|\phi(x_{i^*(c_j, \mathcal{P})}, c_j) - \phi(x_i, c_j)\|^2_{\Lambda(\omega)^{-1}}} + \frac{(b - \beta^\top\phi(x_i, c_j))^2}{\|\phi(x_i, c_j)\|^2_{\Lambda(\omega)^{-1}}}.
\tag{53}
$$

Finally, we conclude that $\mathcal{H}^*(\mathcal{P})^{-1} = \max_{\omega \in \Omega} \min_{c_j \in \mathcal{C}} \Gamma(\omega, c_j, \mathcal{P})$, where

$$
\begin{aligned}
\Gamma(\omega, c_j, \mathcal{P}) = \min\Bigg( &\min_{x_i \neq x_{i^*(c_j, \mathcal{P})}} \Bigg( \frac{((\phi(x_{i^*(c_j, \mathcal{P})}, c_j) - \phi(x_i, c_j))^\top\theta)^2}{\|\phi(x_{i^*(c_j, \mathcal{P})}, c_j) - \phi(x_i, c_j)\|^2_{\Lambda(\omega)^{-1}}}\mathbb{I}\big(x_i \in \mathcal{D}_1(c_j) \cup \mathcal{D}_3(c_j)\big) \\
&+ \frac{(b - \beta^\top\phi(x_i, c_j))^2}{\|\phi(x_i, c_j)\|^2_{\Lambda(\omega)^{-1}}}\mathbb{I}\big(x_i \in \mathcal{D}_2(c_j) \cup \mathcal{D}_3(c_j)\big), \Bigg) \frac{(b - \beta^\top\phi(x_{i^*(c_j, \mathcal{P})}, c_j))^2}{\|\phi(x_{i^*(c_j, \mathcal{P})}, c_j)\|^2_{\Lambda(\omega)^{-1}}}\Bigg).
\end{aligned}
\tag{54}
$$

$\square$

### A.3 Boundary Case Analysis

In this section, we relax Assumption 1 by identifying $\epsilon$-optimal and feasible arms. Specifically, our goal is to identify an $\epsilon$-optimal solution to the following optimization problem.

$$\max_{x_i \in \mathcal{X}} f(x_i, c_j) \quad \text{s.t.} \quad g(x_i, c_j) \le b + \epsilon$$

For each covariate $c_j \in \mathcal{C}$, define the following sets:

$$\mathcal{O}(x_{i^*(c_j, \mathcal{P})}, c_j) = \left\{ \tilde{\mathcal{P}} \in \mathcal{S} : \tilde{\beta}^\top \phi(x_{i^*(c_j, \mathcal{P})}, c_j) \ge b + \epsilon \right\}$$

and

$$\mathcal{O}(x_i, c_j) = \left\{ \tilde{\mathcal{P}} \in \mathcal{S} : \tilde{\theta}^\top (\phi(x_i, c_j) - \phi(x_{i^*(c_j, \mathcal{P})}, c_j)) > \epsilon, \tilde{\beta}^\top \phi(x_i, c_j) \le b + \epsilon \right\}.$$

Then, the set $\mathcal{A}(\mathcal{P})$ can be decomposed as

$$\mathcal{A}(\mathcal{P}) = \left\{ \tilde{\mathcal{P}} \in \mathcal{S} : \exists c_i \in \mathcal{C}, x_{i^*(c_j, \mathcal{P})} \ne x_{i^*(c_j, \tilde{\mathcal{P}})} \right\}$$

$$= \bigcup_{c_i \in \mathcal{C}} \left\{ \tilde{\mathcal{P}} \in \mathcal{S} : x_{i^*(c_j, \mathcal{P})} \ne x_{i^*(c_j, \tilde{\mathcal{P}})} \right\}$$

$$= \bigcup_{c_i \in \mathcal{C}} \left( \left\{ \tilde{\mathcal{P}} \in \mathcal{S} : \tilde{\beta}^\top \phi(x_{i^*(c_j, \mathcal{P})}, c_j) > b + \epsilon \right\} \right.$$

$$\bigcup \left\{ \tilde{\mathcal{P}} \in \mathcal{S} : \exists x_i \in \mathcal{X}, \tilde{\theta}^\top (\phi(x_i, c_j) - \phi(x_{i^*(c_j, \mathcal{P})}, c_j)) > \epsilon, \tilde{\beta}^\top \phi(x_i, c_j) \le b + \epsilon \right\} \right)$$

$$= \bigcup_{c_i \in \mathcal{C}} \left( \mathcal{O}(x_{i^*(c_j, \mathcal{P})}, c_j) \bigcup \left( \bigcup_{x_i \in \mathcal{X} \backslash x_{i^*(c_j, \mathcal{P})}} \mathcal{O}(x_i, c_j) \right) \right)$$

Then, we can express $\mathcal{H}^*(\mathcal{P})^{-1}$ as:

$$\mathcal{H}^*(\mathcal{P})^{-1} = \sup_{\omega \in \Omega} \inf_{\tilde{\mathcal{P}} \in \mathcal{A}(\mathcal{P})} \mathcal{H}(\omega, \mathcal{P}, \tilde{\mathcal{P}})^{-1}$$

$$= \sup_{\omega \in \Omega} \min_{c_j \in \mathcal{C}} \min \left( \inf_{\tilde{\mathcal{P}} \in \mathcal{O}(x_{i^*(c_j, \mathcal{P})}, c_j)} \mathcal{H}(\omega, \mathcal{P}, \tilde{\mathcal{P}})^{-1}, \min_{x_i \in \mathcal{X} \backslash x_{i^*(c_j, \mathcal{P})}} \inf_{\tilde{\mathcal{P}} \in \mathcal{O}(x_i, c_j)} \mathcal{H}(\omega, \mathcal{P}, \tilde{\mathcal{P}})^{-1} \right).$$

The following analysis follows the same approach as in Theorem 1. Therefore, we conclude that

$$\mathbb{E}[\tau] \ge \mathcal{H}^*(\mathcal{P}) \mathrm{kl}(\delta, 1 - \delta)$$

where $\mathcal{H}^*(\mathcal{P})^{-1} = \max_{\omega \in \Omega} \min_{c_j \in \mathcal{C}} \Gamma^\epsilon(\omega, c_j, \mathcal{P})$,

$$\Gamma^\epsilon(\omega, c_j, \mathcal{P}) = \min \left( \min_{x_i \ne x_{i^*(c_j, \mathcal{P})}} \left( \frac{(\epsilon + (\phi(x_{i^*(c_j, \mathcal{P})}, c_j) - \phi(x_i, c_j))^\top \theta)^2}{\|\phi(x_{i^*(c_j, \mathcal{P})}, c_j) - \phi(x_i, c_j)\|^2_{\Lambda(\omega)^{-1}}} \mathbb{I}(x_i \in \mathcal{D}_1(c_j) \cup \mathcal{D}_3(c_j)) \right.\right.$$

$$\left.\left. + \frac{(b + \epsilon - \beta^\top \phi(x_i, c_j))^2}{\|\phi(x_i, c_j)\|^2_{\Lambda(\omega)^{-1}}} \mathbb{I}(x_i \in \mathcal{D}_2(c_j) \cup \mathcal{D}_3(c_j)), \right) \frac{(b + \epsilon - \beta^\top \phi(x_{i^*(c_j, \mathcal{P})}, c_j))^2}{\|\phi(x_{i^*(c_j, \mathcal{P})}, c_j)\|^2_{\Lambda(\omega)^{-1}}} \right).$$

(55)

### A.4 Multiple Constraints Setting

In this section, we present the sample complexity lower bound for the multiple-constraint setting, which follows directly from an extension of the proof of Theorem 1. In the multi-constraint setting, each arm corresponds to a random performance vector $(F(x_i, c_j), G_1(x_i, c_j), \ldots, G_H(x_i, c_j))$, and the sample complexity must separately account for both feasible and infeasible constraints of each arm. Let $\mathcal{I}(x_i, c_j)$ and $\mathcal{F}(x_i, c_j)$ denote the index sets of infeasible and feasible constraints, respectively, for the arm-covariate pair $(x_i, c_j)$. For the $s$-th constraint of the arm-covariate pair $(x_i, c_j)$, the mean performance is given by $g_s(x_i, c_j) = \beta_s^\top \phi(x_i, c_j)$.

**Theorem 4.** *Under Assumptions 1-3, for a fixed confidence level $\delta \in (0, 1/2)$, any $\delta$-PAC algorithm applied to problem instance $\mathcal{P} \in \mathcal{S}$ must satisfy*

$$\mathbb{E}[\tau] \geq \mathcal{H}^*(\mathcal{P})kl(\delta, 1 - \delta), \tag{56}$$

*which leads to*

$$\liminf_{\delta \to 0} \frac{\mathbb{E}[\tau]}{\log(1/\delta)} \geq \mathcal{H}^*(\mathcal{P}), \tag{57}$$

*where $\mathcal{H}^*(\mathcal{P})^{-1} = \max_{\omega \in \Omega} \min_{c_j \in \mathcal{C}} \Gamma(\omega, c_j, \mathcal{P})$,*

$$\Gamma(\omega, c_j, \mathcal{P}) = \min \Bigg( \min_{x_i \neq x_{i^*(c_j)}} \Bigg( \frac{((\phi(x_{i^*(c_j)}, c_j) - \phi(x_i, c_j))^\top \theta)^2}{\|\phi(x_{i^*(c_j)}, c_j) - \phi(x_i, c_j)\|^2_{\Lambda(\omega)^{-1}}} \mathbb{I}\big(x_i \in \mathcal{D}_1(c_j) \cup \mathcal{D}_3(c_j)\big)$$

$$+ \sum_{s \in \mathcal{I}(x_i, c_j)} \frac{(b - \beta_s^\top \phi(x_i, c_j))^2}{\|\phi(x_i, c_j)\|^2_{\Lambda(\omega)^{-1}}} \mathbb{I}\big(x_i \in \mathcal{D}_2(c_j) \cup \mathcal{D}_3(c_j)\big) \Bigg), \min_{s \in \mathcal{F}(x_i, c_j)} \frac{(b - \beta_s^\top \phi(x_{i^*(c_j)}, c_j))^2}{\|\phi(x_{i^*(c_j)}, c_j)\|^2_{\Lambda(\omega)^{-1}}} \Bigg),$$
$$\tag{58}$$

$\Lambda(\omega) = \sum_{z_h \in \mathcal{Z}} \frac{\omega_h}{2\sigma_h^2} \phi(z_h)\phi(z_h)^\top$, *and* $kl(\delta, 1 - \delta) \triangleq \delta \log(\delta/1 - \delta) + (1 - \delta)\log((1 - \delta)/\delta)$.

Intuitively, arms from different classes are governed by different types of constraints. For the best arm, the lower bound is determined by the most critical feasible constraint, i.e., the one closest to violation. In contrast, for infeasible arms, the lower bound reflects the combined effect of all violated constraints.

## A.5    PROOF OF PROPOSITION 1

Proposition 1 follows directly by extending the proof of Theorem 3 in Jedra & Proutiere (2020). The only difference is that Jedra & Proutiere (2020) considered the case where the optimal sampling ratio $\omega^*(\mathcal{P})$ may be non-unique. Specifically, it proposed the following sampling rule:

$$z_{h(t+1)} = \arg\min_{z_h \in \mathcal{Z}} N_h(t) - \sum_{s=1}^{t} \omega_h^*(\hat{\mathcal{P}}(s)) \tag{59}$$

and showed that the empirical sampling ratio converges to the set $\mathcal{M}^*(\mathcal{P})$, defined as

$$\mathcal{M}^*(\mathcal{P}) \leftarrow \arg\max_{\omega \in \Omega} \mathcal{H}(\mathcal{P}, \omega)^{-1}. \tag{60}$$

This sampling rule in (59) can also be applied in our setting to handle the non-unique optimal sampling ratio case. Moreover, if all optimal sampling ratios can be enumerated, one may track a linear combination of them and apply the sampling rule in (10). Following the same analysis as in Lemma 4, we can show that $\mathcal{H}(\mathcal{P}, \omega)^{-1}$ is a continuous function with respect to $(\mathcal{P}, \omega)$. Moreover, $\Omega$ is a simplex, which is a compact, convex, and non-empty set. In addition, $\mathcal{H}(\mathcal{P}, \omega)^{-1}$ is concave with respect to $\omega$, because it can be expressed as the infimum over linear functions of $\omega$. By Berge's theorem, the solution set $\mathcal{M}^*(\mathcal{P})$ is convex, so any linear combination of elements in $\mathcal{M}^*(\mathcal{P})$ also belongs to $\mathcal{M}^*(\mathcal{P})$. Hence, this modification does not affect the convergence of the empirical sampling ratio $\omega(t)$.

### A.6 PROOF OF LEMMA 1

*Proof.* This lemma establishes that the relaxed complexity $\mathcal{U}^*(\mathcal{P})$ serves as an upper bound on the instance-dependent complexity $\mathcal{H}^*(\mathcal{P})$. Note that for each $\omega \in \Omega, c_j \in \mathcal{C}$, we have

$$
\begin{aligned}
\Gamma(\omega, c_j, \mathcal{P}) = \min \Bigg( & \min_{x_i \neq x_{i^*(c_j)}} \Bigg( \frac{((\phi(x_{i^*(c_j)}, c_j) - \phi(x_i, c_j))^\top \theta)^2}{\|\phi(x_{i^*(c_j)}, c_j) - \phi(x_i, c_j)\|^2_{\Lambda(\omega)^{-1}}} \mathbb{I}(x_i \in \mathcal{D}_1(c_j) \cup \mathcal{D}_3(c_j)) \\
& + \frac{(b - \beta^\top \phi(x_i, c_j))^2}{\|\phi(x_i, c_j)\|^2_{\Lambda(\omega)^{-1}}} \mathbb{I}(x_i \in \mathcal{D}_2(c_j) \cup \mathcal{D}_3(c_j)), \Bigg) \frac{(b - \beta^\top \phi(x_{i^*(c_j)}, c_j))^2}{\|\phi(x_{i^*(c_j)}, c_j)\|^2_{\Lambda(\omega)^{-1}}} \Bigg) \\
\geq & \min_{x_i \in \mathcal{X}} \Bigg( \frac{((\phi(x_{i^*(c_j)}, c_j) - \phi(x_i, c_j))^\top \theta)^2}{\|\phi(x_{i^*(c_j)}, c_j) - \phi(x_i, c_j)\|^2_{\Lambda(\omega)^{-1}}} \mathbb{I}(x_i \in \mathcal{D}_1(c_j)) \\
& + \frac{(b - \beta^\top \phi(x_i, c_j))^2}{\|\phi(x_i, c_j)\|^2_{\Lambda(\omega)^{-1}}} \mathbb{I}(x_i \in \{x_{i^*(c_j)}\} \cup \mathcal{D}_2(c_j) \cup \mathcal{D}_3(c_j)) \Bigg) \\
= & \; \Gamma^S(\omega, c_j, \mathcal{P}).
\end{aligned}
\tag{61}
$$

Then, we conclude that

$$
\mathcal{H}^*(\mathcal{P})^{-1} = \max_{\omega \in \Omega} \min_{c_j \in \mathcal{C}} \Gamma(\omega, c_j, \mathcal{P}) \geq \max_{\omega \in \Omega} \min_{c_j \in \mathcal{C}} \Gamma^S(\omega, c_j, \mathcal{P}) = \mathcal{U}^*(\mathcal{P})^{-1},
\tag{62}
$$

and therefore $\mathcal{U}^*(\mathcal{P}) \leq \mathcal{H}^*(\mathcal{P})$. $\qquad\square$

### A.7 RELAXATION GAP ANALYSIS

In this subsection, we analyze the gap between the relaxed bound $\mathcal{U}^*(\mathcal{P})$ and the original bound $\mathcal{H}^*(\mathcal{P})$.

Define the constant

$$
\gamma = \inf \left\{ \rho \in \mathbb{R}_+ : \frac{(b - \beta^\top \phi(x_i, c_j))^2}{\|\phi(x_i, c_j)\|^2_{\Lambda(\omega)^{-1}}} \rho \geq \frac{((\phi(x_{i^*(c_j)}, c_j) - \phi(x_i, c_j))^\top \theta)^2}{\|\phi(x_{i^*(c_j)}, c_j) - \phi(x_i, c_j)\|^2_{\Lambda(\omega)^{-1}}}, \forall x_i \in \mathcal{D}_3(c_j), c_j \in \mathcal{C} \right\}.
\tag{63}
$$

By definition of $\gamma$, it holds that

$$
\begin{aligned}
\Gamma(\omega, c_j, \mathcal{P}) = & \min_{x_i \neq x_{i^*(c_j)}} \Bigg( \frac{((\phi(x_{i^*(c_j)}, c_j) - \phi(x_i, c_j))^\top \theta)^2}{\|\phi(x_{i^*(c_j)}, c_j) - \phi(x_i, c_j)\|^2_{\Lambda(\omega)^{-1}}} \mathbb{I}(x_i \in \mathcal{D}_1(c_j) \cup \mathcal{D}_3(c_j)) \\
& + \frac{(b - \beta^\top \phi(x_i, c_j))^2}{\|\phi(x_i, c_j)\|^2_{\Lambda(\omega)^{-1}}} \mathbb{I}(x_i \in \mathcal{D}_2(c_j) \cup \mathcal{D}_3(c_j)), \frac{(b - \beta^\top \phi(x_{i^*(c_j)}, c_j))^2}{\|\phi(x_{i^*(c_j)}, c_j)\|^2_{\Lambda(\omega)^{-1}}} \Bigg) \\
\leq & (1 + \gamma) \min_{x_i \in \mathcal{X}} \Bigg( \frac{((\phi(x_{i^*(c_j)}, c_j) - \phi(x_i, c_j))^\top \theta)^2}{\|\phi(x_{i^*(c_j)}, c_j) - \phi(x_i, c_j)\|^2_{\Lambda(\omega)^{-1}}} \mathbb{I}(x_i \in \mathcal{D}_1(c_j)) \\
& + \frac{(b - \beta^\top \phi(x_i, c_j))^2}{\|\phi(x_i, c_j)\|^2_{\Lambda(\omega)^{-1}}} \mathbb{I}(x_i \in \{x_{i^*(c_j)}\} \cup \mathcal{D}_2(c_j) \cup \mathcal{D}_3(c_j)) \Bigg) \\
= & \; (1 + \gamma) \Gamma^S(\omega, c_j, \mathcal{P}).
\end{aligned}
$$

Then, it is easy to verify that

$$
\mathcal{U}^*(\mathcal{P}) \leq (1 + \gamma)\mathcal{H}^*(\mathcal{P})
\tag{64}
$$

by using the definition of $\mathcal{U}^*(\mathcal{P})$ and $\mathcal{H}^*(\mathcal{P})$. We use a numerical example to compare the approximation ratio $\Gamma(\omega, c_j, \mathcal{P})/\Gamma^S(\omega, c_j, \mathcal{P})$ under different values of the constraint threshold $b$. For each $b$, we randomly generate 1000 problem instances with $M = 2$ and $K = 4$. The expected objective and constraint values of all arms lie within $[0, 1]$. We then calculate $\Gamma(\omega, c_j, \mathcal{P})$ and $\Gamma^S(\omega, c_j, \mathcal{P})$ using a uniform sampling ratio $\omega$ for the first covariate. Figure 2 shows the average ratio under different constraint thresholds $b$. The results show that the approximation ratio is close to 1 as the constraint threshold $b$ increases.

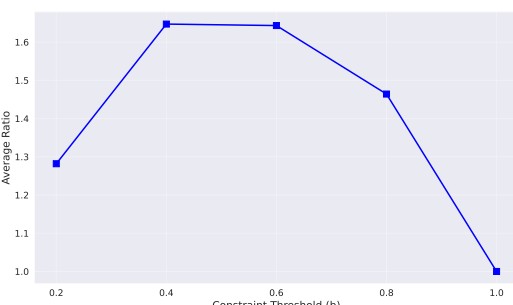

Figure 2: Average ratio under different constraint thresholds $b$

We also propose an alternative relaxed bound $\tilde{\mathcal{U}}^*(\mathcal{P})$ by partitioning the set $\mathcal{D}_3(c_j)$ into two subsets: $\mathcal{M}_1(c_j)$ and $\mathcal{M}_2(c_j)$ where arms in $\mathcal{M}_1(c_j)$ are relatively easy to identify as suboptimal, i.e.,

$$\mathcal{M}_1(c_j) = \left\{ x_i \in \mathcal{D}_3(c_j) : \frac{(b - \beta^\top \phi(x_i, c_j))^2}{\|\phi(x_i, c_j)\|^2_{\Lambda(\omega)^{-1}}} \leq \frac{((\phi(x_{i^*(c_j)}, c_j) - \phi(x_i, c_j))^\top \theta)^2}{\|\phi(x_{i^*(c_j)}, c_j) - \phi(x_i, c_j)\|^2_{\Lambda(\omega)^{-1}}} \right\}. \quad (65)$$

And arms in $\mathcal{M}_2(c_j)$ are easy to identify as infeasible, i.e.,

$$\mathcal{M}_2(c_j) = \left\{ x_i \in \mathcal{D}_3(c_j) : \frac{(b - \beta^\top \phi(x_i, c_j))^2}{\|\phi(x_i, c_j)\|^2_{\Lambda(\omega)^{-1}}} > \frac{((\phi(x_{i^*(c_j)}, c_j) - \phi(x_i, c_j))^\top \theta)^2}{\|\phi(x_{i^*(c_j)}, c_j) - \phi(x_i, c_j)\|^2_{\Lambda(\omega)^{-1}}} \right\}. \quad (66)$$

Based on this, we define a new surrogate objective function:

$$\tilde{\Gamma}^s(\omega, c_j, \mathcal{P}) = \min_{x_i \in \mathcal{X}} \left( \frac{((\phi(x_{i^*(c_j)}, c_j) - \phi(x_i, c_j))^\top \theta)^2}{\|\phi(x_{i^*(c_j)}, c_j) - \phi(x_i, c_j)\|^2_{\Lambda(\omega)^{-1}}} \mathbb{I}(x_i \in \mathcal{D}_1(c_j) \cup \mathcal{M}_1(c_j)) \right.$$

$$\left. + \frac{(b - \beta^\top \phi(x_i, c_j))^2}{\|\phi(x_i, c_j)\|^2_{\Lambda(\omega)^{-1}}} \mathbb{I}(x_i \in \{x_{i^*(c_j)}\} \cup \mathcal{D}_2(c_j) \cup \mathcal{M}_2(c_j)) \right). \quad (67)$$

Using this surrogate function, we can show that:

$$\mathcal{H}^*(\mathcal{P}) \leq \tilde{\mathcal{U}}^*(\mathcal{P}) \leq 2\mathcal{H}^*(\mathcal{P}). \quad (68)$$

The bound for $\mathcal{U}^*(\mathcal{P})$ becomes tight when the objective values of the arms in $\mathcal{D}_3(c_j)$ are close to that of the best arm, implying that arms in $\mathcal{D}_3(c_j)$ can be easily identified as infeasible rather than suboptimal. In this case, the constant $\gamma$ is close to zero. However, when the constraint performance of arms in $\mathcal{D}_3(c_j)$ is close to the threshold, $\gamma$ may exceed 1, and the second bound $\tilde{\mathcal{U}}^*(\mathcal{P})$ should be used. Since the theoretical analysis of the two bounds is essentially the same, except that the second bound requires constructing two subsets during implementation, without loss of generality, we focus on $\mathcal{U}^*(\mathcal{P})$ in the main paper for notational simplicity.

### A.8 PROOF OF THEOREM 2

*Proof.* Consider the following primal optimization problem in (13):

$$\max_{\omega \in \Omega} \min_{c_j \in \mathcal{C}} \Gamma^s(\omega, c_j, \mathcal{P}), \quad (69)$$

where

$$\Gamma^s(\omega, c_j, \mathcal{P}) = \min_{x_i \in \mathcal{X}} \left( \frac{((\phi(x_{i^*(c_j)}, c_j) - \phi(x_i, c_j))^\top \theta)^2}{\|\phi(x_{i^*(c_j)}, c_j) - \phi(x_i, c_j)\|^2_{\Lambda(\omega)^{-1}}} \mathbb{I}(x_i \in \mathcal{D}_1(c_j)) \right.$$

$$\left. + \frac{(b - \beta^\top \phi(x_i, c_j))^2}{\|\phi(x_i, c_j)\|^2_{\Lambda(\omega)^{-1}}} \mathbb{I}(x_i \in \{x_{i^*(c_j)}\} \cup \mathcal{D}_2(c_j) \cup \mathcal{D}_3(c_j)) \right). \quad (70)$$

This problem is equivalent to:

$$\min_{\omega \in \Omega} \max_{c_j \in \mathcal{C}, x_i \in \mathcal{X}} \left( \frac{\|\phi(x_{i^*(c_j)}, c_j) - \phi(x_i, c_j)\|^2_{\Lambda(\omega)^{-1}}}{((\phi(x_{i^*(c_j)}, c_j) - \phi(x_i, c_j))^\top \theta)^2} \mathbb{I}(x_i \in \mathcal{D}_1(c_j)) \right.$$
$$\left. + \frac{\|\phi(x_i, c_j)\|^2_{\Lambda(\omega)^{-1}}}{(b - \beta^\top \phi(x_i, c_j))^2} \mathbb{I}(x_i \in \{x_{i^*(c_j)}\} \cup \mathcal{D}_2(c_j) \cup \mathcal{D}_3(c_j)) \right). \tag{71}$$

By introducing an auxiliary variable $\xi$, we can reformulate the problem as:

$$\min_{\xi, \omega} \xi$$
$$\text{s.t.} \quad \frac{\|\phi(x_{i^*(c_j)}, c_j) - \phi(x_i, c_j)\|^2_{\Lambda(\omega)^{-1}}}{((\phi(x_{i^*(c_j)}, c_j) - \phi(x_i, c_j))^\top \theta)^2} \leq \xi, \forall c_j \in \mathcal{C}, x_i \in \mathcal{D}_1(c_j)$$
$$\frac{\|\phi(x_i, c_j)\|^2_{\Lambda(\omega)^{-1}}}{(b - \beta^\top \phi(x_i, c_j))^2} \leq \xi, \forall c_j \in \mathcal{C}, x_i \in \{x_{i^*(c_j)}\} \cup \mathcal{D}_2(c_j) \cup \mathcal{D}_3(c_j) \tag{72}$$
$$\sum_{h \in [D]} \omega_h = 1$$
$$\omega_h \geq 0, \forall h \in [D]$$

Since we only sample from $D$ design points, the corresponding design matrix $\Phi \in \mathbb{R}^{D \times D}$ is invertible. Then, we have that

$$\Lambda(\omega)^{-1} = \left( \sum_{h \in [D]} \omega_h \frac{\phi(z_h)\phi(z_h)^\top}{2\sigma_h^2} \right)^{-1} = (\Phi^T \Sigma^{-1} \Phi)^{-1} = \Phi^{-1} \Sigma (\Phi^T)^{-1}, \tag{73}$$

where $\Sigma$ is a diagonal matrix with elements $\{2\sigma_h^2 / \omega_h\}_{h \in [D]}$.

Now, for each covariate $c_j \in \mathcal{C}$ and each arm $x_i \in \mathcal{D}_1(c_j)$, we have

$$\frac{\|\phi(x_{i^*(c_j)}, c_j) - \phi(x_i, c_j)\|^2_{\Lambda(\omega)^{-1}}}{((\phi(x_{i^*(c_j)}, c_j) - \phi(x_i, c_j))^\top \theta)^2}$$
$$= \frac{(\phi(x_{i^*(c_j)}, c_j) - \phi(x_i, c_j))^\top \Lambda(\omega)^{-1} (\phi(x_{i^*(c_j)}, c_j) - \phi(x_i, c_j))}{((\phi(x_{i^*(c_j)}, c_j) - \phi(x_i, c_j))^\top \theta)^2}$$
$$= \frac{(\phi(x_{i^*(c_j)}, c_j) - \phi(x_i, c_j))^\top \Phi^{-1} \Sigma (\Phi^T)^{-1} (\phi(x_{i^*(c_j)}, c_j) - \phi(x_i, c_j))}{((\phi(x_{i^*(c_j)}, c_j) - \phi(x_i, c_j))^\top \theta)^2} \tag{74}$$
$$= 2 \sum_{h \in [D]} \frac{\sigma_h^2 [(\Phi^T)^{-1}(\phi(x_{i^*(c_j)}, c_j) - \phi(x_i, c_j))]^2_h}{\omega_h ((\phi(x_{i^*(c_j)}, c_j) - \phi(x_i, c_j))^\top \theta)^2}$$
$$= 2 \sum_{h \in [D]} \frac{\chi_h(x_i, c_j)}{\omega_h},$$

where we define

$$\chi_h(x_i, c_j) = \frac{\sigma_h^2 [(\Phi^T)^{-1}(\phi(x_{i^*(c_j)}, c_j) - \phi(x_i, c_j))]^2_h}{((\phi(x_{i^*(c_j)}, c_j) - \phi(x_i, c_j))^\top \theta)^2}, \tag{75}$$

and $[v]_h$ denotes the $h$th element of the vector $v$.

Similarly, for each covariate $c_j \in \mathcal{C}$ and each arm $x_i \in \{x_{i^*(c_j)}\} \cup \mathcal{D}_2(c_j) \cup \mathcal{D}_3(c_j)$, we have

$$\frac{\|\phi(x_i, c_j)\|^2_{\Lambda(\omega)^{-1}}}{(b - \beta^\top \phi(x_i, c_j))^2}$$

$$= \frac{\phi(x_i, c_j)^\top \Lambda(\omega)^{-1} \phi(x_i, c_j)}{(b - \beta^\top \phi(x_i, c_j))^2}$$

$$= \frac{\phi(x_i, c_j)^\top \Phi^{-1} \Sigma (\Phi^T)^{-1} \phi(x_i, c_j)}{(b - \beta^\top \phi(x_i, c_j))^2} \tag{76}$$

$$= 2 \sum_{h \in [D]} \frac{\sigma_h^2 [(\Phi^T)^{-1} \phi(x_i, c_j)]_h^2}{\omega_h (b - \beta^\top \phi(x_i, c_j))^2}$$

$$= 2 \sum_{h \in [D]} \frac{\chi_h(x_i, c_j)}{\omega_h},$$

where we define

$$\chi_h(x_i, c_j) = \frac{\sigma_h^2 [(\Phi^T)^{-1} \phi(x_i, c_j)]_h^2}{(b - \beta^\top \phi(x_i, c_j))^2}. \tag{77}$$

Hence, the optimization problem becomes:

$$\min_{\omega, \xi} \xi$$

$$\text{s.t.} \sum_{h \in [D]} \frac{\chi_h(x_i, c_j)}{\omega_h} \le \xi, \forall c_j \in \mathcal{C}, x_i \in \mathcal{X} \quad (\lambda_{ij})$$

$$\sum_{h \in [D]} \omega_h = 1, \quad (\nu) \tag{78}$$

$$\omega_h \ge 0, \forall h \in [D]$$

The corresponding Lagrangian function is:

$$L(\xi, \omega, \lambda, \nu) = \xi + \sum_{j \in [M], i \in [K]} \lambda_{ij} \left( \sum_{h \in [D]} \frac{\chi_h(x_i, c_j)}{\omega_h} - \xi \right) + \nu \left( \sum_{h \in [D]} \omega_h - 1 \right). \tag{79}$$

Let $(\xi^*, \omega^*, \lambda^*, \nu^*)$ denote the optimal primal-dual solution. The KKT conditions for this optimization problem are:

$$\sum_{j \in [M], i \in [K]} \lambda_{ij}^* = 1$$

$$- \sum_{j \in [M], i \in [K]} \lambda_{ij}^* \frac{\chi_h(x_i, c_j)}{(\omega_h^*)^2} + \nu^* = 0$$

$$\lambda_{ij}^* \left( \sum_{h \in [D]} \frac{\chi_h(x_i, c_j)}{\omega_h^*} - \xi^* \right) = 0, \forall j \in [M], i \in [K]$$

$$\lambda_{ij}^* \ge 0, \forall j \in [M], i \in [K] \tag{80}$$

$$\sum_{h \in [D]} \frac{\chi_h(x_i, c_j)}{\omega_h^*} \le \xi, \forall c_j \in \mathcal{C}, x_i \in \mathcal{X}$$

$$\sum_{h \in [D]} \omega_h^* = 1$$

$$\omega_h^* \ge 0, \forall h \in [D].$$

From the second and sixth equations, we deduce the optimal form of $\omega_h^*$. Solving the second equation, we obtain:

$$\omega_h^* = \sqrt{\frac{\sum_{j \in [M], i \in [K]} \lambda_{ij}^* \chi_h(x_i, c_j)}{\nu^*}}, \tag{81}$$

Using the sixth equation, we normalize the solution:

$$\omega_h^* = \frac{\sqrt{\sum_{j\in[M],i\in[K]} \lambda_{ij}^* \chi_h(x_i,c_j)}}{\sum_{l\in[D]} \sqrt{\sum_{j\in[M],i\in[K]} \lambda_{ij}^* \chi_l(x_i,c_j)}}. \tag{82}$$

We now derive the Lagrange dual function.

$$
\begin{aligned}
g(\lambda,\nu) &= \inf_{\xi,\omega} L(\xi,\omega,\lambda,\nu) \\
&= \inf_{\xi,\omega}(1 - \sum_{j\in[M],i\in[K]} \lambda_{ij})\xi + \sum_{j\in[M],i\in[K]} \lambda_{ij} \sum_{h\in[D]} \frac{\chi_h(x_i,c_j)}{\omega_h} + \nu(\sum_{h\in[D]} \omega_h - 1) \\
&= \begin{cases} \inf_\omega \sum_{j\in[M],i\in[K]} \lambda_{ij} \sum_{h\in[D]} \frac{\chi_h(x_i,c_j)}{\omega_h} + \nu(\sum_{h\in[D]} \omega_h - 1) & \text{if } \sum_{j\in[M],i\in[K]} \lambda_{ij} = 1, \lambda_{ij} \ge 0 \\ -\infty & \text{o.w.} \end{cases} \\
&= \begin{cases} 2\sqrt{\nu} \sum_{h\in[D]} \sqrt{\sum_{j\in[M],i\in[K]} \lambda_{ij}\chi_h(x_i,c_j)} & \text{if } \sum_{j\in[M],i\in[K]} \lambda_{ij} = 1, \lambda_{ij} \ge 0 \\ -\infty & \text{o.w.} \end{cases}
\end{aligned}
\tag{83}
$$

By optimizing the variable $\nu$, we can obtain that the dual optimization problem is

$$
\begin{aligned}
\max_\lambda \quad & \left( \sum_{h\in[D]} \sqrt{\sum_{j\in[M],i\in[K]} \lambda_{ij}\chi_h(x_i,c_j)} \right)^2 \\
\text{s.t.} \quad & \sum_{j\in[M],i\in[K]} \lambda_{ij} = 1 \\
& \lambda_{ij} \ge 0, \forall i \in [K], j \in [M].
\end{aligned}
\tag{84}
$$

$\square$

### A.9 PROOF OF LEMMA 2

*Proof.* The convexity of the primal optimization problem (13) can be established under more general distributional assumptions.

As shown in the proof of Theorem 1, the optimization problem (13) can be equivalently derived from the following formulation:

$$
\begin{aligned}
\max_{\omega\in\Omega} \min_{c_j\in\mathcal{C}} \min \Big( & \inf_{\tilde{\mathcal{P}}\in\mathcal{O}(x_{i^*(c_j)},c_j)} \mathcal{H}(\omega,\mathcal{P},\tilde{\mathcal{P}})^{-1}, \min_{x_i\in\mathcal{D}_1(c_j)} \inf_{\tilde{\mathcal{P}}\in\mathcal{O}_1(x_i,c_j)} \mathcal{H}(\omega,\mathcal{P},\tilde{\mathcal{P}})^{-1} \\
& \min_{x_i\in\mathcal{D}_2(c_j)\cup\mathcal{D}_3(c_j)} \inf_{\tilde{\mathcal{P}}\in\mathcal{O}_2(x_i,c_j)} \mathcal{H}(\omega,\mathcal{P},\tilde{\mathcal{P}})^{-1} \Big),
\end{aligned}
\tag{85}
$$

where the sets and functionals are defined as follows:

$$
\begin{aligned}
\mathcal{O}(x_{i^*(c_j,\mathcal{P})},c_j) &= \left\{ \tilde{\mathcal{P}} \in \mathcal{S} : \tilde{\beta}^\top \phi(x_{i^*(c_j,\mathcal{P})},c_j) > b \right\}, \\
\mathcal{O}_1(x_i,c_j) &= \left\{ \tilde{\mathcal{P}} \in \mathcal{S} : \tilde{\theta}^\top (\phi(x_i,c_j) - \phi(x_{i^*(c_j,\mathcal{P})},c_j)) > 0 \right\}, \\
\mathcal{O}_2(x_i,c_j) &= \left\{ \tilde{\mathcal{P}} \in \mathcal{S} : \tilde{\beta}^\top \phi(x_i,c_j) \le b \right\}, \\
\mathcal{H}(\omega,\mathcal{P},\tilde{\mathcal{P}})^{-1} &= \sum_{h\in[D]} \omega_h \Big( d(f(z_h),\tilde{f}(z_h)) + d(g(z_h),\tilde{g}(z_h)) \Big).
\end{aligned}
\tag{86}
$$

Note that $\mathcal{H}(\omega, \mathcal{P}, \tilde{\mathcal{P}})^{-1}$ is a convex function of $\tilde{\mathcal{P}}$, due to the convexity of the KL divergence (see Wang et al. (2021)). Therefore, the following problems are convex programs for fixed $\omega \in \Omega$:

$$
\begin{aligned}
\mathcal{L}(x_{i^*(c_j)}, \omega, \mathcal{P}) &= \inf_{\tilde{\mathcal{P}} \in \mathcal{O}(x_{i^*(c_j)}, c_j)} \mathcal{H}(\omega, \mathcal{P}, \tilde{\mathcal{P}})^{-1} \\
\mathcal{L}_1(x_i, \omega, \mathcal{P}) &= \inf_{\tilde{\mathcal{P}} \in \mathcal{O}_1(x_i, c_j)} \mathcal{H}(\omega, \mathcal{P}, \tilde{\mathcal{P}})^{-1} \\
\mathcal{L}_2(x_i, \omega, \mathcal{P}) &= \inf_{\tilde{\mathcal{P}} \in \mathcal{O}_2(x_i, c_j)} \mathcal{H}(\omega, \mathcal{P}, \tilde{\mathcal{P}})^{-1}
\end{aligned}
\tag{87}
$$

The resulting functions $\mathcal{L}(x_{i^*(c_j)}, \omega, \mathcal{P})$, $\mathcal{L}_1(x_i, \omega, \mathcal{P})$, and $\mathcal{L}_2(x_i, \omega, \mathcal{P})$ are concave in $\omega$, as each is defined as the point-wise infimum of functions that are concave in $\omega$. Consequently, the overall objective in (85) is concave in $\omega$, and the problem is a convex maximization problem. Moreover, it is straightforward to verify that this problem is strictly feasible. Hence, by standard results in convex optimization, strong duality holds.

By (78), this optimization problem is equivalent to

$$
\min_\omega f(\omega) = \max_{c_j \in \mathcal{C}, x_i \in \mathcal{X}} \sum_{h \in [D]} \frac{\chi_h(x_i, c_j)}{\omega_h}
$$

$$
\text{s.t.} \sum_{h \in [D]} \omega_h = 1,
\tag{88}
$$

$$
\omega_h \geq 0, \forall h \in [D]
$$

Assume that $\omega$ and $\omega'$ are two optimal solutions such that $f(\omega) = f(\omega^*) = \xi^*$. For any $\lambda \in (0, 1)$, define $\omega'' = \lambda\omega + (1-\lambda)\omega'$. Then, by the strong convexity of $1/\omega_h$ on the interval $(0, \infty)$, we have

$$
\frac{1}{\omega_j''} \leq \lambda \frac{1}{\omega_j} + (1-\lambda) \frac{1}{\omega_j'}.
\tag{89}
$$

Since $\chi_h(x_i, c_j) > 0$ for all $h \in [D]$, $c_j \in \mathcal{C}$, and $x_i \in \mathcal{X}$, it follows that

$$
\sum_{h \in [D]} \frac{\chi_h(x_i, c_j)}{\omega_j''} \leq \lambda \sum_{h \in [D]} \frac{\chi_h(x_i, c_j)}{\omega_j} + (1-\lambda) \frac{\chi_h(x_i, c_j)}{\omega_j'} = \xi^*.
\tag{90}
$$

If $\omega \neq \omega'$, then the inequality holds strictly, contradicting the assumption that both $\omega$ and $\omega'$ are optimal solutions. Hence, the optimal solution is unique.

$\square$

### A.10 PROOF OF LEMMA 3

*Proof.* Consider the dual optimization problem stated in Theorem 2:

$$
\min_\lambda \mathcal{Q}(\lambda, \mathcal{P}) = -\sum_{h \in [D]} \sqrt{\sum_{i \in [K], j \in [M]} \lambda_{ij} \chi_h(x_i, c_j)}
$$

$$
\text{s.t.} \sum_{i \in [K], j \in [M]} \lambda_{ij} = 1, \quad (\phi)
\tag{91}
$$

$$
\lambda_{ij} \geq 0, \quad \forall i \in [K], j \in [M]. \quad (v_{ij})
$$

For any feasible solution $\lambda$, the set of all feasible directions at $\lambda$ is defined by:

$$
\mathcal{F}(\lambda) = \left\{ d \in \mathbb{R}^{KM} : \sum_{j \in [M], i \in [K]} d_{ij} = 0, \, d_{ij} \geq 0, \text{if } \lambda_{ij} = 0 \right\}.
\tag{92}
$$

The Lagrangian function for this problem is:

$$
L(\lambda, \phi, v) = -\sum_{h \in [D]} \sqrt{\sum_{i \in [K], j \in [M]} \lambda_{ij} \chi_h(x_i, c_j)} + \phi \left( \sum_{i \in [K], j \in [M]} \lambda_{ij} - 1 \right) - \sum_{i \in [K], j \in [M]} v_{ij} \lambda_{ij}.
\tag{93}
$$

Let $(\lambda^*, \phi^*, v^*)$ denote an optimal primal-dual solution. The KKT conditions of this optimization problem are given by:

$$-\frac{1}{2} \sum_{h \in [D]} \frac{\chi_h(x_i, c_j)}{\sqrt{\sum_{i \in [K], j \in [M]} \lambda_{ij}^* \chi_h(x_i, c_j)}} + \phi^* - v_{ij}^* = 0$$

$$v_{ij}^* \lambda_{ij}^* = 0$$

$$\lambda_{ij}^* \geq 0 \qquad (94)$$

$$\sum_{i \in [K], j \in [M]} \lambda_{ij}^* = 1$$

$$v_{ij}^* \geq 0$$

From these KKT conditions, we observe that a feasible solution $\lambda^*$ is a stationary point if and only if there exists a $\phi^*$ such that if $\lambda_{ij}^* = 0$, then

$$\phi^* \geq \frac{1}{2} \sum_{h \in [D]} \frac{\chi_h(x_i, c_j)}{\sqrt{\sum_{i \in [K], j \in [M]} \lambda_{ij}^* \chi_h(x_i, c_j)}} \qquad (95)$$

and if $\lambda_{ij}^* > 0$, then

$$\phi^* = \frac{1}{2} \sum_{h \in [D]} \frac{\chi_h(x_i, c_j)}{\sqrt{\sum_{i \in [K], j \in [M]} \lambda_{ij}^* \chi_h(x_i, c_j)}}. \qquad (96)$$

This implies that a feasible solution $\lambda$ is a stationary point of problem (14) if and only if:

$$-\frac{1}{2} \sum_{h \in [D]} \frac{\chi_h(x_i, c_j)}{\sqrt{\sum_{i \in [K], j \in [M]} \lambda_{ij}^* \chi_h(x_i, c_j)}} \geq -\frac{1}{2} \sum_{h \in [D]} \frac{\chi_h(x_{i'}, c_{j'})}{\sqrt{\sum_{i \in [K], j \in [M]} \lambda_{ij}^* \chi_h(x_i, c_j)}}, \qquad (97)$$

for any $(i, j) \in \{(a, b) : a \in [K], b \in [M]\}$ and $(i', j') \in \{(a, b) : a \in [K], b \in [M], \lambda_{ab} > 0\}$. Now, fix a feasible solution $\lambda$ with $\lambda_{mn} > 0$. Define the reduced set:

$$\mathcal{D}^{m,n}(\lambda) = \left\{ e_{ij} - e_{mn} : i \neq m \text{ or } j \neq n \right\} \bigcup \left\{ e_{mn} - e_{ij} : i \neq m \text{ or } j \neq n, \lambda_{ij} > 0 \right\}, \qquad (98)$$

where $e_{ij} \in \mathbb{R}^{KM}$ is obtained by letting $\lambda_{ij}$ equal to one and other elements equal to zero.

According to Proposition 3.4 of Lin et al. (2009), we have:

$$\mathcal{D}^{m,n} \subset \mathcal{F}(\lambda), \quad Conv(\mathcal{D}^{m,n}(\lambda)) = \mathcal{F}(\lambda). \qquad (99)$$

Combining this with the stationary condition (97), we conclude that a feasible solution $\lambda$ is a stationary point of problem (14) if and only if:

$$\nabla \mathcal{Q}(\lambda, \mathcal{P})^\top d \geq 0, \forall d \in \mathcal{D}^{m,n}(\lambda). \qquad (100)$$

$\square$

A.11 PROOF OF THEOREM 3

The proof of Theorem 3 relies on several auxiliary lemmas. Lemma 4 establishes the necessary continuity arguments. Lemma 5 proves the $\delta$-PAC property of the proposed algorithm. Lemmas 6 and 7 present known results from the existing literature. Lemma 8 establishes the convergence of the gradient descent procedures in Algorithm 2. Finally, we derive upper bounds—both almost surely and in expectation—for the stopping time $\tau$.

**Lemma 4.** Let $\mathcal{U}(\omega, \mathcal{P})^{-1} = \min_{c_j \in \mathcal{C}} \Gamma^s(\omega, c_j, \mathcal{P})$ denote the objective function of problem (14). Then, $\mathcal{U}(\omega, \mathcal{P})^{-1}$ is continuous function with respect to both $\omega$ and $\mathcal{P}$. Moreover, the optimal sampling ratio $\omega^*$ satisfies $\omega_h^* > 0$ for all $h \in [D]$.

*Proof.* Recall the following notation from the proof of Lemma 2:

$$\mathcal{O}(x_{i^*(c_j,\mathcal{P})}, c_j) = \left\{ \tilde{\mathcal{P}} \in \mathcal{S} : \tilde{\beta}^\top \phi(x_{i^*(c_j,\mathcal{P})}, c_j) > b \right\},$$

$$\mathcal{O}_1(x_i, c_j) = \left\{ \tilde{\mathcal{P}} \in \mathcal{S} : \tilde{\theta}^\top \left( \phi(x_i, c_j) - \phi(x_{i^*(c_j,\mathcal{P})}, c_j) \right) > 0 \right\},$$

$$\mathcal{O}_2(x_i, c_j) = \left\{ \tilde{\mathcal{P}} \in \mathcal{S} : \tilde{\beta}^\top \phi(x_i, c_j) \leq b \right\},$$

$$\mathcal{H}(\omega, \mathcal{P}, \tilde{\mathcal{P}})^{-1} = \sum_{h \in [D]} \omega_h \left( d(f(z_h), \tilde{f}(z_h)) + d(g(z_h), \tilde{g}(z_h)) \right).$$

(101)

Define the alternative set of problem instances for a context $c_j$ and problem instance $\mathcal{P}$ as:

$$\mathcal{A}'(c_j, \mathcal{P}) = \mathcal{O}(x_{i^*(c_j,\mathcal{P})}, c_j) \bigcup \left( \bigcup_{x_i \in \mathcal{D}_1(c_j)} \mathcal{O}_1(x_i, c_j) \right) \bigcup \left( \bigcup_{x_i \in \mathcal{D}_2(c_j) \cup \mathcal{D}_3(c_j)} \mathcal{O}_2(x_i, c_j) \right),$$

(102)

and $\mathcal{A}'(\mathcal{P}) = \bigcup_{c_j \in \mathcal{C}} \mathcal{A}'(c_j, \mathcal{P})$.

From Lemma 2, for a given context $c_j \in \mathcal{C}$, we have:

$$\mathcal{U}(\omega, \mathcal{P})^{-1} = \min_{c_j \in \mathcal{C}} \Gamma^s(\omega, c_j, \mathcal{P})$$

$$= \min_{c_j \in \mathcal{C}} \inf_{\tilde{\mathcal{P}} \in \mathcal{A}'(c_j, \mathcal{P})} \mathcal{H}(\omega, \mathcal{P}, \tilde{\mathcal{P}})^{-1}$$

$$= \min_{c_j \in \mathcal{C}} \inf_{\tilde{\mathcal{P}} \in \mathcal{A}'(c_j, \mathcal{P})} \sum_{h \in [D]} \omega_h \left( d(f(z_h), \tilde{f}(z_h)) + d(g(z_h), \tilde{g}(z_h)) \right)$$

$$= \min_{c_j \in \mathcal{C}} \inf_{\tilde{\mathcal{P}} \in \mathcal{A}'(c_j, \mathcal{P})} \sum_{h \in [D]} \omega_h \left( \frac{(\theta - \tilde{\theta})^\top \phi(z_h) \phi(z_h)^\top (\theta - \tilde{\theta})}{2\sigma_h^2} + \frac{(\beta - \tilde{\beta})^\top \phi(z_h) \phi(z_h)^\top (\beta - \tilde{\beta})}{2\sigma_h^2} \right)$$

$$= \min_{c_j \in \mathcal{C}} \inf_{\tilde{\mathcal{P}} \in \mathcal{A}'(c_j, \mathcal{P})} (\theta - \tilde{\theta})^\top \Lambda(\omega)(\theta - \tilde{\theta}) + (\beta - \tilde{\beta})^\top \Lambda(\omega)(\beta - \tilde{\beta}).$$

(103)

Now, consider a sequence $(\hat{\mathcal{P}}(t), \omega(t))$ such that: $\lim_{t \to \infty} (\hat{\mathcal{P}}(t), \omega(t)) = (\mathcal{P}, \omega)$. By definition of $x_{i^*(c_j,\mathcal{P})}, \mathcal{D}_1(c_j), \mathcal{D}_2(c_j)$ and $\mathcal{D}_3(c_j)$, we obtain $\lim_{t \to \infty} \mathcal{A}'(c_j, \hat{\mathcal{P}}(t)) = \mathcal{A}(c_j, \mathcal{P})$.

Therefore, for any $\epsilon > 0$, there exists $t_0 > 0$ such that for all $t \geq t_0$, we have

$$\|(\hat{\mathcal{P}}(t), \omega(t)) - (\mathcal{P}, \omega)\|_\infty \leq \epsilon, \quad \mathcal{A}'(c_j, \hat{\mathcal{P}}(t)) = \mathcal{A}(c_j, \mathcal{P}) \tag{104}$$

Since $\mathcal{H}(\omega, \mathcal{P}, \tilde{\mathcal{P}})^{-1}$ is a polynomial in its arguments, it is continuous with respect to $\omega, \mathcal{P}$. Thus, there exists $t_1 > 0$ such that for any $t \geq t_1$:

$$\left| \mathcal{H}(\omega_t, \hat{\mathcal{P}}(t), \tilde{\mathcal{P}})^{-1} - \mathcal{H}(\omega, \mathcal{P}, \tilde{\mathcal{P}})^{-1} \right| \leq \epsilon, \tag{105}$$

Combining both observations, there exists $t_2 > \max(t_0, t_1)$, such that for any $t > t_2$ we have

$$\left| \mathcal{U}(\omega_t, \hat{\mathcal{P}}(t))^{-1} - \mathcal{U}(\omega, \mathcal{P})^{-1} \right| = \left| \min_{c_j \in \mathcal{C}} \inf_{\tilde{\mathcal{P}} \in \mathcal{A}'(c_j, \mathcal{P})} \mathcal{H}(\omega_t, \hat{\mathcal{P}}(t), \tilde{\mathcal{P}})^{-1} - \min_{c_j \in \mathcal{C}} \inf_{\tilde{\mathcal{P}} \in \mathcal{A}'(c_j, \mathcal{P})} \mathcal{H}(\omega, \mathcal{P}, \tilde{\mathcal{P}})^{-1} \right|$$

$$\leq \epsilon, \tag{106}$$

which establishes the continuity of $\mathcal{U}(\omega, \mathcal{P})^{-1}$.

Now, let $\omega^* \in \Omega$ denote the optimal solution of problem (13). Suppose, for contradiction, that there exists $h \in [D]$ such that $\omega_h^* = 0$. Then, one can construct an alternative problem instance $\tilde{\mathcal{P}} \in \mathcal{A}'(\mathcal{P})$ such that $\min_{c_j \in \mathcal{C}} \Gamma(\omega_h^*, c_j, \mathcal{P}) = 0$. This contradicts the optimality of $\omega^*$ because we can always choose a feasible uniform sampling rule $\tilde{\omega} \in \Omega$ with $\tilde{\omega}_h = 1/D, \forall h \in [D]$, which yields $\min_{c_j \in \mathcal{C}} \Gamma(\omega_h^*, c_j, \mathcal{P}) > 0$. Hence, it must hold that $\omega_h^* > 0$ for all $h \in [D]$. □

**Lemma 5.** *The duality-based decomposition algorithm is $\delta$-PAC.*

*Proof.* The stopping rule of the duality-based decomposition algorithm is

$$\tau = \inf\left\{t \in \mathbb{N} : t\mathcal{U}(\hat{\mathcal{P}}(t), \omega(t))^{-1} > \rho(t, \delta)\right\}, \tag{107}$$

where $\mathcal{U}(\hat{\mathcal{P}}(t), \omega(t))^{-1} = \min_{c_j \in \mathcal{C}} \Gamma^s(\omega(t), c_j, \hat{\mathcal{P}}(t))$. To establish the $\delta$-PAC property of the duality-based decomposition algorithm, we must show that

$$\mathbb{P}\left(\tau < \infty, \exists c_j \in \mathcal{C}, x_{\hat{i}(c_j;\tau)} \neq x_{i^*(c_j)}\right) \leq \delta. \tag{108}$$

We begin by noting that

$$\mathbb{P}\left(\tau < \infty, \exists c_j \in \mathcal{C}, x_{\hat{i}(c_j;\tau)} \neq x_{i^*(c_j)}\right)$$

$$\leq \mathbb{P}\left(\exists t \in \mathbb{N}, \exists c_j \in \mathcal{C}, x_{\hat{i}(c_j;\tau)} \neq x_{i^*(c_j)}, t\mathcal{U}(\hat{\mathcal{P}}(t), \omega_t)^{-1} \geq \rho(t, \delta)\right)$$

$$= \mathbb{P}\left(\exists t \in \mathbb{N}, \exists c_j \in \mathcal{C}, x_{\hat{i}(c_j;\tau)} \neq x_{i^*(c_j)}, \inf_{\tilde{\mathcal{P}} \in \mathcal{A}'(\hat{\mathcal{P}}(t))} t\mathcal{H}(\omega_t, \hat{\mathcal{P}}(t), \tilde{\mathcal{P}})^{-1} \geq \rho(t, \delta)\right)$$

$$\leq \mathbb{P}\left(\exists t \in \mathbb{N}, t\mathcal{H}(\omega_t, \hat{\mathcal{P}}(t), \mathcal{P})^{-1} \geq \rho(t, \delta)\right)$$

$$= \mathbb{P}\left(\exists t \in \mathbb{N}, \sum_{h \in [D]} N_h(d(\bar{F}(z_h; t), f(z_h)) + d(\bar{G}(z_h; t), g(z_h))) \geq \rho(t, \delta)\right)$$

$$\leq \sum_{t=1}^{\infty} \mathbb{P}\left(\left[\sum_{h \in [D]} N_h d(\bar{F}(z_h; t), f(z_h)) > \frac{1}{2}\rho(t, \delta)\right] \bigcup \left[\sum_{h \in [D]} N_h d(\bar{G}(z_h; t), g(z_h)) > \frac{1}{2}\rho(t, \delta)\right]\right)$$

$$\leq \sum_{t=1}^{\infty} \mathbb{P}\left(\left[\sum_{h \in [D]} N_h d(\bar{F}(z_h; t), f(z_h)) > \frac{1}{2}\rho(t, \delta)\right]\right) + \sum_{t=1}^{\infty} \mathbb{P}\left(\left[\sum_{h \in [D]} N_h d(\bar{G}(z_h; t), g(z_h)) > \frac{1}{2}\rho(t, \delta)\right]\right) \tag{109}$$

According to Proposition 12 of Garivier & Kaufmann (2016), we have

$$\mathbb{P}\left(\left[\sum_{h \in [D]} N_h d(\bar{F}(z_h; t), f(z_h)) > \frac{1}{2}\rho(t, \delta)\right]\right) \leq e^{-\frac{1}{2}\rho(t,\delta)} \left(\frac{\rho(t, \delta)^2 \log t}{4D}\right)^D e^{D+1}. \tag{110}$$

Similarly, an identical bound holds for the second term

$$\mathbb{P}\left(\left[\sum_{h \in [D]} N_h d(\bar{G}(z_h; t), g(z_h)) > \frac{1}{2}\rho(t, \delta)\right]\right). \tag{111}$$

Thus, if we choose $\rho(t, \delta) = \log(Ct^\alpha/\delta)$, and let $C$ be a constant such that

$$\sum_{t=1}^{\infty} e^{-\frac{1}{2}\rho(t,\delta)} \left(\frac{\rho(t, \delta)^2 \log t}{4D}\right)^D e^{D+1} \leq \frac{\delta}{2}, \tag{112}$$

then both infinite series are bounded above by $\delta/2$, leading to the final result:

$$\mathbb{P}\left(\tau < \infty, \exists c_j \in \mathcal{C}, x_{\hat{i}(c_j;\tau)} \neq x_{i^*(c_j)}\right) \leq \delta. \tag{113}$$

$\square$

The convergence analysis of the duality-based decomposition algorithm relies on a line search procedure to determine the step size. For completeness, we include the canonical line search algorithm along with its associated theoretical results.

---

**Algorithm 3:** Line Search Algorithm

---

1 **Input:** Descent direction $d$, maximum feasible step size $s^{max}$, the current feasible solution $\lambda$, problem instance $\mathcal{P}$, parameter $\alpha$ and $\nu \in (0,1)$.

2 Set $s = s^{max}$

3 **while** $\mathcal{Q}(\lambda + sd, \mathcal{P}) > \mathcal{Q}(\lambda, \mathcal{P}) + \alpha s \nabla \mathcal{Q}(\lambda, \mathcal{P})^\top d$ **do**

4 $\quad \lfloor \quad s \leftarrow \nu s$

5 **return** the step size $s$.

---

**Lemma 6** (Proposition 4.1 in Lin et al. (2009)). *Define a subsequence $\mathcal{T} \subset \{1, 2 \ldots\}$ such that the line search algorithm is invoked at time steps $t \in \mathcal{T}$. Let $\{\lambda(t)\}_{t \in \mathcal{T}}$ denote the corresponding sequence of solutions, and let $\{d(t)\}_{t \in \mathcal{T}}$ denote the associated descent directions. Then, the line search algorithm terminates in a finite number of iterations, producing a step size $s(t)$ that satisfies*

$$\mathcal{Q}(\lambda(t-1) + s(t)d(t), \hat{\mathcal{P}}(t)) \leq \mathcal{Q}(\lambda(t), \hat{\mathcal{P}}(t)) + \alpha s(t) \nabla \mathcal{Q}(\lambda(t-1), \hat{\mathcal{P}}(t))^\top d(t). \tag{114}$$

*Furthermore, suppose that $\lim_{t \to \infty} \lambda(t) = \bar{\lambda}$, and*

$$\lim_{t \to \infty} \mathcal{Q}(\lambda(t-1), \mathcal{P}) - \mathcal{Q}(\lambda(t-1) + s(t)d(t), \mathcal{P}) = 0. \tag{115}$$

*Then, it follows that*

$$\lim_{t \to \infty} s^{max} \nabla \mathcal{Q}(\lambda(t-1), \mathcal{P})^\top d(t) = 0. \tag{116}$$

**Lemma 7** (Lemma 17 in Garivier & Kaufmann (2016)). *Consider the following sampling rule*

$$z_{h(t+1)} = \begin{cases} \arg\min_{z_h \in \mathcal{B}_t} N_h(t) & \text{if } \mathcal{B}_t \neq \emptyset \\ \arg\min_{z_h \in \mathcal{Z}} N_h(t) - t\gamma_h(\hat{\mathcal{P}}(t)) & \text{otherwise} \end{cases}, \tag{117}$$

*where $\mathcal{B}_t = \{z_h \in \mathcal{Z} : N_h(t) < \sqrt{t} - D/2\}$. Then, for every design point $z_h \in \mathcal{Z}$, we have $N_h(t) \geq (\sqrt{t} - D/2)_+ - 1$. Furthermore, for any $\epsilon > 0$ and $t_0 > 0$ such that*

$$\sup_{t \geq t_0} \max_{h \in [D]} \left| \gamma_h(\hat{\mathcal{P}}(t)) - \omega_h^*(\mathcal{P}) \right| \leq \epsilon, \tag{118}$$

*there exists $t_1 > 0$ such that*

$$\sup_{t \geq t_1} \max_{h \in [D]} \left| \frac{N_h(t)}{t} - \omega_h^*(\mathcal{P}) \right| \leq 3(D-1)\epsilon. \tag{119}$$

The following lemma establishes the convergence of the gradient descent procedure in Algorithm 2. The analysis follows the proof of Proposition 6.1 in Lin et al. (2009) and Theorem 5 in Zhou et al. (2024).

**Lemma 8.** *Let $\{\lambda(t)\}$ be the sequence generated by the duality-based algorithm. Then every limit point of this sequence is a stationary point of the dual optimization problem (14).*

*Proof.* According to Lemma 7, the sampling rule of the duality-based decomposition algorithm guarantees that

$$N_h(t) \geq (\sqrt{t} - D/2)_+ - 1. \tag{120}$$

This lower bound implies that the number of samples allocated to each design point grows unbounded as $t \to \infty$. Consequently, by the strong law of large numbers, the estimators converge almost surely:

$$\hat{\theta}(t) \to \theta, \hat{\beta}(t) \to \beta \text{ and } \hat{\mathcal{P}}(t) \to \mathcal{P} \tag{121}$$

As a result, the estimated best arm $x_{\hat{i}(c_j;t)}$ converges almost surely to the true best arm $x_{i^*(c_j)}$ for all $c_j \in \mathcal{C}$ almost surely. This establishes the consistency of the proposed duality-based decomposition algorithm.

We now establish useful continuity properties of the objective function $\mathcal{Q}(\lambda, \mathcal{P})$ and its gradient $\nabla \mathcal{Q}(\lambda, \mathcal{P})$. Recall that

$$\mathcal{Q}(\lambda, \mathcal{P}) = - \sum_{h \in [D]} \sqrt{\sum_{i \in [K], j \in [M]} \lambda_{ij} \chi_h(x_i, c_j, \mathcal{P})}, \tag{122}$$

and for $i \in [K], j \in [M]$,

$$[\nabla \mathcal{Q}(\lambda, \mathcal{P})]_{ij} = - \sum_{h \in [D]} \frac{\chi_h(x_i, c_j, \mathcal{P})}{2\sqrt{\sum_{i \in [K], j \in [M]} \lambda_{ij} \chi_h(x_i, c_j, \mathcal{P})}}. \tag{123}$$

It is straightforward to verify that $\mathcal{Q}(\lambda, \mathcal{P})$ is continuous in $\lambda$. We now show that it is also continuous in $\mathcal{P}$. Since $\hat{\mathcal{P}}(t)) \to \mathcal{P}$ and by definition of $\chi_h(x_i, c_j)$, for sufficiently large $t$, we have

$$|\chi_h(x_i, c_j, \mathcal{P}) - \chi_h(x_i, c_j, \hat{\mathcal{P}}(t))| \leq L\|\mathcal{P} - \hat{\mathcal{P}}(t)\|_\infty, \tag{124}$$

for some constant $L > 0$. Then,

$$\begin{aligned}
&|\mathcal{Q}(\lambda, \mathcal{P}) - \mathcal{Q}(\lambda, \hat{\mathcal{P}}(t))| \\
&= \left| \sum_{h \in [D]} \sqrt{\sum_{i \in [K], j \in [M]} \lambda_{ij} \chi_h(x_i, c_j, \mathcal{P})} - \sum_{h \in [D]} \sqrt{\sum_{i \in [K], j \in [M]} \lambda_{ij} \chi_h(x_i, c_j, \hat{\mathcal{P}}(t))} \right| \\
&\leq \sum_{h \in [D]} \left| \sqrt{\sum_{i \in [K], j \in [M]} \lambda_{ij} \chi_h(x_i, c_j, \mathcal{P})} - \sqrt{\sum_{i \in [K], j \in [M]} \lambda_{ij} \chi_h(x_i, c_j, \hat{\mathcal{P}}(t))} \right| \\
&\leq \sum_{h \in [D]} \frac{\sum_{i \in [K], j \in [M]} \lambda_{ij} |\chi_h(x_i, c_j, \mathcal{P}) - \chi_h(x_i, c_j, \hat{\mathcal{P}}(t))|}{\sqrt{\sum_{i \in [K], j \in [M]} \lambda_{ij} \chi_h(x_i, c_j, \mathcal{P})} + \sqrt{\sum_{i \in [K], j \in [M]} \lambda_{ij} \chi_h(x_i, c_j, \hat{\mathcal{P}}(t))}} \\
&\leq \sum_{h \in [D]} \sum_{i \in [K], j \in [M]} \frac{|\chi_h(x_i, c_j, \mathcal{P}) - \chi_h(x_i, c_j, \hat{\mathcal{P}}(t)))|}{\sqrt{\chi_h(x_i, c_j, \mathcal{P})} + \sqrt{\chi_h(x_i, c_j, \hat{\mathcal{P}}(t))}} \\
&\leq \frac{DKML}{\sqrt{C_0}} \|\hat{\mathcal{P}}(t) - \mathcal{P}\|_\infty \\
&\triangleq \bar{C}\|\hat{\mathcal{P}}(t) - \mathcal{P}\|_\infty,
\end{aligned} \tag{125}$$

where $C_0 = \min_{i \in [K], j \in [M], h \in [D]} \inf_t \chi_h(x_i, c_j, \hat{\mathcal{P}}(t)) > 0$ is some constant and we define $\bar{C} = DKML/\sqrt{C_0}$.

We next show that $\nabla \mathcal{Q}(\lambda, \hat{\mathcal{P}}(t))$ is continuous in $\lambda$. Following the approach of Theorem 5 in Zhou et al. (2024), it holds that

$$\liminf_{t \to \infty} \sum_{i \in [K], j \in [M]} \lambda_{ij}(t) \chi_h(x_i, c_j, \hat{\mathcal{P}}(t)) > 0, \forall i \in [K], j \in [M], h \in [D]. \tag{126}$$

Let $C_{min} > 0$ be a lower bound for $\sum_{i\in[K],j\in[M]} \lambda_{ij}(t)\chi_h(x_i, c_j, \hat{\mathcal{P}}(t))$ for all $i \in [K], j \in [M], h \in [D]$ for sufficiently large $t$. Then,

$$
\begin{aligned}
&\left|[\nabla\mathcal{Q}(\lambda, \hat{\mathcal{P}}(t))]_{ij} - [\nabla\mathcal{Q}(\lambda', \hat{\mathcal{P}}(t))]_{ij}\right| \\
=&\left|\sum_{h\in[D]} \frac{\chi_h(x_i, c_j, \hat{\mathcal{P}}(t))}{2\sqrt{\sum_{i\in[K],j\in[M]} \lambda_{ij}\chi_h(x_i, c_j, \hat{\mathcal{P}}(t))}} - \sum_{h\in[D]} \frac{\chi_h(x_i, c_j, \hat{\mathcal{P}}(t))}{2\sqrt{\sum_{i\in[K],j\in[M]} \lambda'_{ij}\chi_h(x_i, c_j, \hat{\mathcal{P}}(t))}}\right| \\
\leq&\sum_{h\in[D]} \frac{\chi_h(x_i, c_j, \hat{\mathcal{P}}(t))}{2}\left|\frac{1}{\sqrt{\sum_{i\in[K],j\in[M]} \lambda_{ij}\chi_h(x_i, c_j, \hat{\mathcal{P}}(t))}} - \frac{1}{\sqrt{\sum_{i\in[K],j\in[M]} \lambda'_{ij}\chi_h(x_i, c_j, \hat{\mathcal{P}}(t))}}\right| \\
\leq&\sum_{h\in[D]} \frac{C_1}{4C_{min}^{\frac{3}{2}}}\left|\sum_{i\in[K],j\in[M]} (\lambda'_{ij} - \lambda_{ij})\chi_h(x_i, c_j, \hat{\mathcal{P}}(t))\right| \\
\leq&\frac{DKMC_1^2}{4C_{min}^{\frac{3}{2}}}\|\lambda' - \lambda\|_\infty \\
\triangleq&\tilde{C}\|\lambda' - \lambda\|_\infty,
\end{aligned}
\tag{127}
$$

where $C_1 = \max_{i\in[M],j\in[K],h\in[D]} \sup_t \chi_h(x_i, c_j, \hat{\mathcal{P}}(t)) > 0$ is some constant and we define $\tilde{C} = DKMC_1^2/4C_{min}^{\frac{3}{2}}$.

Finally, we show that $\nabla\mathcal{Q}(\lambda, \mathcal{P})$ is continuous with respect to $\mathcal{P}$. We consider

$$
\begin{aligned}
&\left|[\nabla\mathcal{Q}(\lambda, \hat{\mathcal{P}}(t))]_{ij} - [\nabla\mathcal{Q}(\lambda, \mathcal{P})]_{ij}\right| \\
=&\left|\sum_{h\in[D]} \frac{\chi_h(x_i, c_j, \hat{\mathcal{P}}(t))}{2\sqrt{\sum_{i\in[K],j\in[M]} \lambda_{ij}\chi_h(x_i, c_j, \hat{\mathcal{P}}(t))}} - \sum_{h\in[D]} \frac{\chi_h(x_i, c_j, \mathcal{P})}{2\sqrt{\sum_{i\in[K],j\in[M]} \lambda_{ij}\chi_h(x_i, c_j, \mathcal{P})}}\right| \\
\leq&\sum_{h\in[D]} \left|\frac{\chi_h(x_i, c_j, \hat{\mathcal{P}}(t))}{2\sqrt{\sum_{i\in[K],j\in[M]} \lambda_{ij}\chi_h(x_i, c_j, \hat{\mathcal{P}}(t))}} - \frac{\chi_h(x_i, c_j, \mathcal{P})}{2\sqrt{\sum_{i\in[K],j\in[M]} \lambda_{ij}\chi_h(x_i, c_j, \mathcal{P})}}\right| \\
=&\sum_{h\in[D]} \frac{1}{2}\left|\frac{\chi_h(x_i, c_j, \hat{\mathcal{P}}(t))\sqrt{\sum_{i\in[K],j\in[M]} \lambda_{ij}\chi_h(x_i, c_j, \mathcal{P})} - \chi_h(x_i, c_j, \mathcal{P})\sqrt{\sum_{i\in[K],j\in[M]} \lambda_{ij}\chi_h(x_i, c_j, \hat{\mathcal{P}}(t))}}{\sqrt{\sum_{i\in[K],j\in[M]} \lambda_{ij}\chi_h(x_i, c_j, \hat{\mathcal{P}}(t))}\sqrt{\sum_{i\in[K],j\in[M]} \lambda_{ij}\chi_h(x_i, c_j, \mathcal{P})}}\right| \\
\leq&\sum_{h\in[D]} \frac{1}{2C_{min}}\left|\chi_h(x_i, c_j, \hat{\mathcal{P}}(t))\sqrt{\sum_{i\in[K],j\in[M]} \lambda_{ij}\chi_h(x_i, c_j, \mathcal{P})} - \chi_h(x_i, c_j, \hat{\mathcal{P}}(t))\sqrt{\sum_{i\in[K],j\in[M]} \lambda_{ij}\chi_h(x_i, c_j, \hat{\mathcal{P}}(t))}\right| \\
&+ \left|\chi_h(x_i, c_j, \hat{\mathcal{P}}(t))\sqrt{\sum_{i\in[K],j\in[M]} \lambda_{ij}\chi_h(x_i, c_j, \hat{\mathcal{P}}(t))} - \chi_h(x_i, c_j, \mathcal{P})\sqrt{\sum_{i\in[K],j\in[M]} \lambda_{ij}\chi_h(x_i, c_j, \hat{\mathcal{P}}(t))}\right| \\
=&\sum_{h\in[D]} \frac{1}{2C_{min}}\Bigg[\chi_h(x_i, c_j, \hat{\mathcal{P}}(t))\left|\sqrt{\sum_{i\in[K],j\in[M]} \lambda_{ij}\chi_h(x_i, c_j, \mathcal{P})} - \sqrt{\sum_{i\in[K],j\in[M]} \lambda_{ij}\chi_h(x_i, c_j, \hat{\mathcal{P}}(t))}\right| \\
&+ \sqrt{\sum_{i\in[K],j\in[M]} \lambda_{ij}\chi_h(x_i, c_j, \hat{\mathcal{P}}(t))}\left|\chi_h(x_i, c_j, \hat{\mathcal{P}}(t)) - \chi_h(x_i, c_j, \mathcal{P})\right|\Bigg] \\
\leq&\frac{D}{2C_{min}}(C_1\bar{C} + \sqrt{C_1}L)\|\hat{\mathcal{P}}(t) - \mathcal{P}\|_\infty
\end{aligned}
\tag{128}
$$

Define a subsequence $\mathcal{T} \subset \{1, 2\dots\}$ such that the line search algorithm is invoked at time step $t \in \mathcal{T}$. Let $\bar{\lambda}$ be a limit point of the sequence $\{\lambda(t)\}$. Then, by definition, there exists a subsequence $\mathcal{T}_1 \subset \mathcal{T}$ such that

$$
\lim_{t\to\infty, t\in\mathcal{T}_1} \lambda(t-1) = \bar{\lambda}.
\tag{129}
$$

Since the index pair $(m(t), n(t)) \in [K] \times [M]$ takes values from a finite set, we can further extract a subsequence $\mathcal{T}_2 \subset \mathcal{T}_1$ and a fixed index pair $(m, n) \in [K] \times [M]$ such that

$$\lambda_{mn}(t-1) \geq \eta, \mathcal{D}^{m(t),n(t)}(\lambda(t-1)) = \mathcal{D}^{m,n}(\lambda(t-1)), Conv(\mathcal{D}^{m,n}(\bar{\lambda})) = \mathcal{F}(\bar{\lambda}), \quad (130)$$

where

$$\mathcal{F}(\bar{\lambda}) = \{d \in \mathbb{R}^{KM} : \sum_{j \in [M], i \in [K]} d_{ij} = 0, \ d_{ij} \geq 0, \text{if } \bar{\lambda}_{ij} = 0\}. \quad (131)$$

denote the set of all feasible directions at $\bar{\lambda}$.

We proceed by contradiction. Suppose that $\bar{\lambda}$ is not a stationary point of the dual optimization problem (14). Then, by Lemma 3, there exists a feasible direction $\bar{d} \in \mathcal{D}^{m,n}(\bar{\lambda})$ such that

$$\nabla \mathcal{Q}(\bar{\lambda}, \mathcal{P})^\top \bar{d} < 0. \quad (132)$$

From the previous argument, we know that $\lambda(t-1) \to \bar{\lambda}$ as $t \to \infty, t \in \mathcal{T}_2$. Therefore, for sufficiently large $t \in \mathcal{T}_2$, we have that $\bar{d} \in \mathcal{D}^{m,n}(\lambda(t-1))$, due to the continuity of the reduced feasible direction set with respect to $\lambda$. Moreover, since $\hat{\mathcal{P}}(t) \to \mathcal{P}$ almost surely, and $\nabla \mathcal{Q}(\lambda, \mathcal{P})$ is continuous in its arguments, it follows that for sufficiently large $t \in \mathcal{T}_2$,

$$\nabla \mathcal{Q}(\lambda(t-1), \hat{\mathcal{P}}(t))^\top \bar{d} < 0. \quad (133)$$

By Proposition A.1 in Lin et al. (2009), there exists a constant $c > 0$ such that, for sufficiently large $t$, the maximum step size $s^{max}(\bar{d}, \lambda(t-1)) \geq c$. For simplicity, we denote $s^{max}(\bar{d}, \lambda(t-1))$ by $s^{max}$ when no ambiguity arises.

The following analysis is motivated by the proof of Theorem 6 in Zhou et al. (2024), aiming to mitigate the effect of noise and ensure that the objective function is monotone decreasing. Observe that

$$\mathcal{Q}(\lambda(t-1), \mathcal{P}) - \mathcal{Q}(\lambda(t), \mathcal{P})$$
$$= \mathcal{Q}(\lambda(t-1), \mathcal{P}) - \mathcal{Q}(\lambda(t-1), \hat{\mathcal{P}}(t)) + \mathcal{Q}(\lambda(t-1), \hat{\mathcal{P}}(t)) - \mathcal{Q}(\lambda(t), \hat{\mathcal{P}}(t)) + \quad (134)$$
$$\mathcal{Q}(\lambda(t), \hat{\mathcal{P}}(t)) - \mathcal{Q}(\lambda(t), \mathcal{P})$$

By the continuity of $\mathcal{Q}(\lambda, \mathcal{P})$ in $\mathcal{P}$ and the law of the iterated logarithm, we have:

$$\mathcal{Q}(\lambda(t-1), \mathcal{P}) - \mathcal{Q}(\lambda(t-1), \hat{\mathcal{P}}(t)) + \mathcal{Q}(\lambda(t), \hat{\mathcal{P}}(t)) - \mathcal{Q}(\lambda(t), \mathcal{P}) = \mathcal{O}(\sqrt{\log \log t / t}) \quad (135)$$

From the definition of the duality-based decomposition algorithm, for $t \in \mathcal{T}_2$ and sufficiently large $t$, it holds that:

$$s^{max}(d(t), \lambda(t-1)) \nabla Q(\lambda(t-1), \hat{\mathcal{P}}(t))^\top d(t) \leq s^{max}(\bar{d}, \lambda(t-1)) \nabla Q(\lambda(t-1), \hat{\mathcal{P}}(t))^\top \bar{d} < 0 \quad (136)$$

Since the second derivative of $\mathcal{Q}(\lambda, \hat{\mathcal{P}}(t))$ with respect to each $\lambda_{ij}$ is bounded, applying Taylor's theorem yields:

$$\mathcal{Q}(\lambda(t-1) + s(t)d(t), \hat{\mathcal{P}}(t)) \leq \mathcal{Q}(\lambda(t-1), \hat{\mathcal{P}}(t)) + s(t) \nabla \mathcal{Q}(\lambda(t-1), \hat{\mathcal{P}}(t))^\top d(t) + \frac{s(t)^2 \tilde{C}}{2} \|d(t)\|_2^2. \quad (137)$$

Hence, the line search stopping condition is satisfied if

$$\mathcal{Q}(\lambda(t-1), \hat{\mathcal{P}}(t)) + s(t) \nabla \mathcal{Q}(\lambda(t-1), \hat{\mathcal{P}}(t))^\top d(t) + \frac{s(t)^2 \tilde{C}}{2} \|d(t)\|_2^2$$
$$\leq \mathcal{Q}(\lambda(t-1), \hat{\mathcal{P}}(t)) + \alpha s(t) \nabla \mathcal{Q}(\lambda(t-1), \hat{\mathcal{P}}(t))^\top d \quad (138)$$

Letting $s(t) \leq \frac{(\alpha-1)\nabla \mathcal{Q}(\lambda(t-1), \hat{\mathcal{P}}(t))^\top d(t)}{\tilde{C}} = \frac{(\alpha-1)\mathcal{W}(t)}{\tilde{C}}$ ensures the stopping condition is satisfied. Now consider two cases: if $s^{max}(d(t), \lambda(t-1)) \leq \frac{(\alpha-1)\mathcal{W}(t)}{\tilde{C}}$, then the step size selected is $s(t) = s^{max}(d(t), \lambda(t-1))$, and

$$\mathcal{Q}(\lambda(t-1), \hat{\mathcal{P}}(t)) - \mathcal{Q}(\lambda(t), \hat{\mathcal{P}}(t)) \geq -\alpha s^{max}(d(t), \lambda(t-1))\mathcal{W}(t). \quad (139)$$

Otherwise, the algorithm chooses $s(t) = \frac{(\alpha-1)\mathcal{W}(t)}{\tilde{C}}$, resulting in

$$\mathcal{Q}(\lambda(t-1), \hat{\mathcal{P}}(t)) - \mathcal{Q}(\lambda(t), \hat{\mathcal{P}}(t)) \geq \frac{\alpha v(1-\alpha)\mathcal{W}(t)^2}{\tilde{C}}. \tag{140}$$

Therefore, we have that

$$\mathcal{Q}(\lambda(t-1), \hat{\mathcal{P}}(t)) - \mathcal{Q}(\lambda(t), \hat{\mathcal{P}}(t)) \geq \min\left\{ -\alpha s^{max}(d(t), \lambda(t-1))\mathcal{W}(t), \frac{\alpha v(1-\alpha)\mathcal{W}(t)^2}{\tilde{C}} \right\}. \tag{141}$$

By the definition of Algorithm 2

$$\mathcal{Q}(\lambda(t-1), \hat{\mathcal{P}}(t)) - \mathcal{Q}(\lambda(t), \hat{\mathcal{P}}(t)) \geq \Omega\left(\sqrt{\frac{\log t}{t}}\right). \tag{142}$$

Combining this with the earlier bound on the noise error gives:

$$\mathcal{Q}(\lambda(t-1), \mathcal{P}) - \mathcal{Q}(\lambda(t), \mathcal{P}) \geq \mathcal{O}\left(\sqrt{\frac{\log \log t}{t}}\right) + \Omega\left(\sqrt{\frac{\log t}{t}}\right) > 0, \tag{143}$$

which establishes that the objective function is monotone decreasing for sufficiently large $t$.

Moreover, note that $\mathcal{Q}(\lambda(t-1), \mathcal{P})$ is bounded below since, for any feasible $\lambda$,

$$\mathcal{Q}(\lambda, \mathcal{P}) = -\sum_{h \in [D]} \sqrt{\sum_{i \in [K], j \in [M]} \lambda_{ij} \chi_h(x_i, c_j)} \geq -\sum_{h \in [D]} \sqrt{\sum_{i \in [K], j \in [M]} \chi_h(x_i, c_j)}. \tag{144}$$

Then the sequence $\{\mathcal{Q}(\lambda(t-1), \mathcal{P})\}$ will converge to a finite value. By continuity of $\mathcal{Q}(\lambda, \mathcal{P})$ in $\lambda$, we have:

$$\lim_{t \to \infty, t \in \mathcal{T}_2} \mathcal{Q}(\lambda(t-1), \mathcal{P}) = \mathcal{Q}(\bar{\lambda}, \mathcal{P}), \tag{145}$$

which means

$$\lim_{t \to \infty, t \in \mathcal{T}_2} \mathcal{Q}(\lambda(t-1), \mathcal{P}) - \mathcal{Q}(\lambda(t-1) + s(t)d(t), \mathcal{P}) = 0. \tag{146}$$

From Lemma 6, it follows that:

$$\lim_{t \to \infty} s^{max}(d(t), \lambda(t-1))\nabla\mathcal{Q}(\lambda(t-1), \mathcal{P})^\top d(t) = 0, \tag{147}$$

which yields

$$\nabla\mathcal{Q}(\bar{\lambda}, \mathcal{P})^\top \bar{d} = 0 \tag{148}$$

contradicting the assumed condition in (132). Hence, $\bar{\lambda}$ must be a stationary point of the dual problem (14). □

We are now ready to establish the sample complexity upper bound stated in Theorem 3. Our analysis builds on the framework proposed by Garivier & Kaufmann (2016), which has been widely adopted in the BAI literature (Juneja & Krishnasamy, 2019; Wang et al., 2021).

*Proof.* We begin by defining the following clean event:

$$\mathcal{E} = \left\{ \max_{h \in [D]} \left| \frac{N_h(t)}{t} - \omega_h^*(\mathcal{P}) \right| \to 0, \hat{\mathcal{P}}(t) \to \mathcal{P} \right\}. \tag{149}$$

By Lemma 8, every limit point of the sequence $\{\lambda(t)\}$, generated by the algorithm, is a stationary point of the dual problem (14).

Moreover, by Lemma 2, strong duality holds. Hence, we can recover a solution sequence $\gamma(\hat{\mathcal{P}}(t))$ to the primal problem (13) via (16), and every limit point of $\{\gamma(\hat{\mathcal{P}}(t))\}$ is an optimal solution to the primal problem. That is, for any $\epsilon > 0$, there exists $t_0 > 0$ such that:

$$\sup_{t \geq t_0} \max_{h \in [D]} \left| \gamma_h(\hat{\mathcal{P}}(t)) - \omega_h^*(\mathcal{P}) \right| \leq \epsilon. \tag{150}$$

Furthermore, by Lemma 7, there exists $t_1 > 0$ such that

$$\sup_{t \geq t_1} \max_{h \in [D]} \left| \frac{N_h(t)}{t} - \omega_h^*(\mathcal{P}) \right| \leq 3(D-1)\epsilon. \tag{151}$$

In addition, since $N_h(t) \geq (\sqrt{t} - D/2)_+ - 1$, the strong law of large numbers implies that $\hat{\mathcal{P}}(t) \to \mathcal{P}$ almost surely. Therefore, we conclude: $\mathbb{P}(\mathcal{E}) = 1$.

Condition on the clean event $\mathcal{E}$, by Lemma 4, the function $\Gamma^s(\omega, c_j, \mathcal{P})$ is continuous in both $\omega$ and $\mathcal{P}$. Thus, for any $\epsilon > 0$, there exists $t_0 > 0$ such that for all $t \geq t_0$,

$$\mathcal{U}(\hat{\mathcal{P}}(t), \omega_t)^{-1} \geq (1 - \epsilon)\mathcal{U}(\mathcal{P}, \omega^*(\mathcal{P}))^{-1}. \tag{152}$$

Since $\rho(t, \delta) = \log(\frac{Ct^\alpha}{\delta}) = o(t)$, there exists $t_1 > 0$ such that for all $t \geq t_1$, we have

$$\rho(t, \delta) \leq \log(1/\delta) + \epsilon\mathcal{U}(\mathcal{P}, \omega^*(\mathcal{P}))^{-1}t. \tag{153}$$

Then, the stopping time $\tau$ satisfies:

$$\tau = \inf\left\{ t \in \mathbb{N} : t\mathcal{U}(\hat{\mathcal{P}}(t), \omega(t))^{-1} \geq \rho(t, \delta) \right\}$$

$$= t_0 + t_1 + \inf\left\{ t \in \mathbb{N} : t\mathcal{U}(\hat{\mathcal{P}}(t), \omega(t))^{-1} \geq \log(1/\delta) + \epsilon\mathcal{U}(\hat{\mathcal{P}}(t), \omega(t))^{-1}t \right\}$$

$$= t_0 + t_1 + \inf\left\{ t \in \mathbb{N} : t(1 - \epsilon)\mathcal{U}(\mathcal{P}, \omega^*(\mathcal{P}))^{-1} \geq \log(1/\delta) + \epsilon\mathcal{U}(\hat{\mathcal{P}}(t), \omega(t))^{-1}t \right\} \tag{154}$$

$$= t_0 + t_1 + \inf\left\{ t \in \mathbb{N} : t(1 - 2\epsilon)\mathcal{U}(\mathcal{P}, \omega^*(\mathcal{P}))^{-1} \geq \log(1/\delta) \right\}$$

$$= t_0 + t_1 + \frac{\mathcal{U}(\mathcal{P}, \omega^*(\mathcal{P}))\log(1/\delta)}{1 - 2\epsilon}.$$

Therefore,

$$\limsup_{\delta \to 0} \frac{\tau}{\log(1/\delta)} \leq \frac{\mathcal{U}(\mathcal{P}, \omega^*(\mathcal{P}))}{1 - 2\epsilon}, \tag{155}$$

and letting $\epsilon \to 0$, we obtain

$$\mathbb{P}\left( \limsup_{\delta \to 0} \frac{\tau}{\log(1/\delta)} \leq \mathcal{U}^*(\mathcal{P}) \right) = 1. \tag{156}$$

Next, we establish an upper bound on $\mathbb{E}[\tau]$. By Lemma 4, the function $\mathcal{U}(\omega, \mathcal{P})^{-1}$ is continuous in both $\omega$ and $\mathcal{P}$. Therefore, for any $\epsilon > 0$, there exists $\xi_1(\epsilon) > 0$ such that for all $\hat{\mathcal{P}}(t), \omega_t$ satisfying

$$\|\hat{\mathcal{P}}(t) - \mathcal{P}\|_\infty \leq \xi_1(\epsilon), \quad \|\omega_t - \omega^*(\mathcal{P})\|_\infty \leq \xi_1(\epsilon), \tag{157}$$

we have

$$\mathcal{U}(\omega_t, \hat{\mathcal{P}}(t))^{-1} \geq (1 - \epsilon)\mathcal{U}(\omega^*(\mathcal{P}), \mathcal{P})^{-1}. \tag{158}$$

Since the sequence $\gamma(\hat{\mathcal{P}}(t))$ converges to a stationary point $\omega^*(\mathcal{P})$ of the primal optimization problem, there exists $\xi_2(\epsilon) > 0$ such that for any $\hat{\mathcal{P}}(t)$ with

$$\|\hat{\mathcal{P}}(t) - \mathcal{P}\|_\infty \leq \xi_2(\epsilon), \tag{159}$$

we have

$$\|\gamma(\hat{\mathcal{P}}(t)) - \omega^*(\mathcal{P})\|_\infty < \frac{\xi_1(\epsilon)}{3(D-1)}. \tag{160}$$

Define $\xi(\epsilon) = \min\{\xi_1(\epsilon), \xi_2(\epsilon)\}$, define the event

$$\mathcal{E}_T = \bigcap_{t=T^{1/4}}^T \{\|\hat{\mathcal{P}}(t) - \mathcal{P}\|_\infty \leq \xi(\epsilon)\}. \tag{161}$$

Let $\epsilon_1 = \frac{\xi_1(\epsilon)}{3(D-1)}$, then by Lemma 7, there exists a constant $T(\epsilon_1)$ such that for all $T \geq T(\epsilon_1)$, on the event $\mathcal{E}_T$, we have for all $t \geq T^{1/2}$,

$$\|\omega_t - \omega^*(\mathcal{P})\|_\infty \leq 3(D-1)\epsilon_1 = \xi_1(\epsilon). \tag{162}$$

Therefore, let $T \geq T(\epsilon_1)$, on the event $\mathcal{E}_T$, for all $\forall t \geq T^{1/2}$, we have

$$\mathcal{U}(\omega_t, \hat{\mathcal{P}}(t))^{-1} \geq (1-\epsilon)\mathcal{U}(\omega^*(\mathcal{P}), \mathcal{P})^{-1}. \tag{163}$$

This leads to the bound:

$$
\begin{aligned}
\min(\tau, T) &\leq T^{1/2} + \sum_{t=T^{1/2}}^{T} \mathbb{I}(\tau > t) \\
&\leq T^{1/2} + \sum_{t=T^{1/2}}^{T} \mathbb{I}(t\mathcal{U}(\omega_t, \hat{\mathcal{P}}(t))^{-1} \leq \rho(t, \delta)) \\
&\leq T^{1/2} + \sum_{t=T^{1/2}}^{T} \mathbb{I}(t \leq \frac{\rho(T, \delta)}{(1-\epsilon)\mathcal{U}(\omega^*(\mathcal{P}), \mathcal{P})^{-1}}) \\
&\leq T^{1/2} + \frac{\rho(T, \delta)\mathcal{U}(\omega^*(\mathcal{P}), \mathcal{P})}{(1-\epsilon)}.
\end{aligned}
\tag{164}
$$

Define

$$T_1^*(\delta) = \inf\left\{ T \in \mathbb{N} : T^{1/2} + \frac{\rho(T, \delta)\mathcal{U}(\omega^*(\mathcal{P}), \mathcal{P})}{1-\epsilon} \leq T \right\} \tag{165}$$

Then for all $T \geq \max(T(\epsilon_1), T_1^*(\delta))$, it holds that $\mathcal{E}_T \subset (\tau \leq T)$.

Thus, we obtain:

$$
\begin{aligned}
\mathbb{E}[\tau] &= \sum_{T=1}^{\infty} \mathbb{P}(\tau \geq T) \\
&\leq T(\epsilon_1) + T_1^*(\delta) + \sum_{T=1}^{\infty} \mathbb{P}(\tau \geq T) \\
&= T(\epsilon_1) + T_1^*(\delta) + \sum_{T=1}^{\infty} \left( \mathbb{P}(\mathcal{E}_T)\mathbb{P}(\tau \geq T | \mathcal{E}_T) + \mathbb{P}(\mathcal{E}_T^c)\mathbb{P}(\tau \geq T | \mathcal{E}_T^c) \right) \\
&\leq T(\epsilon_1) + T_1^*(\delta) + \sum_{T=1}^{\infty} \mathbb{P}(\mathcal{E}_T^c)
\end{aligned}
\tag{166}
$$

By Lemma 18 of Garivier & Kaufmann (2016), we know

$$T_1^*(\delta) = \frac{\mathcal{U}(\omega^*(\mathcal{P}), \mathcal{P})}{1-\epsilon}(\mathcal{O}(\log(1/\delta)) + \mathcal{O}(\log\log(1/\delta))) \tag{167}$$

To upper bound $\sum_{T=1}^{\infty} \mathbb{P}(\mathcal{E}_T^c)$, observe:

$$
\begin{aligned}
&\mathbb{P}(\mathcal{E}_T^c) \\
&= \mathbb{P}\left( \bigcup_{t=T^{1/4}}^{T} \left\{ \|\hat{\mathcal{P}}(t) - \mathcal{P}\|_\infty > \xi(\epsilon) \right\} \right) \\
&\leq \sum_{t=T^{1/4}}^{T} \sum_{h=1}^{D} \mathbb{P}\left( \left|\bar{F}(z_h; t) - f(z_h)\right| > \xi(\epsilon) \right) + \mathbb{P}\left( \left|\bar{G}(z_h; t) - g(z_h)\right| > \xi(\epsilon) \right).
\end{aligned}
\tag{168}
$$

Since we have

$$
\mathbb{P}(\bar{F}(z_h; t) < f(z_h) - \xi(\epsilon))
$$

$$
= \mathbb{P}(\bar{F}(z_h; t) < f(z_h) - \xi(\epsilon), N_h(t) \geq \sqrt{t} - D)
$$

$$
\leq \sum_{s=\sqrt{t}-D}^{t} \mathbb{P}(\bar{F}_s(z_h) \leq f(z_h) - \xi(\epsilon)) \tag{169}
$$

$$
\leq \sum_{s=\sqrt{t}-D}^{t} e^{(-sd(f(z_h)-\xi(\epsilon),f(z_h)))}
$$

$$
\leq \frac{1}{1 - e^{d(f(z_h)-\xi(\epsilon),f(z_h))}} e^{-(\sqrt{t}-D)d(f(z_h)-\xi(\epsilon),f(z_h))},
$$

where $\bar{F}_s(z_h)$ denotes the empirical mean of the first $s$ samples. Similarly, we can also show that

$$
\mathbb{P}(\bar{F}(z_h; t) > f(z_h) - \xi(\epsilon)) \leq \frac{1}{1 - e^{d(f(z_h)+\xi(\epsilon),f(z_h))}} e^{-(\sqrt{t}-D)d(f(z_h)+\xi(\epsilon),f(z_h))}, \tag{170}
$$

By choosing

$$
C = \min_{h \in [D]} \min(d(f(z_h) - \xi(\epsilon), f(z_h)), d(f(z_h) + \xi(\epsilon), f(z_h)),
$$
$$
d(g(z_h) - \xi(\epsilon), g(z_h)), d(g(z_h) + \xi(\epsilon), g(z_h))), \tag{171}
$$

and

$$
B = \sum_{h \in [D]} \left( \frac{e^{Dd(f(z_h)-\xi(\epsilon),f(z_h))}}{1 - e^{d(f(z_h)-\xi(\epsilon),f(z_h))}} + \frac{e^{Dd(f(z_h)+\xi(\epsilon),f(z_h))}}{1 - e^{d(f(z_h)+\xi(\epsilon),f(z_h))}} \right.
$$
$$
\left. + \frac{e^{Dd(g(z_h)-\xi(\epsilon),g(z_h))}}{1 - e^{d(g(z_h)-\xi(\epsilon),g(z_h))}} + \frac{e^{Dd(g(z_h)+\xi(\epsilon),g(z_h))}}{1 - e^{d(g(z_h)+\xi(\epsilon),g(z_h))}} \right). \tag{172}
$$

Therefore,

$$
\mathbb{P}(\mathcal{E}_T^c) \leq B \sum_{t=T^{1/4}}^{T} \exp(-C\sqrt{t}) \leq BT \exp(-CT^{1/8}), \tag{173}
$$

and therefore $\sum_{T=1}^{\infty} \mathbb{P}(\mathcal{E}_T^c) \leq \infty$. Finally, this leads to the conclusion:

$$
\limsup_{\delta \to 0} \frac{\mathbb{E}[\tau]}{\log(1/\delta)} \leq \frac{1}{1 - \epsilon} \mathcal{U}(\omega^*(\mathcal{P}), \mathcal{P}). \tag{174}
$$

Letting $\epsilon \to 0$ completes the proof. □

## A.12 COMPUTATIONAL COMPLEXITY

Since the main difference between Algorithm 1 (TS) and the proposed Algorithm (DSR) lies in how the empirical optimal sampling ratio is computed, we focus on this step. In TS, assuming gradient descent is used, evaluating the objective function involves a matrix inversion $\mathcal{O}(D^3)$, an inner minimization over $K$ arms and $M$ covariates $\mathcal{O}(MK)$, and gradient computation $\mathcal{O}(D)$, leading to a total per-iteration complexity of $\mathcal{O}\left(\frac{1}{\epsilon}(D^3 + MK + D)\right)$, where $\epsilon$ denotes the allowed error precision for the optimization problem. In DSR, only one gradient step is performed per iteration. The matrix inversion involved in the dual objective function is done once and reused, while each iteration involves objective evaluation and descent direction computation $\mathcal{O}(MK)$ and line search $\mathcal{O}(\log(1/\epsilon'))$ for precision $\epsilon'$, resulting in a total per-iteration complexity of $\mathcal{O}(MK + \log(1/\epsilon'))$.

## A.13 NUMERICAL EXPERIMENT

This subsection provides the detailed parameter settings and pseudo-code for the benchmark algorithms used in the numerical experiments.

**DSR**. Algorithm 4 outlines the complete pseudo-code for the proposed duality-based decomposition algorithm. The overall framework follows the structure of the Track-and-Stop algorithm,

with the key difference being that the sampling ratio $\gamma(\hat{\mathcal{P}}(t))$ is computed using Algorithm 2. In our implementation, we adopt a heuristic step size of $s(t) = 0.01$ and a threshold parameter $\rho(t, \delta) = \log(\log(t) + 1)/\delta$, the latter of which is commonly used in the best arm identification (BAI) literature (Garivier & Kaufmann, 2016; Wang et al., 2021).

---

**Algorithm 4:** Duality-based Decomposition Algorithm (DSR)

1 **Input:** Covariate set $\mathcal{C}$, arm set $\mathcal{X}$, design point set $\mathcal{Z}$, confidence level $\delta$, $\lambda(0) = 1/KM$.
2 **Initialization:** Sample each design point $z_h \in \mathcal{Z}$ $n_0$ times.
3 Set $t \leftarrow n_0 D$ and update $N_h(t), \omega_h(t), \hat{\mathcal{P}}(t), \Lambda(\omega(t))$.
4 **while** $t\mathcal{H}(\hat{\mathcal{P}}(t), \omega(t))^{-1} < \rho(t, \delta)$ **do**
5      **if** $\mathcal{B}_t \neq \emptyset$ **then**
6          $z_{h(t+1)} = \arg\min_{z_h \in \mathcal{B}_t} N_h(t)$
7      **else**
8          $\gamma(\hat{\mathcal{P}}(t)) \leftarrow$ Algorithm 2 $(\mathcal{C}, \mathcal{X}, \mathcal{Z}, \kappa_0, \eta, \hat{\mathcal{P}}(t), \hat{\theta}(t), \hat{\beta}(t), \lambda(t-1))$
9          $z_{h(t+1)} = \arg\min_{z_h \in \mathcal{Z}} N_h(t) - t\gamma_h(\hat{\mathcal{P}}(t))$
10      Sample the design point $z_{h(t+1)}$ and obtain the observation $Z_{t+1}$.
11      Set $t \leftarrow t + 1$, and update $N_h(t), \omega_h(t), \hat{\mathcal{P}}(t), \Lambda(\omega(t))$.
12 **return** For each covariate $c_j \in \mathcal{C}$, recommend the estimated best arm:
$$x_{\hat{i}(c_j;\tau)} = \arg\max_{x_i \in \mathcal{X}} \hat{\theta}(\tau)^\top \phi(x_i, c_j) \quad \text{s.t.} \quad \hat{\beta}(\tau)^\top \phi(x_i, c_j) \leq b$$

---

**USR**. Algorithm 5 presents the pseudo-code for the USR algorithm. At each time step $t$, it samples all design points uniformly, without incorporating any information from the arms.

---

**Algorithm 5:** USR Algorithm

1 **Input:** Covariate set $\mathcal{C}$, arm set $\mathcal{X}$, design point set $\mathcal{Z}$, confidence level $\delta$.
2 **while** $t\mathcal{H}(\hat{\mathcal{P}}(t), \omega(t))^{-1} < \rho(t, \delta)$ **do**
3      $z_{h(t+1)} = \arg\min_{z_h \in \mathcal{Z}} N_h(t)$
4      Sample the design point $z_{h(t+1)}$ and obtain the observation $Z_{t+1}$.
5      Set $t \leftarrow t + 1$, and update $N_h(t), \omega_h(t), \hat{\mathcal{P}}(t), \Lambda(\omega(t))$.
6 **return** For each covariate $c_j \in \mathcal{C}$, recommend the estimated best arm:
$$x_{\hat{i}(c_j;\tau)} = \arg\max_{x_i \in \mathcal{X}} \hat{\theta}(\tau)^\top \phi(x_i, c_j) \quad \text{s.t.} \quad \hat{\beta}(\tau)^\top \phi(x_i, c_j) \leq b$$

---

Algorithm 6 presents the pseudo-code for the BCSR, GOSR, and GFSR algorithms. All three algorithms employ a score-based approach to determine the sampling rule, with the key distinction being how each algorithm defines its respective score.

**BCSR**. This algorithm is inspired by the state-of-the-art Best Challenger algorithm proposed by Garivier & Kaufmann (2016). It relies solely on the optimality information of each arm. For each design point, the score at time step $t$ is defined as:

$$S_h(\hat{\mathcal{P}}(t), \omega(t)) = \frac{(\hat{f}(z_h; t) - \hat{f}(x_{\hat{i}(c_j;t)}, c_j))^2}{\sigma_h^2 / N_h(t)}, \tag{175}$$

where $x_{\hat{i}(c_j;t)} = \arg\max_{x_i \in \mathcal{X}} \hat{\theta}(t)^\top \phi(x_i, c_j)$ denotes the estimated best arm under covariate $c_j$. This score captures a trade-off between the estimated optimality gap and the sampling variance.

If the design point corresponds to the estimated best arm, then its score is defined as:

$$S_h(\hat{\mathcal{P}}(t), \omega(t)) = \min_{z_h \in \mathcal{Z} \setminus (x_{\hat{i}(c_j;t)}, c_j)} S_h(\hat{\mathcal{P}}(t), \omega(t)). \tag{176}$$

meaning the best arm is assigned the minimum score. The algorithm then randomly selects among arms with the lowest score for sampling.

---

**Algorithm 6:** BCSR/GOSR/GFSR Algorithm

---

1 **Input:** Covariate set $\mathcal{C}$, arm set $\mathcal{X}$, design point set $\mathcal{Z}$, confidence level $\delta$.
2 **Initialization:** Sample each design point $z_h \in \mathcal{Z}$ $n_0$ times.
3 Set $t \leftarrow n_0 D$ and update $N_h(t), \omega_h(t), \hat{\mathcal{P}}(t), \Lambda(\omega(t))$.
4 **while** $t\mathcal{H}(\hat{\mathcal{P}}(t), \omega(t))^{-1} < \rho(t, \delta)$ **do**
5     **if** $\mathcal{B}_t \neq \emptyset$ **then**
6        $z_{h(t+1)} = \arg\min_{z_h \in \mathcal{B}_t} N_h(t)$
7     **else**
8        $z_{h(t+1)} = \arg\min_{z_h \in \mathcal{Z}} S_h(\hat{\mathcal{P}}(t), \omega(t))$
9     Sample the design point $z_{h(t+1)}$ and obtain the observation $Z_{t+1}$.
10     Set $t \leftarrow t+1$, and update $N_h(t), \omega_h(t), \hat{\mathcal{P}}(t), \Lambda(\omega(t))$.
11 **return** For each covariate $c_j \in \mathcal{C}$, recommend the estimated best arm:

$$x_{\hat{i}(c_j; \tau)} = \arg\max_{x_i \in \mathcal{X}} \hat{\theta}(\tau)^\top \phi(x_i, c_j) \quad \text{s.t. } \hat{\beta}(\tau)^\top \phi(x_i, c_j) \leq b$$

---

**GOSR**. This algorithm is motivated by the surrogate optimization problem (13) and relies solely on optimality information. For each covariate $c_j \in \mathcal{C}$, the estimated best arm is defined as $x_{\hat{i}(c_j; t)} = \arg\max_{x_i \in \mathcal{X}} \hat{\theta}(\tau)^\top \phi(x_i, c_j)$. For each design point, the score at time step $t$ is defined as

$$S_h(\hat{\mathcal{P}}(t), \omega(t)) = \frac{(\hat{f}(z_h; t) - \hat{f}(x_{\hat{i}(c_j; t)}, c_j))^2}{\|\phi(x_{i^*(c_j)}, c_j) - \phi(x_i, c_j)\|^2_{\Lambda(\omega)^{-1}}}, \tag{177}$$

Similarly, the score for the estimated best arm is defined according to (176).

**GFSR**. The general algorithmic framework of GFSR is identical to that of GOSR, with the key distinction that GFSR relies solely on feasibility information to determine the sampling rule. Specifically, the score for each design point at time step $t$ is defined as

$$S_h(\hat{\mathcal{P}}(t), \omega(t)) = \frac{(\hat{g}(z_h; t) - \hat{g}(x_{\hat{i}(c_j; t)}, c_j))^2}{\|\phi(x_i, c_j)\|^2_{\Lambda(\omega)^{-1}}}, \tag{178}$$

where the score quantifies the deviation in feasibility performance. The score for the estimated best arm is defined in the same way as in (176).

**Comparison with Frank-Wolfe Sampling (Wang et al., 2021).** Wang et al. (2021) propose a general framework for pure exploration via Frank–Wolfe. However, our constrained setting with covariate selection leads to a more complex sample complexity bound, making their algorithm unsuitable for our problem. First, the presence of constraints complicates the gradient computation in Proposition 1 of Wang et al. (2021). The gradient calculation depends on the most confusing alternative instance. When constraints are considered, the alternative problem instance set $\mathcal{A}(\mathcal{P})$ becomes more complex, as it depends on both the optimality and feasibility of the arms. We need to classify arms into four subclasses and construct the alternative instance for each class differently. For infeasible arms with worse performance, the alternative instance is particularly complex, as it depends simultaneously on both the objective and the constraint performance measures. Second, the covariate selection setting makes the Frank–Wolfe update, which involves solving a game, more complicated. Wang et al. (2021) handle non-smooth objectives via the $r$-subdifferential subspace. In our setting, covariate selection introduces an additional layer of optimization over all possible covariates in the sample complexity lower bound. This increases the number of non-smooth points in the overall objective, making the Frank-Wolfe update, which solves the game over a simplex and the convex hull of the gradient vectors, more time-consuming.

We also compare the numerical performance of the proposed DSR with Frank-Wolfe Sampling (FWS) on the same problem used in the numerical experiment. Each algorithm is run for 3000 iterations, and we report the total running time and the empirical PCI over 30 independent macro replications. The results show that DSR completes in 56 seconds, whereas FWS takes 917 seconds, which is approximately 16 times longer than DSR. Moreover, Figure 3 shows that DSR achieves a PCI exceeding 0.9, while the PCI of FWS is below 0.8. Therefore, DSR also demonstrates superior empirical performance compared with FWS.

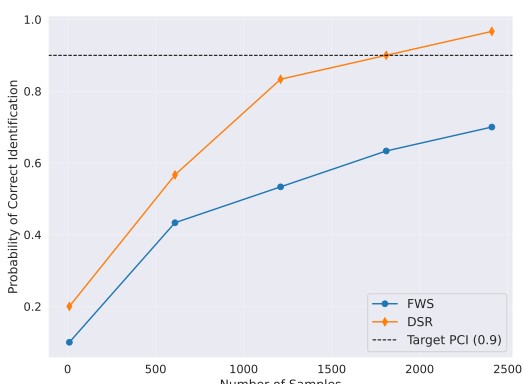

Figure 3: Comparison of empirical PCI

Table 1: Sample complexity comparison of various algorithms under different gaps

| Method | Mean (0.2) | Lower | Upper | Mean(0.3) | Lower | Upper |
|--------|-----------|----------|----------|-----------|---------|----------|
| USR | 12786.50 | 9756.38 | 15816.62 | 12313.53 | 9405.00 | 15222.06 |
| DSR | 5282.73 | 4023.10 | 6542.37 | 4100.33 | 2969.65 | 5231.01 |
| GOSR | 21274.73 | 14772.71 | 27776.76 | 8861.53 | 7200.30 | 10522.77 |
| GFSR | 6537.10 | 5241.74 | 7832.46 | 6526.23 | 5312.21 | 7740.25 |
| BCSR | 16936.70 | 12649.17 | 21224.23 | 7825.07 | 6505.59 | 9144.54 |

**Parameter setting.** The experimental setup is inspired by the numerical example in Soare et al. (2014). There are two covariates, $\mathcal{C} = \{c_1, c_2\}$, four arms, $\mathcal{X} = \{x_1, \ldots, x_4\}$, and one constraint. The threshold parameter $b$ in the constraint of problem (1) is set to $b = 0.5$. The dimension of the unknown parameter vectors $\theta$ and $\beta$ is $D = 7$. Specifically, $\theta = [1.0, 0.0, 0.0, 0.0, 1.0, 1.2, 0.0]^\top$, and $\beta = [0.45, 0.0, 0.0, 0.0, 0.6, 0.8]^\top$. Let $e_l \in \mathbb{R}^D$ denote the $l$th standard basic vector, with the $l$th element equal to one and all other elements zero. The feature vectors of the arm-covariate pairs are defined as $\phi(x_1, c_1) = e_1, \phi(x_2, c_1) = e_2, \ldots, \phi(x_3, c_2) = e_7$, and $\phi(x_4, c_2) = [\cos(0.4), \sin(0.4), 0, \ldots, 0]^\top$. The design point set is $\mathcal{Z} = \{(x_1, c_1), (x_2, c_1), \ldots, (x_3, c_2)\}$ with $|\mathcal{Z}| = 7$, meaning that the design points correspond to the standard basis vectors in $\mathbb{R}^D$. The variance of each arm-covariate pair is independently drawn from a uniform distribution over $[0.5, 1.0]$. For computational convenience during implementation, we use a heuristic step size $s(t) = 0.01$ and a threshold parameter $\rho(t, \delta) = \log(\log(t) + 1)/\delta$, the latter of which is also employed in the BAI literature (Garivier & Kaufmann, 2016; Wang et al., 2021).

**Robustness evaluation.** We report additional sample complexity results for small ($\Delta = 0.2$) and large ($\Delta = 0.3$) feasibility and optimality gaps to assess the robustness of the proposed algorithm across different problem instances. Table 1 summarizes the sample complexity of various algorithms at a confidence level of $\delta = 0.1$, with "lower" and "upper" indicating the 90% confidence interval bounds. Our proposed Algorithm DSR consistently outperforms other methods, and larger gaps correspond to lower sample complexity.

We also evaluate the algorithm's performance when the Gaussian noise assumption is violated. In this example, the problem setting remains the same, but the noise follows a standard $t$-distribution with 3 degrees of freedom, scaled by 0.1. Figure 4 presents the empirical sample complexity of the algorithms based on 30 macro-replications. The proposed DSR method continues to outperform the other benchmarks.

**Effect of covariate selection rule.** We examine the importance of covariate selection by comparing the sample complexity of DSR under different covariate selection rules. Specifically, we consider four rules: (1) **OPT**: active covariate selection according to the optimal sampling ratio; (2) **Uniform**: covariates are passively sampled from a uniform distribution; (3) **Covariate 1**: the two covariates are sampled with probabilities 0.8 and 0.2, respectively; (4) **Covariate 2**: the two covariates are

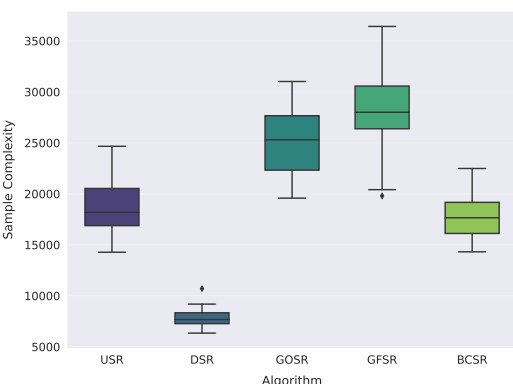

Figure 4: Empirical sample complexity over $t$-distribution noise

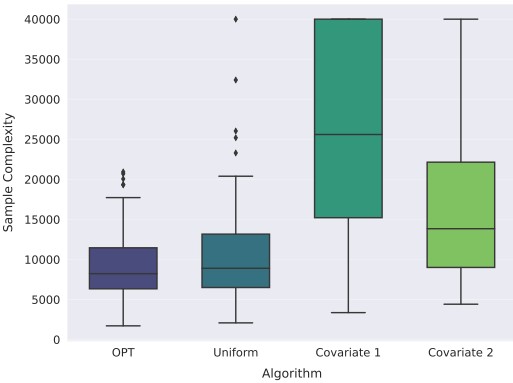

Figure 5: Sample complexity of DSR under different covariate selection rules

sampled with probabilities 0.2 and 0.8, respectively. Conditional on the covariate, the arm is sampled according to the optimal sampling ratio. To control the computation time, we set a maximum iteration limit of 40000, the algorithm terminates once the total number of samples reaches this threshold. Figure 5 presents the empirical sample complexity based on 100 independent macro-replications of DSR under the four covariate selection rules. The results indicate that optimal active covariate selection plays a crucial role in reducing sample complexity.

**Initial design points in $\mathcal{Z}$.** Figure 6 compares the sample complexity of DSR using three groups of different initial design points $\mathcal{Z}$ in the current numerical example. The result shows that, although different initial design points do lead to variations in sample complexity, the differences are not substantial. This indicates that DSR is relatively robust to the choice of initial design points.

**Problem scale and noise level.** We compare the sample complexity under different problem scales and noise levels. To control computation time, we impose a maximum iteration limit of 80000, and the algorithm terminates once the total number of samples reaches this threshold. Figure 7 reports the empirical sample complexity based on 30 independent macro-replications. As the problem size and noise level increase, the total number of samples required by all algorithms also increases. However, DSR consistently outperforms the other benchmarks.

**Experiments compute resources.** The numerical experiments were conducted on a Windows machine equipped with an Intel® Xeon® Silver 4210R CPU @ 2.40GHz. Running the algorithm for 100 replications took less than 1 hour.

### A.14    PERSONALIZED TREATMENT FOR DIABETES MANAGEMENT

Diabetes mellitus (DM) affects over 500 million people globally (World Health Organization), with type 2 diabetes (T2D) comprising 90–95% of cases. Managing T2D is complex, with treatment op-

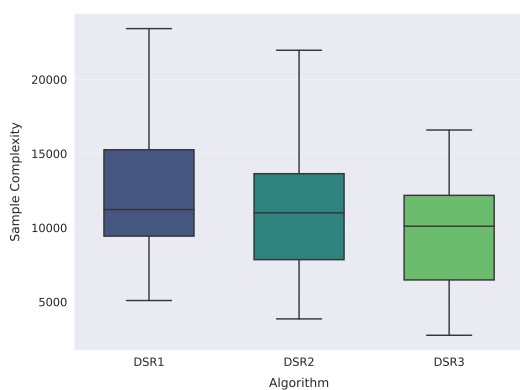

Figure 6: Sample complexity of DSR under different initial design points

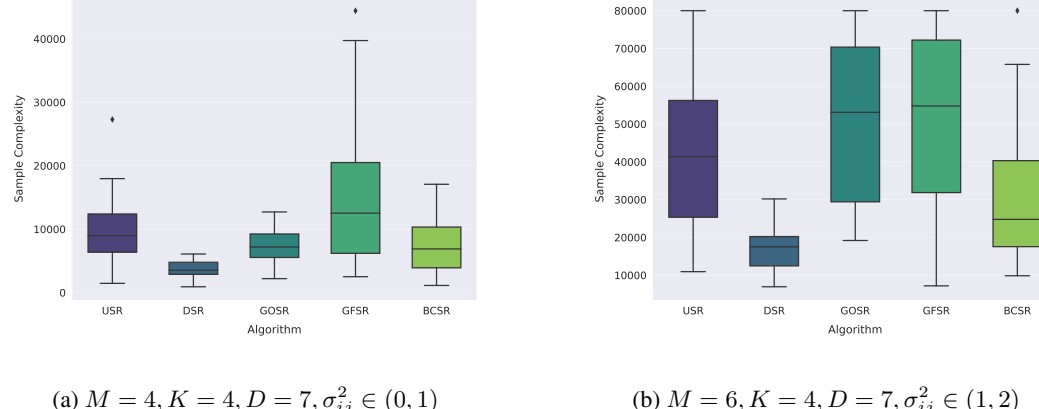

(a) $M = 4, K = 4, D = 7, \sigma_{ij}^2 \in (0, 1)$       (b) $M = 6, K = 4, D = 7, \sigma_{ij}^2 \in (1, 2)$

Figure 7: Sample complexity across different problem scales and noise levels

Table 2: Comparison of Methods with different confidence level $\delta$

| Method | Mean (0.1) | Lower | Upper | Mean (0.2) | Lower | Upper |
|--------|-----------|----------|----------|-----------|----------|----------|
| USR | 38661.00 | 30180.15 | 47141.85 | 25130.70 | 19243.79 | 31017.61 |
| DSR | 13127.07 | 10211.51 | 16042.62 | 11114.93 | 8236.33 | 13993.54 |
| GOSR | 16892.83 | 13055.05 | 20730.62 | 13779.97 | 10091.68 | 17468.25 |
| GFSR | 51852.90 | 41498.39 | 62207.41 | 49358.80 | 37399.20 | 61318.40 |
| BCSR | 17004.70 | 12753.25 | 21256.15 | 13786.23 | 9995.75 | 17576.71 |

tions ranging from lifestyle modifications to various pharmacological therapies such as Metformin, each with differing efficacy and side effect profiles depending on individual patient characteristics (covariates). Therefore, it is important to identify the most suitable treatment plan tailored to each patient's specific characteristics.

We model this as a constrained linear BAI problem with covariate selection. Based on ADA/EASD clinical guidelines, we consider four drug classes—Metformin, Sulfonylureas, SGLT2 inhibitors, and GLP-1 receptor agonists—each with distinct benefits and risks. For example, Metformin improves insulin sensitivity and is generally well-tolerated; however, it is contraindicated in patients with severe renal impairment.

Patient covariates include HbA1c, BMI, and cardiovascular risk. Drug features include dose, frequency, hypoglycemia risk, and renal adjustment threshold. The goal is to identify the treatment that maximizes glycemic improvement while maintaining adverse effects below a risk threshold for each patient.

Table 2 compares the sample complexity of various algorithms in a setting with 2 patients, 7-dimensional features (D = 7), and confidence levels $\delta = 0.1$ and $\delta = 0.2$. Our algorithm DSR, which balances feasibility and optimality, consistently achieves the lowest sample complexity.

