# OpenReview forum: "Constrained Linear Best Arm Identification with Covariate Selection"
_ICLR.cc/2026/Conference — Submitted to ICLR 2026_

### Official Review · Reviewer_Ey3L · 2025-10-24

**Soundness:** 2
**Presentation:** 3
**Contribution:** 3
**Rating:** 2
**Confidence:** 4

**Summary:**

The paper addresses fixed-confidence constrained best-arm identification (BAI) with active covariate selection in a discrete contextual linear bandit setting: each arm-covariate pair $(x_i,c_j)$ yields Gaussian observations with means linear in $\phi(x_i, c_j)$ optimizing $f = \phi^{\top}\theta$ subject to $g = \phi^\top\beta < b$. It derives an instance-dependent lower bound and a track and stop (TaS) procedure to match the allocations according to the proposed lower bound. Furthermore, it relaxes the primal sampling allocation to a dual program over simplex weights $\lambda$ with objective $-\sum_h \sqrt{\sum_{I,j}\lambda_{ij}\kappa_h(x_i,c_j)}$ and recovers the optimal static ratio corresponding to a surrogate complexity measure. A duality-based sampler (DSR) implements these ideas and, on synthetic tasks, outperforms uniform/greedy baselines in sample complexity.

**Strengths:**

- **Interesting setting.** Studies fixed-confidence constrained BAI with active covariate selection. To the best of my knowledge, this problem has not been looked at in the BAI context for pure strategies, even though there is some work in mixed-strategy setting.
- **Novel dual formulation & algorithm design.** Provides a TaS-based solution based on the problem complexity, and then relaxes to a surrogate complexity and a one-step gradient descent based strategy that ensures the convergence of the arm visitation fractions to the optimal solution of the surrogate problem.

**Weaknesses:**

(1) **Title/motivation (“covariate selection”):** The paper never exhibits a concrete instance where active covariate choice reduces sample complexity relative to a passive/contextual baseline while holding everything else fixed. Given the title, this is a material gap. A targeted experiment (same problem, (i) covariates passively sampled; (ii) agent forced to a fixed covariate; (iii) agent allowed to choose covariates) would resolve the ambiguity.

(2) **Setup/Assumption 1:** The model is essentially a discrete contextual linear bandit (transductive flavor), but Assumption 1 excludes the tight-constraint case. That case is exactly when feasibility and optimality trade off most sharply and is the main motivation for constraints. The paper should include the boundary regime in statements/proofs. Furthermore, the paper states that this “can be relaxed by identifying $\varepsilon$-optimal and feasible arms, as discussed in Degenne & Koolen (2019),” but no modification of the lower bound, GLRT, or proofs is given for the boundary case.

(3) There are some technical inconsistencies -- see "questions".

(4) Given the reference to “Fast pure exploration via Frank–Wolfe” (Wang–Tzeng–Proutière 2021), it is natural to ask whether FWS (which is computationally attractive and has optimality guarantees for linear BAI) can be adapted to your constrained + covariate-active setting. The paper neither discusses obstacles nor provides negative/positive results or experiments. If TaS is retained only to bridge to your dual relaxation, please articulate why FWS is inadequate (computationally or statistically) here, or provide a minimal comparison/ablation. As it stands, the algorithmic motivation is incomplete.

**Questions:**

There are some core technical inconsistencies. Here are my major concerns:

(1) I think that in (37), the RHS of the constraint should be $b$, not $0$. If that is the case, it is not obvious to me me how (38) follows from (37) -- can the authors provide the detailed steps?

(2) There is a discrepancy between the definition of $\chi_h$ in Theorem 2 and in (72) -- which one is correct?

Minor point:

Proposition 1 is inaccurate as is. It should be asymptotically optimal *up to $\alpha$*.

---

> ### Author Response · Authors · 2025-11-21
>
> Thank you for your insightful and constructive comments. We appreciate the opportunity to clarify the title and algorithmic motivation, and to provide additional results that address your concerns.
>
> **Comment 1:**
>
> > Title/motivation (“covariate selection”): The paper never exhibits a concrete instance where active covariate choice reduces sample complexity relative to a passive/contextual baseline while holding everything else fixed. Given the title, this is a material gap. A targeted experiment (same problem, (i) covariates passively sampled; (ii) agent forced to a fixed covariate; (iii) agent allowed to choose covariates) would resolve the ambiguity.
>
> **Response 1:**
>
> The sample complexity lower bound in Theorem 1 highlights the critical role of covariate selection. The second-layer optimization is taken over all possible covariates, implying that achieving optimal sample complexity requires actively selecting covariates.
>
> We provide a numerical example to illustrate this intuition in Appendix A.13 of the revised version. We examine the importance of covariate selection by comparing the sample complexity of DSR under different covariate selection rules. Specifically, we consider four rules: (1) **OPT**: active covariate selection according to the optimal sampling ratio; (2) **Uniform**: covariates are passively sampled from a uniform distribution; (3) **Covariate 1**: the two covariates are sampled with probabilities 0.8 and 0.2, respectively; (4) **Covariate 2**: the two covariates are sampled with probabilities 0.2 and 0.8, respectively. Conditional on the covariate, the arm is sampled according to the optimal sampling ratio. To control the computation time, we set a maximum iteration limit of $40000$, the algorithm terminates once the total number of samples reaches this threshold. The following table presents the empirical sample complexity based on $100$ independent macro-replications of DSR under the four covariate selection rules. The results indicate that optimal active covariate selection plays a crucial role in reducing sample complexity.
>
> | **Algorithm** | **Mean** | **90% CI Lower** | **90% CI Upper** |
> |---------------|----------|------------------|------------------|
> | OPT           | 9061.55  | 8368.99          | 9754.11          |
> | Uniform       | 10657.00 | 9621.36          | 11692.60         |
> | Covariate 1   | 25468.30 | 23365.70         | 27570.90         |
> | Covariate 2   | 17677.20 | 15853.30         | 19501.10         |
>
> **Comment 2:**
>
> > Setup/Assumption 1: The model is essentially a discrete contextual linear bandit (transductive flavor), but Assumption 1 excludes the tight-constraint case. That case is exactly when feasibility and optimality trade off most sharply and is the main motivation for constraints. The paper should include the boundary regime in statements/proofs. Furthermore, the paper states that this “can be relaxed by identifying $\epsilon$-optimal and feasible arms, as discussed in Degenne & Koolen (2019),” but no modification of the lower bound, GLRT, or proofs is given for the boundary case.
>
> **Response 2:**
>
> Extending the sample complexity result in Theorem 1 to include the boundary case is straightforward. We have added a new Section A.3 in the revised version to discuss this extension.
>
> **Comment 3:**
>
> > I think that in (37), the RHS of the constraint should be $b$ not $0$. If that is the case, it is not obvious to me me how (38) follows from (37) -- can the authors provide the detailed steps?
>
> **Response 3:**
>
> Thank you for your careful reading. This was a typo; the correct value is $b$ not $0$. We have corrected it in the revised version. In addition, we have included the detailed derivation from (37) to (38) to improve clarity.
>
>
> **Comment 4:**
>
> > There is a discrepancy between the definition of $\chi_h$ in Theorem 2 and in (72) -- which one is correct?
>
> **Response 4:**
>
> Thank you for your careful reading. This was a typo. The expression in Theorem 2 is correct, and we have revised the expression in (72) (now (77) in the updated version).
>
> **Comment 5:**
>
> Proposition 1 is inaccurate as is. It should be asymptotically optimal up to $\alpha$.
>
> **Response 5:**
>
> We have clarified this in the revised version.

---

> > ### Author Response · Authors · 2025-11-21
> >
> > **Comment 6:**
> >
> > > Given the reference to “Fast pure exploration via Frank–Wolfe” (Wang–Tzeng–Proutière 2021), it is natural to ask whether FWS (which is computationally attractive and has optimality guarantees for linear BAI) can be adapted to your constrained + covariate-active setting. The paper neither discusses obstacles nor provides negative/positive results or experiments. If TaS is retained only to bridge to your dual relaxation, please articulate why FWS is inadequate (computationally or statistically) here, or provide a minimal comparison/ablation. As it stands, the algorithmic motivation is incomplete.
> >
> > **Response 6:**
> >
> > Thank you for your insightful suggestion. Wang et al. (2021) propose a general framework for pure exploration via Frank–Wolfe. However, our constrained setting with covariate selection leads to a more complex sample complexity bound, making their algorithm unsuitable for our problem.
> >
> > First, the presence of constraints complicates the gradient computation in Proposition 1 of Wang et al. (2021). The gradient calculation depends on the most confusing alternative instance. When constraints are considered, the alternative problem instance set $\mathcal{A}(\mathcal{P})$ becomes more complex, as it depends on both the optimality and feasibility of the arms. We need to classify arms into four subclasses and construct the alternative instance for each class differently. For infeasible arms with worse performance, the alternative instance is particularly complex, as it depends simultaneously on both the objective and the constraint performance measures.
> >
> > Second, the covariate selection setting makes the Frank–Wolfe update, which involves solving a game, more complicated. Wang et al. (2021) handle non-smooth objectives via the $r$-subdifferential subspace. In our setting, covariate selection introduces an additional layer of optimization over all possible covariates in the sample complexity lower bound. This increases the number of non-smooth points in the overall objective, making the Frank-Wolfe update, which solves the game over a simplex and the convex hull of the gradient vectors, more time-consuming.
> >
> > The key message is that we introduce an interesting dual perspective for the complicated BAI problem. When the primal problem is difficult to solve, we can instead derive its dual formulation, which has a more favorable structure and can be solved efficiently, such as through the decomposition method we propose. To the best of our knowledge, this dual-based perspective is the first to be introduced in the BAI literature.
> >
> > We also compare the numerical performance of the proposed DSR with Frank Wolfe Sampling (FWS) on the same problem used in the numerical experiment. Each algorithm is run for 3000 iterations, and we report the total running time and the empirical probability of correct identification (PCI) over 30 independent macro replications. The results show that DSR completes in 56 seconds, whereas FWS takes 917 seconds, which is approximately 16 times longer than DSR. Moreover, DSR achieves a PCI exceeding 0.9, while the PCI of FWS is below 0.8. Therefore, DSR also demonstrates superior empirical performance compared with FWS. Because of the long running time of FWS, evaluating its sample complexity measure is very time consuming. These discussions and results are included in Appendix A.13 of the revised manuscript.
> >
> > [1] Wang P A, Tzeng R C, Proutiere A. ast pure exploration via frank-wolfe[J]. Advances in Neural Information Processing Systems, 2021, 34: 5810-5821.

---

> > > ### Comment · Reviewer_Ey3L · 2025-11-27
> > >
> > > Thanks, some of my concerns have been addressed. I have accordingly modified my score for the paper.

---

### Official Review · Reviewer_pghX · 2025-10-25

**Soundness:** 3
**Presentation:** 3
**Contribution:** 3
**Rating:** 8
**Confidence:** 2

**Summary:**

The paper studies fixed-confidence best arm identification (BAI) with both constraints and active covariate selection in a linear setting. The goal is to return, for every covariate, the feasible arm with maximal expected objective while keeping the constraint mean below a threshold. The authors derive an instance-dependent lower bound on sample complexity via a multi-level optimization and proves asymptotic tightness by a Track-and-Stop style algorithm. The paper then introduces a surrogate objective and proves an upper bound with a constant relaxation gap. To overcome the high computational complexity in original algorithm, a duality-based decomposition algorithm that is computationally efficient is proposed and proven to be $\delta$-PAC and achieves the relaxed complexity with supportive synthetic experiments and an application example.

**Strengths:**

1. Novel setting: The constrained, covariate-selective linear BAI captures realistic decision problems (e.g., treatment efficacy vs. side effects per patient group; inventory trade-offs) and differs materially from classical BAI and contextual bandits.
2. The information-theoretic derivation leads to a closed-form that decomposes feasibility and optimality contributions per covariate; the multi-constraint extension is spelled out.
3. The adaptation of Track-and-Stop to this setting (with an estimator-driven sampling ratio and generalized likelihood ratio stopping rule) matches the lower bound -> Asymptotically optimal. Proof for this adaption is sound and correct (with the right set of assumptions)
4. On synthetic problems (2 covariates, 4 arms, one constraint), DSR shows lower sample complexity than uniform and greedy baselines (USR/GOSR/GFSR) and a modified Best Challenger (BCSR), while achieving the target PCI; an application example appears in the appendix.
5. Algorithm novelty is clear and justified: The dual formulation (Theorem 2) and the resulting DSR updates are neat; the per-iteration complexity advantages over a direct Track-and-Stop implementation are explained.

**Weaknesses:**

1. Tightness of the relaxed bound in practice. While Lemma 1 and the appendix give a constant-factor relaxation $U^{\ast}\le CH^{\ast}$, the paper does not quantify the _empirical_ gap between $U^{\ast}$ and $H^{\ast}$ (e.g., as a function of geometry, noise, or constraints). Since DSR targets $U^{\ast}$, understanding when $U^{\ast}\approx H^{\ast}$ is critical to interpreting the practical optimality of DSR. A plot of $U^{\ast}/H^{\ast}$ over instances would strengthen the story.
2. The main text evaluates only a small synthetic instance (K=4 arms, M=2 covariates, one constraint), with heavier experiments relegated to the appendix and limited discussion of ablations (e.g., scaling in K, M, D; multiple constraints; noise levels).
3. The results hinge on linearity and Gaussian noise; Assumption 1 excludes ties and arms on the constraint boundary. While common in theory, it would help to show robustness to mild misspecification (e.g., sub-Gaussian noises).

**Questions:**

1. In large arm–covariate spaces, how do you construct the fixed set (Z) of size (D)? Any principled scheme (e.g., D-optimal design) or adaptive refinement? How sensitive is DSR to suboptimal choices of (Z)?
2. Do the proofs or algorithms extend to sub-Gaussian noise, and do your empirical conclusions persist when the linear model is slightly misspecified?

---

> ### Author Response · Authors · 2025-11-21
>
> Thank you for your helpful and insightful suggestions. We appreciate the opportunity to clarify and address your concerns.
>
> **Comment 1:**
>
> > Tightness of the relaxed bound in practice. While Lemma 1 and the appendix give a constant-factor relaxation $\mathcal{U^\star}\leq C\mathcal{H}^\star$, the paper does not quantify the empirical gap between $\mathcal{U^\star}$ and $\mathcal{H}^\star$. (e.g., as a function of geometry, noise, or constraints). Since DSR targets $\mathcal{U^\star}$,  understanding when $\mathcal{U^\star}\approx \mathcal{H}^\star$ is critical to interpreting the practical optimality of DSR. A plot of $\mathcal{U}^\star/\mathcal{H}^\star$  over instances would strengthen the story.
>
> **Response 1:**
>
> Thank you for your helpful suggestion. In Appendix A.7, we clarify that, in theory, the bound for $\mathcal{U}^*(\mathcal{P})$ becomes tight when the objective values of the arms in$\mathcal{D}_3(c_j)$ are close to that of the best arm, implying that arms in $\mathcal{D}_3(c_j)$ can be easily identified as infeasible rather than suboptimal. Following your suggestion, we also include a numerical experiment in Appendix A.7 of the revised version to evaluate the tightness of the surrogate objective function. We randomly generate problem instances and compute the empirical ratio between the original and surrogate objectives. The results show that the approximation ratio is close to $1$ as the constraint threshold $b$ increases.
>
> **Comment 2:**
>
> > The main text evaluates only a small synthetic instance (K=4 arms, M=2 covariates, one constraint), with heavier experiments relegated to the appendix and limited discussion of ablations (e.g., scaling in K, M, D; multiple constraints; noise levels).
>
> **Response 2:**
>
> Due to page limits, we have to include many experimental results in the Appendix. We have added additional numerical results to evaluate the performance of DSR under varying problem scales and noise levels in Appendix A.13 of the revised version. Our conclusions remain consistent across these different settings.
>
> **Comment 3:**
>
> > The results hinge on linearity and Gaussian noise; Assumption 1 excludes ties and arms on the constraint boundary. While common in theory, it would help to show robustness to mild misspecification (e.g., sub-Gaussian noises).
>
> **Response 3:**
>
> In the revised version, we relax the boundary assumption by identifying an $\epsilon$-optimal and feasible arm in Appendix A.3. We also include new numerical results to evaluate the robustness of our method under different noise distributions; see **Response 5** for details.
>
> **Comment 4:**
>
> > In large arm–covariate spaces, how do you construct the fixed set (Z) of size (D)? Any principled scheme (e.g., D-optimal design) or adaptive refinement? How sensitive is DSR to suboptimal choices of (Z)?
>
> **Response 4:**
>
> There are many methods in the experimental design literature for selecting the fixed set $Z$. Examples include space-filling designs [1] and Latin hypercube sampling [2]. Other experimental design criteria, such as $D$-optimality or $G$-optimality, also provide good choices. We compare the sample complexity of DSR using three groups of different initial design points in the current numerical examples. The results show that, although different initial design points do lead to variations in sample complexity, the differences are not substantial. This indicates that DSR is relatively robust to the choice of initial design points. The results are presented in Appendix A.13 of the revised manuscript.
>
> [1] Ankenman B, Nelson B L, Staum J. Stochastic kriging for simulation metamodeling[J]. Operations research, 2010, 58(2): 371-382.
>
> [2] Shen H, Hong L J, Zhang X. Ranking and selection with covariates for personalized decision making[J]. INFORMS Journal on Computing, 2021, 33(4): 1500-1519.
>
> **Comment 5:**
>
> > Do the proofs or algorithms extend to sub-Gaussian noise, and do your empirical conclusions persist when the linear model is slightly misspecified?
>
> **Response 5:**
>
> The algorithm can be extended to sub-Gaussian distributions in a straightforward manner. For two sub-Gaussian distributions with means $\theta$ and $\theta^{\prime}$ and a shared surrogate variance $\sigma^2$, we have
>
> $$KL(\theta ||\theta^\prime) \geq \frac{(\theta - \theta^\prime)^2}{2\sigma^2}$$
>
> Thus, the sample complexity under the Gaussian assumption serves as an upper bound for the sub-Gaussian case. The algorithm remains valid and effective, with guarantees still holding. We also evaluate the algorithm’s performance when the Gaussian noise assumption is violated, as reported in Appendix A.13 of the revised version. In this example, the problem setting remains the same, but the noise follows a standard $t$-distribution with 3 degrees of freedom, scaled by 0.1. The proposed DSR method continues to outperform the other benchmarks.

---

### Official Review · Reviewer_3yJn · 2025-10-28

**Soundness:** 3
**Presentation:** 3
**Contribution:** 2
**Rating:** 4
**Confidence:** 3

**Summary:**

This paper studies the constrained linear best arm identification (BAI) problem under the fixed-confidence setting. It proves an instance-dependent sample-complexity lower bound and argues that a Track-and-Stop–based algorithm achieves asymptotically optimal performance. The authors introduce a surrogate optimization problem for the lower bound, propose an algorithm that solves this surrogate, and provide performance guarantees. Experiments are used to demonstrate the method's effectiveness.

**Strengths:**

- Presents an effective approach to the constrained linear BAI problem.
- Shows strong empirical performance on both synthetic and real datasets.
- Most of the proof was sound (though I don't see that much novelty) for me.

**Weaknesses:**

The design points are assumed to be fixed and finite, which can be a limitation for practical applications.

The main weakness, in my view, is that the significance of the duality-based decomposition algorithm (the paper's primary contribution) may not be clearly established. In particular, its advantage over the Track-and-Stop–based approach may not be clear.
- In Appendix A.4, the authors states that the optimal sampling ratio set $\mathcal{M}^*$ is convex. Is there a proof for this statement?
- Is there evidence that $\Gamma$ is non-convex? If $\Gamma$ were convex, using standard convex optimization, one could then execute a Track-and-Stop procedure in polynomial time.
- (If above is the case) how much computational savings does the proposed method actually deliver? Since the proposed duality-based method also relies on convex optimization, the paper should report the trade-off between statistical performance and computational cost.
- Lemma 1 states an upper bound on $\mathcal{H}^* $ by $\mathcal{U}^*$, but the tightness of this bound is not clarified.

The paper mentions that it proves $ \mathcal{U}^* \le C \mathcal{H}^*$ for some constant $C > 0$ (around line329); however, the proof in Appendix A.6 appears to be for a surrogate $\tilde{U}$, and the relationship between $\tilde{U}$ and $U$ is not clearly specified, making the claim a bit imprecise.

**Questions:**

1) Appendix A.4 says the optimal sampling ratio can be non-unique. Could you provide a concrete example and intuition for when non-uniqueness arises?

See also Weaknesses. I’m happy to update my score based on the rebuttal/clarifications.

---

> ### Author Response · Authors · 2025-11-21
>
> Thank you for your insightful and constructive feedback, which has helped us further improve the paper.
>
> **Comment 1:**
>
> > Most of the proof was sound (though I don't see that much novelty) for me.
>
> **Response 1:**
>
> We briefly outline the novel proof techniques as follows:
>
>  * The construction of the alternative instance incorporating covariates, feasibility, and optimality is novel. The detailed analysis of the most indistinguishable instance for each arm class leads to new sample complexity results in Theorem 1.
>
>  *  The derivation of a surrogate objective function with a constant relaxation gap is novel. It simplifies the complexity of the original objective, making it more tractable while ensuring performance guarantees.
>
> *  An interesting duality-based analysis, along with the strong duality and convexity results in Theorem 2 and Lemma 2, is novel. While existing analyses typically focus on the primal problem, which is more complex in our setting, the dual formulation leads to a simpler problem that inspires the algorithm design. The established strong duality and convexity also ensure convergence. Our analysis provides a new theoretical perspective for BAI, it conveys a message that when the primal problem is difficult, maybe its dual problem is easy to solve.
>
> *  The novelty of Theorem 3 lies in establishing the convergence of the decomposition algorithm for the dual problem. In the analysis of Track-and-Stop, it amuses there is an optimization oracle, the convergence of empirical sampling ratio is straightforward due the the law of large numbers and continuity argument. However, we make a non-trial analysis on the convergence of duality-based decomposition algorithm in Lemma 8, which is more challenging since the parameter in objective function is the empirical estimate, we need to eliminate the noise effect and establish the convergence of the gradient update step in the analysis.
>
> These novel techniques provide valuable insights for designing efficient and approximately optimal algorithms for complex BAI problems, such as the constrained BAI with covariate selection considered in this work.
>
> **Comment 2:**
>
> > The design points are assumed to be fixed and finite, which can be a limitation for practical applications.
>
> Although the fixed and finite set of design points might be a limitation, it is a well-established formulation, commonly known as the transductive bandit setting in linear BAI [1,2]. It reflects practical constraints where some arm-covariate pairs are infeasible or costly to sample, requiring the agent to infer performance from a fixed, accessible set. This approach is also standard in simulation and experimental design literature [3–5], where carefully chosen design points provide sufficient information. We adopt this setting for several reasons. Theoretically, De la Garza’s result [6] shows that in linear regression, sampling from $D$ design points can capture as much information as sampling from more than $D$ points. Practically, this reduces computational burden in large-scale problems and aligns with real-world constraints (e.g., switching costs). By reducing the problem to only $D$ design points, we gain efficiency without losing essential information. Technically, the fixed design decomposes the regression variance in a structured way and enables efficient algorithm design.
>
> [1] Fiez T, Jain L, Jamieson K G, et al. Sequential experimental design for transductive linear bandits[J]. Advances in neural information processing systems, 2019, 32.
>
> [2] Hübotter J, Treven L, As Y, et al. Transductive active learning: Theory and applications[J]. Advances in Neural Information Processing Systems, 2024, 37: 124686-124755.
>
> [3] Ankenman B, Nelson B L, Staum J. Stochastic kriging for simulation metamodeling[J]. Operations research, 2010, 58(2): 371-382.
>
> [4] Wang T, Xu J, Hu J Q, et al. Optimal computing budget allocation for regression with gradient information[J]. Automatica, 2021, 134: 109927.
>
> [5] Yu K, Bi J, Tresp V. Active learning via transductive experimental design[C]//Proceedings of the 23rd international conference on Machine learning. 2006: 1081-1088.
>
> [6] De la Garza A. Spacing of information in polynomial regression[J]. The Annals of Mathematical Statistics, 1954, 25(1): 123-130.

---

> > ### Author Response · Authors · 2025-11-21
> >
> > **Comment 3:**
> >
> > > In Appendix A.4, the authors states that the optimal sampling ratio set $\mathcal{M}^*$ is convex. Is there a proof for this statement?
> >
> > **Response 3:**
> >
> > The convexity of the set $\mathcal{M}^*$ is a direct consequence of Berge’s theorem. On Page 17 of the revised version, we have clarified that the assumptions required for Berge’s theorem are satisfied in our setting.
> >
> > **Comment 4:**
> >
> > > Is there evidence that $\Gamma$ is convex is non-convex? If $\Gamma$ were convex, using standard convex optimization, one could then execute a Track-and-Stop procedure in polynomial time.
> >
> > **Response 4:**
> >
> > *  $\Gamma$ is convex, the analysis proceeds identically to that in Lemma 2, where we establish the convexity of $\Gamma^{s}$.
> >
> > * Track-and-Stop proposed in [1] assumes access to an optimization oracle that outputs the optimal sampling ratio. This oracle must be invoked at every iteration, which is computationally expensive even when the underlying problem is convex.
> >
> >  * The original Track-and-Stop algorithm is designed for canonical BAI, where the KKT conditions admit closed-form expressions and the resulting optimization can be solved via a simple bisection method. In contrast, our setting requires solving a significantly more complex multi-layer optimization problem (with an additional layer induced by covariates). The subproblems differ across arm classes, the constraints lead to more complex objective functions, the KKT conditions do not have closed-form solutions, and evaluating the objective further involves costly matrix inversions. These key differences render existing approaches impractical and motivate the development of new algorithms.
> >
> > * Executing a Track-and-Stop procedure with an optimization oracle in our setting would therefore be prohibitively time-consuming.
> >
> > [1] Garivier A, Kaufmann E. Optimal best arm identification with fixed confidence[C]//Conference on Learning Theory. PMLR, 2016: 998-1027.
> >
> > **Comment 5:**
> >
> > > How much computational savings does the proposed method actually deliver? Since the proposed duality-based method also relies on convex optimization, the paper should report the trade-off between statistical performance and computational cost.
> >
> > **Response 5:**
> >
> > We compare the one-step computational complexity in Appendix A.11. Since the main difference between the Algorithm 1 (TS) and the proposed Algorithm (DSR) lies in how the empirical optimal sampling ratio is computed, we focus on this step. In TS, assuming gradient descent is used, evaluating the objective function involves a matrix inversion $\mathcal{O}(D^3)$, an inner minimization over $K$ arms and $M$ covariates $\mathcal{O}(MK)$, and gradient computation $\mathcal{O}(D)$, leading to a total per-iteration complexity of $\mathcal{O}\left(\frac{1}{\epsilon}(D^3 + MK + D)\right)$ for $\epsilon$-accuracy. In DSR, only one gradient step is performed per iteration. The matrix inversion involved in the dual objective function is done once and reused, while each iteration involves objective evaluation and descent direction computation $\mathcal{O}(MK)$ and line search $\mathcal{O}(\log(1/\epsilon'))$ for precision $\epsilon'$, resulting in a total per-iteration complexity of $\mathcal{O}(MK + \log(1/\epsilon'))$.
> >
> > **Comment 6:**
> >
> > > Lemma 1 states an upper bound on $\mathcal{H}^\star$ by $\mathcal{U}^\star$, but the tightness of this bound is not clarified.
> >
> > **Response 6:**
> >
> > In Appendix A.6, we establish a bound on the relaxation gap, i.e., $\mathcal{U}^\star(\mathcal{P})\leq C\mathcal{H}^\star(\mathcal{P})$ for some instance-dependent constant $C>1$. The explicit expression for $C$ is provided in Appendix A.6. We also include a numerical example in the revised version to illustrate the tightness of this approximation. The results show that the surrogate objective closely matches the original objective value.

---

> > > ### Author Response · Authors · 2025-11-21
> > >
> > > **Comment 7:**
> > >
> > > The paper mentions that it proves $\mathcal{U}^\star(\mathcal{P})\leq C\mathcal{H}^\star(\mathcal{P})$ for some constant $C>0$; however, the proof in Appendix A.6 appears to be for a surrogate $\tilde{\mathcal{U}}$, and the relationship between $\tilde{\mathcal{U}}$ and $\mathcal{U}$  is not clearly specified, making the claim a bit imprecise.
> > >
> > > **Response 7:**
> > >
> > > The upper bound constant $C$ for $\mathcal{U}^\star(\mathcal{P})$ is instance-dependent. This bound becomes tight (with $C$ approaching $1$) when the objective values of the arms in $\mathcal{D}_3(c_j)$ are close to that of the best arm , and for certain problem instances, the constant $C$ may indeed equal $1$. To obtain an instance-independent bound, we derive the upper bound $\tilde{\mathcal{U}}^*(\mathcal{P})$, which satisfies $\mathcal{H}^\star(\mathcal{P})\leq \tilde{\mathcal{U}}^\star(\mathcal{P})\leq 2\mathcal{H}^\star(\mathcal{P})$. Since the theoretical analysis of these two bounds is essentially the same, we focus on $\mathcal{U}^\star(\mathcal{P})$ in the main paper for simplicity, as its implementation is more straightforward than that of ${\mathcal{U}}^\star(\mathcal{P})$.
> > >
> > > **Comment 8:**
> > >
> > > Appendix A.4 says the optimal sampling ratio can be non-unique. Could you provide a concrete example and intuition for when non-uniqueness arises?
> > >
> > > **Response 8:**
> > >
> > > The non-unique sampling ratios in linear BAI stem from the fact that sampling one arm may provide information that is also obtainable by sampling another arm. For example, consider three arms with feature vectors $x_1 = (1,0),x_2=(0,1)$ and $x_3 = (-1,0)$; and let the regression parameter be $\theta = (1,1)$. In this case, sampling arms $x_1$ and $x_3$ provides the same information about $\theta$, since they differ only by a sign in the first coordinate. Consequently, any allocation of the form $(\alpha (1-\omega^\star_2),\omega^\star_2,(1-\alpha) (1-\omega^\star_2)), \alpha\in[0,1]$, is an optimal sampling ratio.

---

### Official Review · Reviewer_6UNn · 2025-11-01

**Soundness:** 3
**Presentation:** 3
**Contribution:** 2
**Rating:** 6
**Confidence:** 3

**Summary:**

This paper studies a $\delta$-PAC, constrained best arm identification (BAI) algorithm that identifies the best arm across multiple covariates simultaneously. The authors formulate a problem-specific lower bound and devise a track-and-stop algorithm that tracks this lower bound. However, since the bound is computationally intractable, they introduce a convex dual and a corresponding decomposition algorithm, which substantially improves computational efficiency.

**Strengths:**

If the claim that this problem has not been studied before is correct, computing a problem-specific lower bound itself constitutes a nontrivial contribution. Consequently, the naturally arising track-and-stop algorithm is an excellent algorithm that, from the perspective of existing BAI methods, is asymptotically optimal. Furthermore, in this line of work, many track-and-stop methods submit even when the optimal ratio is intractable, citing only theoretical value. This paper goes further by considering the dual form and achieving substantial numerical improvements in computational capability.

**Weaknesses:**

Weaknesses:

1) About the problem setting itself: In the main text, covariates become, effectively, another action that a learning agent can control. Although one cannot freely select $x_i$, if one can choose among a list of pairs $(x_i, c_j)$, then the somewhat lengthy and complex $\phi(x_i, c_j)$ could simply be used as a vector in the form $a_h$ (the authors abbreviate this as $z_h$). From this perspective, even with Assumption 2, the problem seems to collapse into a relatively ordinary linear BAI with linear constraints. While the fact that the constraint changes per $z_h$ (more precisely per covariate $c_j$) is annoying, it is not difficult to imagine that the results would be derived as incremental changes on the linear BAI foundation.

1-1) (Minor) In that light, I would appreciate if the authors could modify the phrase before Eq. (1) that starts with “given a covariate $c_j$ ...”. As a reader, this phrasing leads me to intuit covariates observed passively and drawn randomly, as mentioned in the related works under covariate selection, which caused some confusion when reading the latter parts.

1-2) In fact, up to Section 3.2 there is not much that I find surprising. The work increasingly feels multi-objective with constraints appearing and disappearing, a linear BAI flavored pattern that is hard to shake off.

**Questions:**

0) About sample complexity: is there any aspect worth mentioning as a technical novelty? Are there analytical difficulties arising from partitioning the sets into $D_1, D_2, D_3$?

1) If there is an interesting part, it is the dual section. As a reviewer, I currently view this paper through the lens of linear BAI, and I am curious about the scalability of this result.

1-1) In canonical settings, an efficient ratio-computation method is known. But is the dual form and the associated efficient computation feasible in a covariate-passive, linear BAI environment? Does the constraint help or hinder computing the optimal ratio more efficiently than baseline methods?

1-2) Is this method reducible to existing approaches? For example, can it be applied to linear BAI with a single covariate (i.e., standard linear BAI) to compute the ratio efficiently?

2) Do the constraint and reward variance $\sigma_{ij}$ need to be equal always (e.g., Assumption 3, Eq. (8))? Are there properties that can be derived using this specific property? Also, must the distributions be Gaussian? If there are any Gaussian-specific lemmas leveraged, please mention them.

If possible during the discussion period, I will try to understand the value of the dual form more deeply; as of now, my score is slightly above the threshold.

---

> ### Author Response · Authors · 2025-11-21
>
> Thank you for your valuable comments and suggestions. We appreciate the opportunity to address your concerns and further improve our paper.
>
> **Comment 1:**
> >  It is not difficult to imagine that the results would be derived as incremental changes on the linear BAI foundation.
>
> **Response 1:**
>
> The problem does not reduce to a standard linear BAI with linear constraints. The introduction of covariate selection adds another layer of optimization in the sample complexity lower bound in Theorem 1, which increases the number of non-smooth points in the objective function and creates additional challenges for algorithm design. The newly added numerical results in Appendix A.13 of the revised version demonstrate that active covariate selection is important for reducing the sample complexity.
>
> The constraint setting is not an incremental changes. It introduces distinct theoretical and algorithmic challenges. Constrained optimization is fundamentally different from its unconstrained counterpart, requiring arms to be classified by both feasibility and optimality. Algorithms must balance exploration to learn both the optimality and feasibility of arms. As demonstrated in our numerical results, BAI methods that ignore feasibility constraints often perform poorly in this setting.
>
> **Comment 2:**
> >   I would appreciate if the authors could modify the phrase...
>
> **Response 2:**
> We have modified this sentence as "The agent aims to solve the following stochastic optimization problem..." .
>
> **Comment 3:**
> >  In fact, up to Section 3.2 there is not much that I find surprising. The work increasingly feels multi-objective with constraints appearing and disappearing, a linear BAI flavored pattern that is hard to shake off.
>
> **Response 3:**
>
> We would like to emphasize that our covariate selection and constraint formulation is novel and differs from both multi-objective BAI and the standard linear BAI setting. Multi-objective BAI is fundamentally different from constrained BAI: in multi-objective BAI, the agent seeks to optimize across multiple performance metrics simultaneously and identify the Pareto set. In contrast, constrained BAI requires satisfying hard constraints on certain metrics, and the goal is to identify the best feasible arm.
>
> Decision-making with covariate information is a central research topic in many fields. To the best of our knowledge, we are the first to consider active covariate selection in the constrained BAI setting and to provide a tight sample complexity lower bound. This setting introduces nontrivial challenges for algorithm design, as discussed in **Response 1**.
>
> **Comment 4:**
> > About sample complexity: is there any aspect worth mentioning as a technical novelty? Are there analytical difficulties arising from partitioning the sets into
>
> **Response 4:**
>
> The construction of alternative instances that jointly incorporate covariates, feasibility, and optimality is novel. The set of alternative instances is defined as
>
> $\mathcal{A}(\mathcal{P}) =\\{ \mathcal{P}\in \mathcal{S}: \exists c_j \in \mathcal{C}, x_{i^\star(c_j,\mathcal{P})} \neq x_{i^\star(c_j,\tilde{\mathcal{P}})}\\}$
>
>  We provide a nontrivial decomposition of this set into several subclasses according to covariate, feasibility, and optimality structure. For each subclass, the least distinguishable alternative instance is obtained by solving a corresponding convex optimization problem. For example,
>
>  $\inf_{\tilde{\mathcal{P}}\in\mathcal{O}(x_i,c_j)}\mathcal{H}(\omega, \mathcal{P}, \tilde{\mathcal{P}})$
>
> The subset $\mathcal{O}(x_i,c_j)$ encodes the feasibility and optimality requirements of the arm, making it substantially more complex to analyze compared with the canonical or linear BAI setting.
>
> A careful analysis of the optimal value of each problem then yields the new sample complexity results in Theorem 1. Compared to existing linear BAI lower bounds, our formulation introduces an additional layer of optimization over the covariate, i.e. $\min_{c_j\in \mathcal{C}}\Gamma(\omega,c_j,\mathcal{P})$ and it explicitly reveals how arms in different subclasses contribute to the overall sample complexity in distinct ways, depending jointly on their feasibility and optimality characteristics.

---

> > ### Author Response · Authors · 2025-11-21
> >
> > **Comment 5:**
> > > In canonical settings, an efficient ratio-computation method is known. But is the dual form and the associated efficient computation feasible in a covariate-passive, linear BAI environment? Does the constraint help or hinder computing the optimal ratio more efficiently than baseline methods?
> >
> > **Response 5:**
> >
> > We can also derive a dual formulation and design a corresponding algorithm for the covariate-passive linear BAI setting, which is a special case of our framework and is relatively easier to handle. The constraint makes computing the optimal ratio considerably more challenging. Due to the constraint, each arm class leads to a distinct optimization problem. In particular, the case of infeasible arms with worse performance is the most involved, as it requires simultaneously accounting for both optimality and feasibility considerations. The resulting subproblems differ across arm classes, the constraints introduce more complex non-smooth objective functions, the KKT conditions do not admit closed-form solutions, and evaluating the objective often requires computationally expensive matrix inversions. These difficulties render existing approaches inapplicable in our setting.
> >
> > **Comment 6:**
> >
> > > Is this method reducible to existing approaches? For example, can it be applied to linear BAI with a single covariate (i.e., standard linear BAI) to compute the ratio efficiently?
> >
> > **Response 6:**
> >
> > The dual method is general enough to handle the linear BAI problem, which appears as a special case of our framework with a single covariate and all arms known to be feasible. It can also accommodate the canonical BAI setting without imposing any linear assumptions on the reward function, and it is more efficient than repeatedly invoking a numerical solver to compute the optimal ratio in each iteration.
> >
> > The key message is that we introduce an interesting dual perspective for the complicated BAI problem. When the primal problem is difficult to solve, we can instead derive its dual formulation, which has a more favorable structure and can be solved efficiently, for example using the decomposition method we propose. To the best of our knowledge, this dual-based perspective is the first to be introduced in the BAI literature.
> >
> > **Comment 7:**
> >
> > > Do the constraint and reward variance $\sigma^2_{ij}$ need to be equal always (e.g., Assumption 3, Eq. (8))? Are there properties that can be derived using this specific property? Also, must the distributions be Gaussian? If there are any Gaussian-specific lemmas leveraged, please mention them.
> >
> > **Response 7:**
> >
> > The constraint and reward variance $\sigma^2_{ij}$ do not need to be equal in general. A useful property in the equal-variance case is that the linear regression parameters $\hat{\theta}$ and $\hat{\beta}$ can be computed more efficiently. The GLS estimator is given by $(X^\top \Sigma^{-1} X)^{-1}X^\top \Sigma^{-1}\bar{Y}$. When the variances are the same, $\hat{\theta}$ and $\hat{\beta}$ share the same $\Sigma^{-1}$, which reduces computational cost. If the variances differ, this term needs to be computed separately for each case and does not affect the theoretical analysis.
> >
> > The analysis extends to distributions within a single-parameter exponential family. For Gaussian distributions, however, the KL divergence has a closed form, which allows us to derive a closed-form, tight sample complexity lower bound as stated in Theorem 1. This bound can also be applied to sub-Gaussian noise (see **Response 5** for Reviewer pghX), although at the cost of some tightness.

---

### Author Response · Authors · 2025-11-29
**Summary of Rebuttal for Area Chair Consideration**

Dear Area Chairs,

Please find below a summary of the reviewers’ comments and our corresponding responses:

**Response to reviewer 6UNn:**

Reviewer 6UNn acknowledges the **nontrivial lower-bound contribution** and the interest of the **duality-based algorithm**.


- **Difference with Linear BAI:**  We clarify the distinct theoretical and algorithmic challenges. As noted by Reviewers pghX and Ey3L, the constrained linear BAI with covariate selection is novel and interesting.

- **Sample Complexity:** We clarify the technical novelty underlying the derivation of the sample-complexity lower bound.

- **Generalizability of Algorithm:** We clarify that our algorithm also solves linear and canonical BAI as special cases.

- **Others:** We clarify the variance assumption and extension to sub-Gaussian cases.

**Response to reviewer 3yJn:**

Reviewer 3yJn acknowledges the **algorithm’s effectiveness** and **strong empirical performance**.

- **Technical Novelty:** We clarify the main technical novelties of our proof, including the alternative-instance construction, surrogate objective, duality-based analysis, and algorithm optimality.

- **Design Points:**  We justify the rationality of using fixed, finite design points, supported by existing literature.

- **Algorithm Motivation:** We analyze the one-step computational complexity of the dual algorithm and Track-and-Stop, showing significant savings, and clarify that Track-and-Stop relies on an optimization oracle, making it prohibitively time-consuming.

- **Others:** We provide a detailed proof for a statement, clarify the relationship between surrogate objectives, and include a linear BAI example to address the reviewer’s question.

Reviewer 3yJn is willing to update the score based on the rebuttal and clarifications.


**Response to reviewer pghX:**

Reviewer pghX acknowledges the **novel setting**, **derivation of the lower bound**, **strong empirical results**, and **algorithmic novelty**.

- **Tightness of Surrogate Problem:** Following the reviewer’s suggestion, we show that the approximation gap is narrow using randomly generated instances.

- **Ablation Studies:** We include new numerical results demonstrating the effects of problem scale, noise levels, and other factors.

- **Robustness:** We extend results to boundary cases and provide numerical evidence for performance under varying noise.

- **Design Points Selection:** We present literature on design point selection and present numerical results showing robustness across different choices.

Reviewer pghX notes that most of the feedback and questions have been addressed and intends to maintain a positive score.

**Response to reviewer Ey3L:**

Reviewer Ey3L acknowledges the **interesting setting** and **novel dual formulation**, and **algorithm design**.

- **Covariate Selection:** Following the reviewer's suggestion, we included numerical results to show the importance of the covariate selection.

- **Assumption 1:** Following the reviewer's suggestion, we included a new section to extend our results to the boundary case.

- **Technical:** We added proof details for the part that the reviewer thinks is nontrivial and corrected some typos.

- **Algorithm Motivation:** Following the reviewer's suggestion, we added a new section to clarify why FWS is unsuitable for our problems and included new numerical results to show that the dual algorithm outperforms FWS both in performance and computation time.

Reviewer Ey3L notes that some concerns have been addressed and has accordingly increased the score for the paper.

We hope the Area Chair will consider our rebuttal in evaluating the paper.

Thank you very much for your time and attention!

---

### Meta-Review · Area_Chair_cV6B · 2025-12-29

**Summary:**

This paper tackles the problem of constrained linear best arm identification (BAI) while considering the implications of covariate selection in a fixed-confidence framework. The authors derive an instance-dependent sample complexity lower bound through a multi-level optimization problem. They establish a connection between feasibility and optimality, leading to a lower bound. To address computational challenges, they propose a relaxation using a duality-based decomposition algorithm, which updates fewer coordinates and performs efficiently. Experimental results show the algorithm's effectiveness.

Reviewers' concerns summarized:

Novelty and Contribution: Questions were raised about the paper's novelty regarding established linear BAI frameworks and the significance of the duality-based algorithm compared to the Track-and-Stop method.

Sample Complexity: Reviewers queried the discussions around the sample complexity, its technical novelty, and how the constraints affect the computation of optimal ratios compared to existing methods.

Empirical Evidence: Some reviewers noted a lack of concrete examples demonstrating the advantages of active covariate selection over passive scenarios and suggested empirical experiments to highlight this. In particular, there were requests for further clarification on why the proposed method was superior to the Frank–Wolfe Sampling method in this context, including empirical results to validate claims.

**Reviewer Concerns:**

The authors provided clarifications on the distinct challenges posed by covariate selection, asserting that their work presents a novel formulation of the BAI problem that incorporates both feasibility and optimality.

They acknowledged and corrected typographical errors, clarified the relationships between variables in their proofs, and expanded discussions on sample complexity and convergence of their algorithms.

Empirical comparisons showing how active covariate selection reduces sample complexity were included, supporting their claims with numerical experiments. They also demonstrated that their duality-based algorithm significantly outperformed the Frank–Wolfe method in terms of running time and accuracy in correctly identifying the best arm.

However, as the AC, I also went through the paper in some detail. I was unconvinced about two points, which made me tilt the decision to a rejection.

In (68), it is shown that $H(P)\le \tilde{U}(P)\le 2 H(P)$. However, the relation of $\tilde{U}(P)$ to $U(P)$, the actual upper bound, is unclear. Furthermore this critical inequality is not proved, but the authors merely mentioned that "we can show that" in the line below (67). This type of inequalities show the tightness between the upper and lower bounds, and hence, is critical in a theory paper. Without it, how close the algorithmic guarantee is to the lower bound remains unclear.

Next, some empirical comparisons to an LUCB-type algorithm for constraints such as VA-LUCB (Hou et al. 2023) would be good. It is well known that track-and-stop-type algorithms generally perform better that elimination-type ones. Hence, it is a good practice to compare the TaS-type algorithm here to elimination-type algorithms to demonstrate the former's superiority.

**Reviewer Scores:**

While Reviewer 3yJn mentioned that they are willing to increase their score, one of the questions was also regarding the gap between $H^*$ and $U^*$, which is also a key concern of mine. Ultimately, they didn't increase their score from 4.

I acknowledged that Reviewer Ey3L increased their score for the paper.

Overall, Reviewer 3yJn's concern regarding the tightness was not adequately addressed during the rebuttal phase and, to me, this is the main weakness of the paper. Hence, my rejection decision.

---

### Decision · Program_Chairs · 2026-01-26

Reject